# Understanding the Generalization of Stochastic Gradient Adam in Learning Neural Networks

**Xuan Tang**[1]  **Han Zhang**[1]  **Yuan Cao**[1]  **Difan Zou**[1,2]
[1]School of Computing and Data Science, The University of Hong Kong
[2]Institute of Data Science, The University of Hong Kong
{xuantang8,hzhang23}@connect.hku.hk, {yuancao,dzou}@hku.hk

## Abstract

Adam is a popular and widely used adaptive gradient method in deep learning, which has also received tremendous focus in theoretical research. However, most existing theoretical work primarily analyzes its full-batch version, which differs fundamentally from the stochastic variant used in practice. Unlike SGD, stochastic Adam does not converge to its full-batch counterpart even with infinitesimal learning rates. We present the first theoretical characterization of how batch size affects Adam's generalization, analyzing two-layer over-parameterized CNNs on image data. Our results reveal that while both Adam and AdamW with proper weight decay $\lambda$ converge to poor test error solutions, their mini-batch variants can achieve near-zero test error. We further prove Adam has a strictly smaller effective weight decay bound than AdamW, theoretically explaining why Adam requires more sensitive $\lambda$ tuning. Extensive experiments validate our findings, demonstrating the critical role of batch size and weight decay in Adam's generalization performance.

## 1   Introduction

Adaptive gradient methods, such as Adam (Kingma and Ba, 2015) and its variant AdamW (Loshchilov and Hutter, 2019), have emerged as widely adopted optimizers for training deep learning models across diverse tasks (He et al., 2016; Ma and Hovy, 2016). More recently, Adam and its variants have also been used to train large language models (LLMs) like GPT (Brown et al., 2020), LLaMA (Touvron et al., 2023), and Deepseek (Bi et al., 2024). In practice, Adam is known for its fast convergence during training, yet its generalization performance varies significantly depending on the task. Despite its empirical success, the theoretical understanding of Adam remains incomplete, especially regarding its generalization performance.

Recent theoretical work has sought to analyze the task-dependent behavior of Adam and compare it with other optimizers like gradient descent (GD). For instance, Wilson et al. (2017) demonstrated that adaptive methods like Adam exhibit poor generalization on linear models, while GD and stochastic gradient descent (SGD) can achieve zero test error. Further, Zhou et al. (2020) theoretically characterized the generalization gap between SGD and Adam through local convergence analysis, though their work did not account for neural network architectures or test error behavior. Other studies have focused on the implicit bias of adaptive methods: Wang et al. (2022) analyzed momentum's role in generalization, proving that GD with momentum and its adaptive variants converge to the $\ell_2$ max-margin solution; Xie and Li (2024) showed that full-batch AdamW converges only to a KKT point under an $\ell_\infty$ norm constraint; and Zhang et al. (2024) established Adam's convergence to a maximum $\ell_\infty$-margin classifier in linear logistic regression with separable data. In nonconvex settings, Zou et al. (2023b) revealed that full-batch Adam and GD converge to distinct solutions with differing generalization performance, which shows that even with weight decay, Adam fails to achieve low test error in overparameterized CNNs. Following the nonconvex analysis of Adam vs.

39th Conference on Neural Information Processing Systems (NeurIPS 2025).

GD by Zou et al. (2023b), Li et al. (2025) show that Sign Gradient Descent—a sign-only surrogate for Adam (Balles and Hennig, 2018; Bernstein et al., 2018)—achieves fast convergence but poor generalization when training two-layer Transformers.

While existing theoretical analyses have provided valuable insights into the behavior of full-batch Adam, these results may not fully capture the characteristics of stochastic gradient Adam commonly used in practice. Notably, although stochastic gradient descent (SGD) and full-batch GD exhibit similar training dynamics in expectation (Bottou, 2012), stochastic gradient Adam demonstrates fundamentally different behavior from its full-batch counterpart—a distinction that persists even with vanishingly small learning rates. This gap raises important questions about how stochastic gradient Adam, particularly with small batch sizes, affects model generalization, an aspect that remains largely unexplored in current literature.

Motivated by this, in this paper, we investigate how the generalization of mini-batch Adam and AdamW differs from that of large-batch Adam. We analyze the convergence and generalization of Adam (and AdamW) with different batch sizes on two-layer over-parameterized convolutional neural networks (CNNs) for an image data model. This analysis follows the settings outlined in the recent study of full-batch Adam in Zou et al. (2023b). We also compare the sensitivity of the weight decay parameters $\lambda$ for effective weight decay in Adam and AdamW.

The main contributions of this paper are summarized as follows.

- Theorem 4.1 and 4.4 rigorously prove that in the large-batch regime, both Adam and AdamW converge to solutions with poor test error in nonconvex settings, even with proper weight decay. This extends prior results for full-batch Adam to AdamW, showing that adaptive methods inherently overfit noise in low-stochasticity training. Real-world data experiments in Figure 1 demonstrate that large-batch Adam and AdamW suffer drastic test error increases, while synthetic experiments in Appendix D confirm this failure stems from noise-dominated solutions.

- For mini-batch training, theorem 4.2 and 4.5 prove that stochastic Adam and AdamW achieve near-zero test error in nonconvex settings with appropriate weight decay. The key mechanism is twofold: (i) stochastic gradients implicitly regularize the optimization trajectory by slowing noise fitting while preserving feature learning dynamics, preventing Adam from overfitting noise patches; (ii) weight decay explicitly suppresses residual noise components. This synergy ensures convergence to solutions dominated by true features. Real-world data experiments in Figure 1 demonstrate that mini-batch Adam and AdamW significantly improve test performance, with synthetic-data experiments in Appendix D further validating our theoretical insights. Moreover, under constant $\beta_1, \beta_2$ hyperparameters, we prove stochastic Adam and AdamW *can be rigorously approximated by* SignSGD (Bernstein et al., 2018) and SignSGDW (with decoupled decay) respectively. This extends the known full-batch Adam→SignGD correspondence to stochastic regimes—a crucial advancement given mini-batch noise fundamentally modifies approximation dynamics. Our analysis in Appendix C reveals this approximation holds precisely when gradient magnitudes dominate optimization noise (e.g., $|g_{t,j,r}^{(t)}[k]| \geq \widetilde{\Theta}(\eta)$ where $\eta$ is the learning rate).

- Corollary 4.3 and 4.6 derive distinct theoretical upper bounds for weight decay parameters in nonconvex settings: Adam permits a strictly smaller maximum effective $\lambda$ than AdamW. This arises because Adam's adaptive gradient normalization amplifies the effective impact of weight decay, causing excessive regularization to destabilize updates. In contrast, AdamW's decoupled weight decay mechanism avoids this issue. Experiments in Figure 2 validate that exceeding Adam's upper bound (e.g., $\lambda > 0.05$) leads to catastrophic test error increases, while AdamW tolerates much larger $\lambda$ values (e.g., $\lambda = 0.5$) without significant performance degradation. This demonstrates that the interplay between batch size and weight decay is critical: mini-batch training enables effective regularization, but Adam's narrow tolerance demands precise $\lambda$ calibration.

The rest of paper is organized as follows. Section 2 discusses the works that are most closely related to this paper. Section 3 describes the problem settings. Section 4 presents the main results of this paper. Section 5 provides the proof outline of stochastic gradient Adam. Section 6 concludes this paper and discusses future research directions. Additional experiments and all experimental details can be found in Appendix D. All proofs are provided in the remaining appendices (Appendix A– C).

**Notation.** Scalars are denoted by lowercase letters $x, y, \ldots$, vectors by bold lowercase letters $\mathbf{x}, \ldots$, and matrices by bold uppercase letters $\mathbf{A}, \ldots$. For any integer $d \geq 1$, denote the set $[d] = \{1, \ldots, d\}$.

For $x \in \mathbb{R}$, define $[x]_+ = \max\{x, 0\}$ and $\mathrm{sgn}(x) = x/|x|$ for $x \neq 0$, $\mathrm{sgn}(0) = 0$. For $\mathbf{x} = (x_1, \ldots, x_d)^\top \in \mathbb{R}^d$, define $\|\mathbf{x}\|_p = (\sum_{i=1}^d |x_i|^p)^{1/p}$ $(p \geq 1)$ and $\mathrm{supp}(\mathbf{x}) = \{i \in [d] : x_i \neq 0\}$. For real sequences $\{a_n\}, \{b_n\}$, denote $a_n = O(b_n)$ if there exist $C, N > 0$, s.t. $|a_n| \leq C|b_n|, \forall n \geq N$; denote $a_n = \Omega(b_n)$ if $b_n = O(a_n)$; $a_n = \Theta(b_n)$ if both $O(b_n)$ and $\Omega(b_n)$ hold; denote $a_n = o(b_n)$ if for any $C > 0$, there exist $N > 0$, s.t. $|a_n| < C|b_n|, \forall n \geq N$; and denote $a_n = \omega(b_n)$ if $b_n = o(a_n)$. We write $\widetilde{O}(\cdot), \widetilde{\Omega}(\cdot), \widetilde{\Theta}(\cdot)$ to suppress logarithmic factors, $a_n = \mathrm{poly}(b_n)$ if $a_n = \Theta(b_n^D)$ for some $D > 0$, and $a_n = \mathrm{polylog}(b_n)$ if $a_n = \mathrm{poly}(\log b_n)$.

## 2   Related Work

**Adaptive Optimization Methods.**   There are a series of papers on adaptive gradient methods, including AdaGrad (Duchi et al., 2011), Adam (Kingma and Ba, 2015), AdamW (Loshchilov and Hutter, 2019), and second-order information methods (Yao et al., 2021; Liu et al., 2024). The convergence of Adam and related methods has been analyzed in a line of papers under various conditions (Chen et al., 2019; Guo et al., 2021; Défossez et al., 2022). However, some work presented the possible case where Adam fails to converge to an optimal solution even in simple one-dimensional convex settings (Reddi et al., 2018). The generalization performance of Adam has been investigated and compared with that of gradient descent in Wilson et al. (2017); Zhou et al. (2020); Zou et al. (2023b). To better understand the performance of Adam, Bernstein et al. (2018, 2019); Kunstner et al. (2023) analyzed its similarity with signGD. Similar works have also been done for AdamW (Xie and Li, 2024). Loshchilov and Hutter (2019) demonstrated that improper use of weight decay in Adam could lead to poor generalization performance, and proposed the AdamW that improves generalization in comparison to Adam. Recent work has also highlighted the role of weight decay in modern deep learning setups, showing its impact on optimization dynamics and generalization (D'Angelo et al., 2024). While $L_2$ regularization and weight decay are equivalent for standard SGD and GD (with rescaled by learning rate), that is not the case for adaptive methods like stochastic gradient Adam and full-batch Adam (Loshchilov and Hutter, 2019; Zhang et al., 2019; Zhuang et al., 2022). However, the true reason why Adam with weight decay fails to improve the generalization remains unclear. Therefore, the current understanding of how batch size and weight decay influence the generalization performance of Adam is still relatively limited.

**Implicit bias.**   Implicit bias refers to the tendency of machine learning algorithms to favor certain solutions. This phenomenon has also been studied in neural networks theoretically to understand how they generalize and converge to solutions. Lyu and Li (2019) and Ji and Telgarsky (2020) studied the implicit bias of gradient descent on the homogeneous neural networks. Kunin et al. (2023) extended the results to a wider class of networks with varying degree of homogeneity. Cai et al. (2024) focused on the large stepsize gradient descent on two-layer non-homogeneous networks. Frei et al. (2022) analyzed the implicit bias of gradient flow in two-layer fully-connected neural networks with leaky ReLU activations for nearly-orthogonal data. Kou et al. (2024) extended this results and analyzed the implicit bias of gradient descent on similar settings. For Adam and AdamW, the implicit bias of Adam have been analyzed in Wang et al. (2021, 2022); Zhang et al. (2024), and the implicit bias of AdamW has been analyzed in (Xie and Li, 2024). Recently, Cattaneo et al. (2024) showed that Adam penalizes the $\ell_1$-norm of perturbed gradients, favoring flat minima. Our work complements this view by analyzing, in a discrete-time feature learning setting, how batch size and weight decay jointly regulate noise suppression and generalization.

**Feature learning.**   There are a series of papers that studied the feature learning theory in neural networks. Allen-Zhu and Li (2020) investigated the feature learning of ensemble methods and knowledge distillation in deep learning when applied to data with multi-view features. Cao et al. (2022) examined the benign overfitting in the supervised learning of two-layer convolutional neural networks, and proved that under certain conditions on signal-to-noise ratio (SNR), arbitrary small training and test loss can be achieved. Zou et al. (2023b) compared the feature learning of full-batch Adam and GD on two-layer convolutional neural networks. It demonstrated that GD learns the features, but full-batch Adam, even with proper regularization, may still fail. Some works have studied the feature learning of contrastive learning method (Zhang and Cao, 2024), federated learning (Huang et al., 2024b) on two-layer convolutional neural networks, and multi-modal contrastive learning on single-layer ReLU networks (Huang et al., 2024a). Additionally, some papers have analyzed feature learning on other architectures, such as transformers (Jelassi et al., 2022; Li et al.,

2025), and diffusion models (Han et al., 2025a,b); and other training configurations (Zou et al., 2023a; Lu et al., 2024). Unlike the aforementioned works, this paper focuses on the feature learning of Adam and AdamW algorithms with different batch sizes on the two-layer convolutional neural networks.

## 3 Problem Setup

In this paper, we train the two-layer convolutional neural network (CNN) with Adam and AdamW on the training dataset $\mathcal{S} := \{(\mathbf{x}_i,\ y_i)\}_{i=1}^n$ of size $n$, which is generated from a data model $\mathcal{D}$. In this section, we introduce the data model $\mathcal{D}$, the two-layer CNN model, and the training details of two algorithms (Adam and AdamW) analyzed in this paper.

**Data Model.** We adopt the feature-noise patch concatenation framework from Definition 3.1, aligning with previous studies (Allen-Zhu and Li, 2020; Cao et al., 2022; Jelassi et al., 2022; Zou et al., 2023b; Huang et al., 2024b,a; Zhang and Cao, 2024; Li et al., 2025; Han et al., 2025a).

**Definition 3.1.** *Let each data point $(\mathbf{x}, y)$ consist of a feature vector $\mathbf{x} \in \mathbb{R}^{2d}$ and a label $y \in \{-1, 1\}$. The data is generated as follows:*

$$\mathbf{x} = [\mathbf{x}_1^\top, \mathbf{x}_2^\top]^\top,$$

*where $\mathbf{x}_1$ and $\mathbf{x}_2$ represent two distinct feature patches. One of these patches corresponds to the signal patch and consists of a feature vector $y \cdot \mathbf{v}$, where $\mathbf{v} \in \mathbb{R}^d$ is assumed to be a sparse vector, specifically 1-sparse. The other patch represents the noise patch and is a noise vector denoted by $\boldsymbol{\xi}$. Without loss of generality, we assume $\mathbf{v} = [1, 0, \ldots, 0]^\top$. The data is generated from the following distribution $\mathcal{D}$:*

1. *The label $y$ is generated as a Rademacher random variable with $y \in \{-1, +1\}$.*

2. *Randomly select $s$ coordinates from the set $[d] \setminus \{1\}$ with equal probability. This selection is represented by a binary vector $\mathbf{s} \in \{0, 1\}^d$. Then generate $\boldsymbol{\xi}$ from the Gaussian distribution $\mathcal{N}(\mathbf{0}, \sigma_p^2 \mathbf{I}_d)$ and apply the masking operation such that $\boldsymbol{\xi} = \boldsymbol{\xi} \odot \mathbf{s}$, where $\odot$ denotes element-wise multiplication. Finally, add feature noise to the vector $\boldsymbol{\xi}$ by updating it as $\boldsymbol{\xi} = \boldsymbol{\xi} - \alpha y \mathbf{v}$, where $\alpha \in (0, 1)$ controls the strength of the feature noise.*

3. *One of the two patches $\mathbf{x}_1, \mathbf{x}_2$ is randomly selected and is assigned as $y \cdot \mathbf{v}$, representing the signal patch, while the other patch is assigned as $\boldsymbol{\xi}$, representing the noise patch.*

*We set $s = \Theta\left(d^{1/2}/n^2\right)$, $\sigma_p^2 = \Theta\left(1/(s \cdot \mathrm{polylog}(n))\right)$, $\alpha = \Theta\left(\sigma_p \cdot \mathrm{polylog}(n)\right)$ in this paper.*

The data model formalizes image classification dynamics where localized label-relevant features coexist with global noise—aligning with CNN behaviors: sparse mid-layer activations (Papyan et al., 2017) vs. non-informative regions as independent noise (Yang, 2019). By isolating 1-sparse feature and $s$-sparse noise patches, we distill the feature learning vs. noise memorization interplay. Though our analysis uses a simplified single feature/noise patch model for clarity, the results can be extended to broader settings (e.g., multi-patch or denser features/noises) by assuming sub-Gaussian noise and using concentration inequalities (e.g., Bernstein bounds) to control overlapping or structured perturbations, with similar qualitative behavior expected as long as the total noise remains controlled.

**Two-layer CNN model.** We define the two-layer CNN considered in this paper as follows.

**Definition 3.2.** *Given the data $(\mathbf{x}, y) \sim \mathcal{D}$ and the activation function $\sigma(x) = [x]_+^q$ with $q \geq 3$, the $j$-th output of the neural network $F$ with width $m$ is*

$$F_j(\mathbf{W}, \mathbf{x}) = \sum_{r=1}^m \left[\sigma(\langle \mathbf{w}_{j,r}, \mathbf{x}_1 \rangle) + \sigma(\langle \mathbf{w}_{j,r}, \mathbf{x}_2 \rangle)\right] = \sum_{r=1}^m \left[\sigma(\langle \mathbf{w}_{j,r}, y\mathbf{v} \rangle) + \sigma(\langle \mathbf{w}_{j,r}, \boldsymbol{\xi} \rangle)\right],$$

*where $\mathbf{w}_{j,r}$ is the weight at the $r$-th neuron and initialized from Gaussian distribution $\mathcal{N}(\mathbf{0}, \sigma_0^2 \mathbf{I}_d)$. In this paper, we assume $j \in \{\pm 1\}$ for clarity, ensuring the logit index matches the data label. Additionally, we also assume $m = \mathrm{polylog}(n)$ and $\sigma_0 = \Theta(d^{-1/4})$.*

**Training algorithm.** We investigate the behavior of stochastic Adam and AdamW, starting from same initializations and training on the same dataset $\mathcal{S} = \{(\mathbf{x}_i, y_i)\}_{i=1}^n$. The loss function for each data point $(\mathbf{x}_i, y_i)$ is denoted as $L_i(\mathbf{W}) = -\log \frac{e^{F_{y_i}(\mathbf{W}, \mathbf{x}_i)}}{\sum_{j \in \{-1,1\}} e^{F_j(\mathbf{W}, \mathbf{x}_i)}}$.

For stochastic **Adam** and **AdamW**, the CNN model is trained by minimizing the empirical loss function

$$(\textbf{Adam}) \quad L(\mathbf{W}) = \frac{1}{n} \sum_{i=1}^n L_i(\mathbf{W}) + \frac{\lambda}{2} \|\mathbf{W}\|_F^2, \tag{3.1}$$

$$(\textbf{AdamW}) \quad L(\mathbf{W}) = \frac{1}{n} \sum_{i=1}^n L_i(\mathbf{W}), \tag{3.2}$$

where $\|\cdot\|_F$ denotes the Frobenius norm, $\lambda$ is the weight decay regularization of **Adam**. Therefore, the stochastic gradient $g_{t,j,r}^{(t)}$ can be calculated as

$$(\textbf{Adam}) \quad g_{t,j,r}^{(t)} = \frac{1}{B} \sum_{i \in \mathcal{I}_t} \nabla_{\mathbf{w}_{j,r}^{(t)}} L_i(\mathbf{W}^{(t)}) + \lambda \mathbf{w}_{j,r}^{(t)},$$

$$(\textbf{AdamW}) \quad g_{t,j,r}^{(t)} = \frac{1}{B} \sum_{i \in \mathcal{I}_t} \nabla_{\mathbf{w}_{j,r}^{(t)}} L_i(\mathbf{W}^{(t)}),$$

where the subscript $t$ of $g_{t,j,r}^{(t)}$ represents the batch $\mathcal{I}_t$ at the $t$-th iteration and the superscript $t$ of $g_{t,j,r}^{(t)}$ represents the model $\mathbf{W}^{(t)}$ at the $t$-th iteration. Herein, we emphasize a fundamental distinction: Adam's stochastic gradients inherently incorporate weight decay regularization, whereas AdamW's gradients remain regularization-free—a deliberate design choice to prevent momentum-based normalization from destabilizing regularization effects (Loshchilov and Hutter, 2019). This architectural distinction, also analytically demonstrated in our proof, crucially impacts the training process. The momentum estimates $\mathbf{m}_{j,r}^{(t)}$, $\mathbf{v}_{j,r}^{(t)}$ of Adam/AdamW are updated as follows

$$\mathbf{m}_{j,r}^{(t+1)} = \beta_1 \mathbf{m}_{j,r}^{(t)} + (1 - \beta_1) \cdot g_{t,j,r}^{(t)}, \tag{3.3}$$

$$\mathbf{v}_{j,r}^{(t+1)} = \beta_2 \mathbf{v}_{j,r}^{(t)} + (1 - \beta_2) \cdot [g_{t,j,r}^{(t)}]^2, \tag{3.4}$$

where $\beta_1$, $\beta_2$ are the hyperparameters of Adam/AdamW and we initialize $\mathbf{m}_{j,r}^{(0)} = \mathbf{v}_{j,r}^{(0)} = \mathbf{0}$. Finally, the update rule of stochastic Adam/AdamW for model $\mathbf{W}$ can be formulated as

$$(\textbf{Adam}) \quad \mathbf{w}_{j,r}^{(t+1)} = \mathbf{w}_{j,r}^{(t)} - \eta \cdot \frac{\mathbf{m}_{j,r}^{(t)}}{\sqrt{\mathbf{v}_{j,r}^{(t)}} + \epsilon}, \tag{3.5}$$

$$(\textbf{AdamW}) \quad \mathbf{w}_{j,r}^{(t+1)} = (1 - \eta\lambda)\mathbf{w}_{j,r}^{(t)} - \eta \cdot \frac{\mathbf{m}_{j,r}^{(t)}}{\sqrt{\mathbf{v}_{j,r}^{(t)}} + \epsilon}, \tag{3.6}$$

where $\eta$ is the learning rate, $\epsilon = \Theta(\lambda\eta)$ is stability constant and $\lambda$ is the decoupled weight decay parameter of **AdamW**. In particular, in (3.4), (3.5) and (3.6), the square $(\cdot)^2$, square root $\sqrt{\cdot}$, and division $\cdot/\cdot$ all denote entry-wise calculations. The details of gradient calculation and its expansion can be found in Appendix A.

## 4 Main Results

In this section, we present the main results of our study. We begin by introducing the primary metric used to evaluate generalization performance: the classification error rate.

Given training dataset $\mathcal{S} = \{(\mathbf{x}_i, y_i)\}_{i=1}^n$ generated from data model $\mathcal{D}$ in Definition 3.1. We define the training error $\text{err}_{\mathcal{S}}(\mathbf{W})$ and test error $\text{err}_{\mathcal{D}}(\mathbf{W})$ of model $\mathbf{W}$ as follows,

$$\text{err}_{\mathcal{S}}(\mathbf{W}) = \mathbb{E}_{(\mathbf{x},y)\sim\mathcal{S}} \mathbb{1}\left[ F_y(\mathbf{W}, \mathbf{x}) \leq F_{-y}(\mathbf{W}, \mathbf{x}) \right],$$

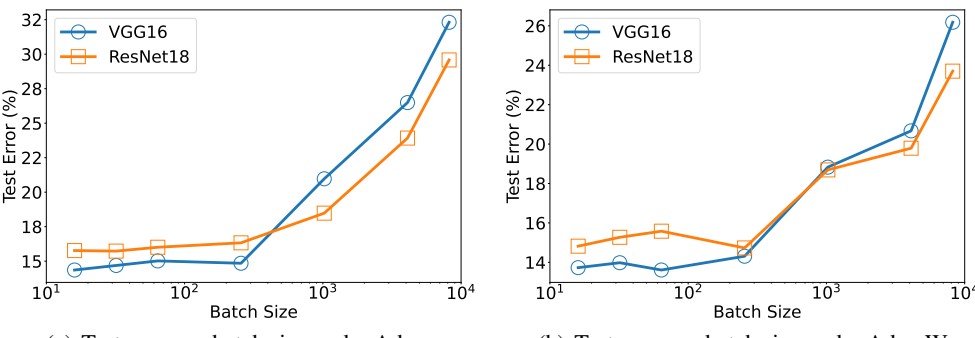

(a) Test error vs. batch size under Adam      (b) Test error vs. batch size under AdamW

Figure 1: Test error vs. batch size for VGG16 and ResNet18 on CIFAR-10.

$$\mathrm{err}_{\mathcal{D}}(\mathbf{W}) = \mathbb{E}_{(\mathbf{x},y)\sim\mathcal{D}}\, \mathbb{1}\left[F_y(\mathbf{W},\mathbf{x}) \leq F_{-y}(\mathbf{W},\mathbf{x})\right].$$

While theoretical analyses often prioritize mathematically tractable surrogate losses (e.g., cross-entropy, hinge loss), classification error rate remains the most direct and practical performance metric. Unlike continuously approximated surrogate losses, error rate directly quantifies discrete misclassification events, better reflecting models' true decision-making ability in classification tasks.

## 4.1 Theoretical Results for Adam

The following Theorem 4.1 characterizes the behavior of Adam in the large-batch regime.

**Theorem 4.1** (Large-batch Adam). *Suppose $\eta = \frac{1}{\mathrm{poly}(n)}$ and $\lambda$ satisfies $0 < \lambda = o(\frac{\sigma_0^{q-2}\sigma_p}{n})$, we train our CNN model in Definition 3.2 on loss function (3.1) for $T = \frac{\mathrm{poly}(n)}{\eta}$ epochs using Adam (3.5) with batch size $B$ satisfies $\frac{n}{B} = \Theta(1)$. Then with probability at least $1 - n^{-1}$, we have*

- *The training error is zero: $\mathrm{err}_{\mathcal{S}}(\mathbf{W}^{(T)}) = 0$.*

- *The test error is high: $\mathrm{err}_{\mathcal{D}}(\mathbf{W}^{(T)}) \geq \frac{1}{2} - o(1)$.*

Theorem 4.1 extends Zou et al. (2023b)'s full-batch analysis to large-batch regimes ($B = \Theta(n)$). Basically, it states that under the nearly same data model in Zou et al. (2023b), large-batch Adam cannot effectively learn the feature vector from the training dataset, and finally attains a nearly 0.5 test accuracy, despite its perfect fitting on the training data points.

In stark contrast, we further provide Theorem 4.2, which proves that stochastic gradient Adam with a smaller batch size can achieve good generalization performance.

**Theorem 4.2** (Mini-batch Adam). *Suppose $\eta = \frac{1}{\mathrm{poly}(n)}$ and $0 < \lambda = o(\frac{\sigma_0^{q-2}\sigma_p}{n})$, we train our CNN model in Definition 3.2 on loss (3.1) for $T = \frac{\mathrm{poly}(n)}{\eta}$ epochs using stochastic Adam (3.5) with batch size $B$ satisfies $\frac{n}{B} \geq \Theta(\log \epsilon^{-1})$, where $\epsilon$ is the hyperparameter of Adam. Then with high probability at least $1 - n^{-1}$, we have*

- *The training error is zero: $\mathrm{err}_{\mathcal{S}}(\mathbf{W}^{(T)}) = 0$.*

- *The test error is near-zero: $\mathrm{err}_{\mathcal{D}}(\mathbf{W}^{(T)}) = o(1)$.*

Our theoretical results demonstrate that mini-batch Adam achieves near-perfect test accuracy when the ratio $n/B$ is large, significantly outperforming its large-batch counterpart which exhibits only random-guessing performance. This advantage can be attributed to three fundamental properties of stochastic Adam optimization: First, since the feature vector is shared across all data points, its learning remains robust regardless of batch size. In contrast, noise vectors are data-specific and vary across different samples. When using mini-batches, only a subset of noise vectors is exposed during each update, creating an inherent asymmetry in learning dynamics. More importantly, Adam's coordinate-wise normalization amplifies this effect: it maintains consistent learning rates for shared features while substantially slowing down noise memorization. This selective suppression of

noise learning explains the superior generalization performance of mini-batch Adam compared to large-batch implementations.

Besides the results on the generalization performance, we further deliver the following corollary, which states the feasible range of the weight decay in Adam. Theorems 4.1 and 4.2 directly yield:

**Corollary 4.3** (Effective weight decay in Adam). *Suppose the same conditions as in Theorem 4.1 and 4.2. If $\lambda = \omega(\sigma_0^{q-2})$, then with probability at least $1 - n^{-1}$, training stuck at the initialization.*

This corollary provides a theoretical upper bound on the effective weight decay that allows Adam to successfully train models, aligning well with previous empirical observations that Adam typically performs better with small weight decay values compared to AdamW (Loshchilov and Hutter, 2019). This sensitivity arises because weight decay regularization is implicitly entangled with the normalization step in Adam, i.e., when the gradient of weight decay is greater than that of the cross-entropy loss, it will fully dominate the Adam update. In the next subsection, we will show that weight decay will exhibit a different behavior in AdamW, leading to a different feasible range for $\lambda$.

### 4.2  Theoretical Results for AdamW

We first establish the theoretical results of Adamw in Theorem 4.4 under large-batch training.

**Theorem 4.4** (Large-batch AdamW). *Suppose $\eta = \frac{1}{\mathrm{poly}(n)}$, $\lambda = \widetilde{\Omega}(\frac{B^2}{n} \wedge 1)$ and $\lambda = \widetilde{O}(1)$, we train our CNN model in Definition 3.2 on loss function (3.2) for $T = \frac{\mathrm{poly}(n)}{\eta}$ epochs using AdamW (3.6) with batch size $B$ satisfies $\frac{n}{B} = \Theta(1)$ or $\frac{n}{B} = o(s\sigma_p)$. Then with probability at least $1 - n^{-1}$, we have*

- *The training error is zero:* $\mathrm{err}_{\mathcal{S}}(\mathbf{W}^{(T)}) = 0$.

- *The test error is high:* $\mathrm{err}_{\mathcal{D}}(\mathbf{W}^{(T)}) \geq \frac{1}{2} - o(1)$.

The results for large-batch AdamW closely resemble those of large-batch Adam in Theorem 4.1: the learned model consistently exhibit test errors of at least $1/2 - o(1)$, performing no better than random guessing. This phenomenon arises from the training dynamics in the early stages, where the influence of weight decay is minimal. As a result, both Adam and AdamW exhibit similar behavior, tending to fit noise. By the time the decoupled weight decay in AdamW begins to take effect, the model has already overfit to the feature noise $-\alpha y\mathbf{v}$. The weight decay then guides the model toward nearby local minima, effectively preserving the previously memorized noise. Then at test time, this overfitting to feature noise causes the model to predict labels that are systematically misaligned with the true labels, leading to test performance that is no better than random guessing.

In contrast to the results for large-batch in Theorem 4.4, the following Theorem 4.5 characterizes the generalization ability of mini-batch AdamW.

**Theorem 4.5** (Mini-batch AdamW). *Suppose $\eta = \frac{1}{\mathrm{poly}(n)}$, $\lambda = \widetilde{\Omega}(\frac{B^2}{n} \wedge 1)$ and $\lambda = \widetilde{O}(1)$, we train our CNN model in Definition 3.2 on loss function (3.2) for $T = \frac{\mathrm{poly}(n)}{\eta}$ epochs using stochastic AdamW (3.6) with batch size $B$ satisfies $\frac{n}{B} \geq \Theta(\log \epsilon^{-1})$ and $\frac{n}{B} = \omega(s\sigma_p \vee n^{1/2})$. Then with probability at least $1 - n^{-1}$, we have*

- *The training error is zero:* $\mathrm{err}_{\mathcal{S}}(\mathbf{W}^{(T)}) = 0$.

- *The test error is near-zero:* $\mathrm{err}_{\mathcal{D}}(\mathbf{W}^{(T)}) = o(1)$.

Under mini-batch training, AdamW achieves near-zero test error with partial similarity to Adam. The extended iterations per epoch slow early-stage noise overfitting (notably for $s$-sparse noise). However, AdamW's decoupled weight decay penalizes weights independently of gradients, exerting significant regularization only in later phases to converge toward generalizable minima. Thus, AdamW can leverage a much larger $\lambda$ than Adam, which is shown as follows:

**Corollary 4.6.** *Regarding the effective weight decay coefficients of Adam and AdamW for achieving good generalization performance, we have $\lambda_{\mathrm{Adam}} \sim \sigma_0^{q-2} \ll \frac{B^2}{n} \wedge 1 \sim \lambda_{\mathrm{AdamW}}$.*

Corollary 4.6 reveals a fundamental gap between the effective weight decay regimes of Adam and AdamW, consistent with empirical observations. For **Adam**, the admissible weight decay $\lambda_{\mathrm{Adam}}$

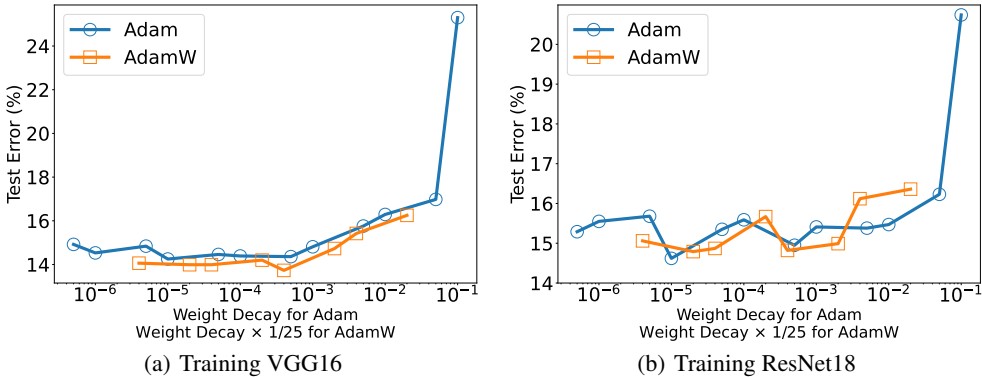

Figure 2: Test error vs. weight decay (batch size = 16), comparing Adam and AdamW on each model.

is extremely small, bounded above by an initialization-dependent term $\sigma_0^{q-2}$. Consequently, even moderate weight decay values can destabilize training due to the entanglement between gradient updates and weight decay, leading to suboptimal generalization. In contrast, **AdamW** decouples weight decay from the gradient update, applying regularization directly on the weights. As a result, it requires a larger weight decay to effectively suppress noise overfitting and exhibits greater robustness. The lower bound $\widetilde{\Omega}(\frac{B^2}{n} \wedge 1)$ serves as a sufficient condition ensuring that weight decay is strong enough to prevent noise amplification. This robustness is reflected in its much broader effective range—from $\widetilde{\Omega}(\frac{B^2}{n} \wedge 1)$ up to $\widetilde{O}(1)$—representing a large, constant-order window independent of initialization. This theoretical separation explains the empirical fact that Adam requires delicate tuning and is highly sensitive to weight decay, whereas AdamW is considerably more robust and easier to tune Loshchilov and Hutter (2019). Our experiments (Figure 2) further corroborate this prediction.

**Experiments.** We train VGG16 and ResNet18 on CIFAR-10 with Adam ($\lambda = 5 \times 10^{-4}$) and AdamW ($\lambda = 1 \times 10^{-2}$), selecting the optimal learning rate from $\{5 \times 10^{-4}, 1 \times 10^{-4}, 1 \times 10^{-5}\}$ and varying the batch size to measure test error (Figure 1). Both optimizers exhibit a sharp performance degradation once the batch size exceeds a critical threshold, in line with theoretical predictions of large-batch generalization collapse (Theorem 4.1 and 4.4). Separately, at a fixed batch size of 16 (Figure 2), we find that Adam's error spikes for $\lambda > 0.05$, whereas AdamW remains robust even up to $\lambda = 0.5$, highlighting the benefit of decoupled weight decay in adaptive optimizers. This observation is consistent with our theoretical analysis (Corollary 4.3 and 4.6). Additional experiments, including the dynamics of feature learning and noise memorization (Figures 3, 4), sensitivity to weight decay (Figures 5, 6), error bars across random seeds and momentum parameters (Figures 7, 8, 9, 10), and large-scale vision experiments with ResNet-50 on ImageNet-1K (Figures 11, 12), all corroborate our theoretical findings. Complete experimental details and results are provided in Appendix D.

## 5 Proof Outline of the Main Results

In this section, we mainly outline the proof sketch for the theorem 4.2 in Section 4. Proof sketches for remaining theorems are deferred to Appendix B. Following the two-stage analysis framework of Cao et al. (2022); Zou et al. (2023b), we decompose the proof into two distinct stages:

**Stage I: Pattern Learning.** During the initial phase of training, the effect of regularization is negligible. The model operates in an underfitting regime, where it rapidly learns dominant patterns in the training data, leading to improved empirical performance on test error.

**Stage II: Regularization.** As training progresses, the model's classification accuracy on the training set approaches convergence, resulting in diminished gradient magnitudes. At this stage, regularization dominates the optimization dynamics, steering the model converge to a local minima. Due to the nonconvex nature of the loss landscape, the model retains the patterns acquired during the pattern learning stage.

Furthermore, motivated by the behavioral similarity between Adam and SignGD when the learning rate is sufficiently small or $\beta_1, \beta_2$ approach zero (Balles and Hennig, 2018; Bernstein et al., 2018),

we present results for SignSGD. We subsequently extend these results to stochastic Adam, which provided in Appendix C. The update rules for SignSGD are given as follows:

$$(\text{SignSGD}) \quad \mathbf{w}_{j,r}^{(t+1)} = \mathbf{w}_{j,r}^{(t)} - \eta \cdot \mathrm{sgn}(g_{t,j,r}^{(t)}), \tag{5.1}$$

where $g_{t,j,r}^{(t)}$ in (5.1) denotes the stochastic gradient of (3.1). The detailed update rules of Adam with the SignSGD approximation are provided in Eqs. (B.3) and (B.4), while those of AdamW are given in Eqs. (B.5) and (B.6).

Next, following the framework of feature learning (Allen-Zhu and Li, 2020; Cao et al., 2022; Zou et al., 2023b; Han et al., 2025a), we primarily focus on two key quantities: 1) **Feature Learning** $\langle \mathbf{w}_{j,r}, j\mathbf{v} \rangle$: This term captures the alignment between the learned weight vector $\mathbf{w}_{j,r}$ and the true feature direction $j\mathbf{v}$, reflecting the model's ability to extract meaningful latent structures from the data. 2) **Noise Memorization** $\langle \mathbf{w}_{y_i,r}, \boldsymbol{\xi}_i \rangle$: This term measures the correlation between $\mathbf{w}_{y_i,r}$ and the noise patch $\boldsymbol{\xi}_i$ of individual samples, characterizing the extent to which the model overfits to stochastic perturbations or idiosyncrasies in the training set. This decomposition allows us to separately analyze the model's generalization behavior (driven by feature learning) and its memorization capacity (influenced by noise fitting).

We first clarify that, although the sketch appears straightforward, the underlying process is non-trivial. Numerous intricate and interesting details arise, which we elaborate on in the proof presented in Appendix C.

We begin by characterizing the dynamics of feature learning and noise memorization under large-batch training, to facilitate a comparative analysis with mini-batch regimes, as formalized in Lemma 5.1.

**Large-batch Adam.** We consider $\frac{n}{B} = \Theta(1)$, which is the large-batch setting.

**Lemma 5.1.** *Given the training dataset $\mathcal{S}$, if $\frac{n}{B} = \Theta(1)$, $\eta = 1/\mathrm{poly}(d)$ and $0 < \lambda = o(\sigma_0^{q-2}\sigma_p/n)$, then for any $t \leq T_0$ with $T_0 = \widetilde{O}(\frac{1}{\eta s \sigma_p})$ and any $i \in [n]$,*

$$\langle \mathbf{w}_{j,r}^{(t+1)}, j\mathbf{v} \rangle \leq \langle \mathbf{w}_{j,r}^{(t)}, j\mathbf{v} \rangle + \Theta(\eta), \quad \langle \mathbf{w}_{y_i,r}^{(t+1)}, \boldsymbol{\xi}_i \rangle = \langle \mathbf{w}_{y_i,r}^{(t)}, \boldsymbol{\xi}_i \rangle + \widetilde{\Theta}(\eta s \sigma_p).$$

We observe that Lemma 5.1 is identical to Lemma 5.2 in (Zou et al., 2023b), allowing us to directly extend their full-batch Adam results to the large-batch setting. The remainder of the proof follows identically, as the underlying theoretical machinery remains unchanged under this batch size scaling $\frac{n}{B} = \Theta(1)$. We observe that under large-batch setting, the optimization dynamics of Adam closely resemble those of the full-batch setting. This similarity arises because the algorithm traverses the entire dataset within few iterations, resulting in nearly identical momentum estimates and, consequently, comparable training dynamics between large-batch and full-batch regimes.

We next consider the mini-batch setting, which yields conclusions that differ fundamentally from those in the large-batch setting.

**Mini-batch Adam.** We consider $\frac{n}{B} \geq \Theta(\log \epsilon^{-1})$, which is the mini-batch setting.

**Lemma 5.2** (Stage I). *Given the training dataset $\mathcal{S}$, if $\frac{n}{B} \geq \Theta(\log \epsilon^{-1})$, $\eta = 1/\mathrm{poly}(d)$ and $0 < \lambda = o(\sigma_0^{q-2}\sigma_p/n)$, then for any $t \leq T_0$ with $T_0 = \widetilde{O}(\frac{1}{\eta s \sigma_p})$ and any $i \in [n]$,*

$$\langle \mathbf{w}_{j,r}^{\left((t+1)\cdot\frac{n}{B}\right)}, j \cdot \mathbf{v} \rangle = \langle \mathbf{w}_{j,r}^{(t\cdot\frac{n}{B})}, j \cdot \mathbf{v} \rangle + \Theta(\eta \cdot \frac{n}{B}), \quad \langle \mathbf{w}_{j,r}^{(t)}, \boldsymbol{\xi}_i \rangle \leq \widetilde{\Theta}(\eta s \sigma_p).$$

Compared to Lemma 5.1, Lemma 5.2 reveals fundamentally different optimization dynamics. Specifically, feature learning progressively increases throughout **Stage I**, whereas noise memorization remains suppressed near the initialization. The key distinction from the large-batch setting lies in the fact that, under the mini-batch regime, traversing the entire dataset requires many iterations. Since noise is sparse and uncorrelated across samples while features are dense and shared, feature learning can proceed effectively during the early training phase without being hindered by weight decay regularization. In contrast, noise memorization is strongly suppressed by weight decay due to its sparsity. As training progresses, the momentum estimates in Adam gradually forget the gradient contributions from noise, allowing weight decay to dominate. As a result, recently acquired noise

memorization is continually erased, keeping noise-related parameters close to their initialization throughout **Stage I**.

In the following lemma 5.3, we show that the patterns learned by the model during **Stage I** are preserved in **Stage II**, due to the nature of our non-convex optimization landscape.

**Lemma 5.3** (Stage II). *Suppose the same conditions hold as in Lemma 5.2. For $t > T_0$, $j \in \{\pm 1\}$, $r \in [m]$, $i \in [n]$, let $r^* = \operatorname{argmax}_{r \in [m]} \langle \mathbf{w}_{j,r}^{(t)}, j\mathbf{v} \rangle$, then $\langle \mathbf{w}_{j,r^*}^{(t)}, j\mathbf{v} \rangle = \widetilde{\Theta}(1)$ and $\langle \mathbf{w}_{y_i,r}^{(t)}, \boldsymbol{\xi}_i \rangle \leq \widetilde{\Theta}(\eta s \sigma_p)$.*

Given Lemma 5.2 and Lemma 5.3, we can characterize the convergence rate of Adam as follows.

**Lemma 5.4** (Convergence). *Suppose the same conditions hold as in Lemma 5.2 and 5.3, if the step size $\eta = O(d^{-\frac{1}{2}})$, then for any t,*

$$\mathbb{E}\left[L(\mathbf{W}^{(t+1)}) - L(\mathbf{W}^{(t)})\right] \leq -\eta \|\nabla L(\mathbf{W}^{(t)})\|_1 + \widetilde{\Theta}(\eta^2 d).$$

Combine Lemma 5.3 and Lemma 5.4, we observe that the model successfully learns the true features and eventually converges to a local minimum, retaining strong generalization performance.

# 6 Conclusion and Limitation

In this work, we theoretically and empirically analyze the impact of varying batch sizes and weight decay parameters on the generalization of Adam and AdamW when learning two-layer CNNs. Our results demonstrate that large-batch Adam and AdamW inherently overfit noise-dominated solutions even with weight decay, while their mini-batch counterparts achieve strong generalization through the synergy of implicit stochastic gradient regularization and explicit weight decay. Furthermore, we establish that Adam's adaptive gradient normalization imposes stricter constraints on weight decay parameters compared to AdamW, necessitating precise calibration for stable optimization.

While our theoretical framework provides insights into the interplay between batch size, weight decay, and generalization, several limitations highlight critical directions for future research. First, the current analysis is restricted to two-layer networks. Extending the results to deeper architectures and investigating how batch size influences the dynamics of hierarchical feature learning presents a promising direction. Second, our work focuses on image data, and an important direction is to extend the analysis to domains with fundamentally different data structures, such as NLP, where batch size and weight decay may impact model performance through different mechanisms. Finally, other critical hyperparameters, such as momentum, learning rate schedules, and gradient clipping, are not considered in our analysis, and some modern vision architectures succeed with large batches (Liu et al., 2022, 2023; Chen et al., 2024) despite our theoretical predictions, suggesting that additional factors like architectural design and normalization may play a significant role.

## Acknowledgments

We would like to thank the anonymous reviewers and area chairs for their helpful comments. Xuan Tang and Difan Zou acknowledge the support from NSFC 62306252, Hong Kong ECS award 27309624, Guangdong NSF 2024A1515012444, and the central fund from HKU IDS. Yuan Cao is partially supported by NSFC 12301657 and Hong Kong ECS award 27308624.

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

# A Preliminaries

In this section, we give the asymptotic equations for all the parameters we use, some useful lemmas, and the gradient and weight update equations.

## A.1 Asymptotic Equations

First, we give the asymptotic equations for all the parameters we use in the proof.

**Condition A.1.** *Suppose that the following conditions on data model 3.1 hold,*

1. *The dimension $d$ satisfies $d = \text{poly}(n)$.*

2. *The number of noise coordinates $s$ satisfies $s = \Theta\left(\frac{d^{1/2}}{n^2}\right)$.*

3. *The variance parameter $\sigma_p^2$ of the noise vector satisfies $\sigma_p^2 = \Theta\left(\frac{1}{s \cdot \text{polylog}(n)}\right)$.*

4. *The feature noise strength $\alpha$ satisfies $\alpha = \Theta\left(\sigma_p \cdot \text{polylog}(n)\right)$.*

**Condition A.2.** *Suppose that the following conditions on hyper-parameters hold,*

1. *The initialization variance of the model weights $\sigma_0^2$ satisfies $\sigma_0^2 = \Theta\left(\frac{1}{d^{1/2}}\right)$.*

2. *The width of the network $m$ satisfies $m = \text{polylog}(n)$.*

3. *The learning rate $\eta$ satisfies $\eta = \frac{1}{\text{poly}(n)}$.*

4. *The parameter $\epsilon$ in Stochastic Adam and Stochastic AdamW satisfies $\epsilon = \Theta(\lambda\eta)$.*

Based on the parameter configuration, we claim that the following equations hold, which will be frequently used in the subsequent proof.

$$\alpha = \omega\left((s\sigma_p)^{1-q}\sigma_0^{q-1}\right), \quad \alpha = o\left(\frac{s\sigma_p^2}{n}\right), \quad \sigma_0 = o\left(\frac{1}{s\sigma_p}\right), \quad \eta = o\left(\lambda\sigma_0^q\sigma_p^q\right). \tag{A.1}$$

The following Lemma A.3 describes the initialization.

**Lemma A.3.** *At the initialization, for $\forall j \in \{\pm 1\}, r \in [m], i \in [n]$, we have*

$$|\langle \mathbf{w}_{j,r}^{(0)}, \mathbf{v}\rangle| = \widetilde{\Theta}(\sigma_0), \quad |\langle \mathbf{w}_{j,r}^{(0)}, \boldsymbol{\xi}_i\rangle| = \widetilde{\Theta}(s^{1/2}\sigma_p\sigma_0 + \alpha\sigma_0) = \widetilde{\Theta}(s^{1/2}\sigma_p\sigma_0), \quad \mathbf{w}_{j,r}^{(0)}[k] = \widetilde{\Theta}(\sigma_0),$$

*Proof.* By Definition 3.2, we have

$$\mathbf{w}_{j,r}^{(0)} \sim \mathcal{N}(0, \sigma_0^2 \boldsymbol{I}_d).$$

For data $(\mathbf{x}_i, y_i) \sim \mathcal{D}$, by Definition 3.1, let $\mathcal{B}_i = \text{supp}(\boldsymbol{\xi}_i)\backslash\{1\}$, we have

$$\mathbf{v} = (1, 0, \ldots, 0)^\top,$$
$$\boldsymbol{\xi}_i[1] = -\alpha y_i,$$
$$\boldsymbol{\xi}_i[k] \sim \mathcal{N}(0, \sigma_p^2). \qquad \text{(for } k \in \mathcal{B}_i)$$

Therefore, condition on the training dataset $\mathcal{S}$, we have

$$\langle \mathbf{w}_{j,r}^{(0)}, \mathbf{v}\rangle \sim \mathcal{N}(0, \sigma_0^2).$$

By standard Gaussian tails, we get

$$|\langle \mathbf{w}_{j,r}^{(0)}, \mathbf{v}\rangle| = \widetilde{\Theta}(\sigma_0),$$
$$\mathbf{w}_{j,r}^{(0)}[k] = \widetilde{\Theta}(\sigma_0).$$

For $\langle \mathbf{w}_{j,r}^{(0)}, \boldsymbol{\xi}_i\rangle$, we have

$$\langle \mathbf{w}_{j,r}^{(0)}, \boldsymbol{\xi}_i\rangle = -\alpha\mathbf{w}_{j,r}^{(0)}[1] \cdot y_i + \sum_{k \in \mathcal{B}_i} \mathbf{w}_{j,r}^{(0)}[k] \cdot \boldsymbol{\xi}_i[k],$$

$$\text{where} \quad -\alpha \mathbf{w}_{j,r}^{(0)}[1] \cdot y_i \sim \mathcal{N}(0, \alpha^2 \sigma_0^2),$$

$$\sum_{k \in \mathcal{B}_i} \mathbf{w}_{j,r}^{(0)}[k] \cdot \boldsymbol{\xi}_i[k] \sim \mathcal{N}\left(0, \sigma_0^2 \cdot \left(\sum_{k \in \mathcal{B}_i} \boldsymbol{\xi}_i[k]^2\right)\right)$$

$$\sim \mathcal{N}(0, s\sigma_p^2 \sigma_0^2),$$

since $\boldsymbol{\xi}_i[k] \sim \mathcal{N}(0, \sigma_p^2)$. Therefore, we have

$$|\langle \mathbf{w}_{j,r}^{(0)}, \boldsymbol{\xi}_i \rangle| = \widetilde{\Theta}(s^{1/2} \sigma_p \sigma_0 + \alpha \sigma_0) = \widetilde{\Theta}(s^{1/2} \sigma_p \sigma_0).$$

$\square$

### A.2  Preliminary Lemmas

The following lemma studies non-overlap support property of noise patch in data model $\mathcal{D}$ in Definition 3.1.

**Lemma A.4** (Non-overlap support, Lemma C.1 in Zou et al. (2023b)). *Let $\{(\mathbf{x}_i, y_i)\}_{i=1,\ldots,n}$ be the training dataset sampled according to Definition 3.1. Moreover, let $\mathcal{B}_i = supp(\boldsymbol{\xi}_i)\backslash\{1\}$ be the support of $\mathbf{x}_i$ except the first coordinate. Then with probability at least $1 - n^{-2}$, $\mathcal{B}_i \cap \mathcal{B}_j = \emptyset$ for all $i, j \in [n]$.*

### A.3  Gradients and Updates

We first calculate the gradient of the individual loss function $L_i$ with respect to $\mathbf{w}_{j,r}^{(t)}$.

**Lemma A.5.** *Consider the CNN model defined in Eq. 3.2. Let $(\mathbf{x}_i, y_i)$ be a data generated from data model $\mathcal{D}$ in Definition 3.1. The gradient of $L_i(\mathbf{W}) = -\log \frac{e^{F_{y_i}(\mathbf{W}, \mathbf{x}_i)}}{\sum_{j \in \{-1,1\}} e^{F_j(\mathbf{W}, \mathbf{x}_i)}}$ with respect to $\mathbf{w}_{j,r}$ is:*

$$\nabla_{\mathbf{w}_{j,r}} L_i(\mathbf{W}) = -\nabla_{\mathbf{w}_{j,r}} F_j(\mathbf{W}, \mathbf{x}_i) \cdot \ell_{j,i}$$
$$= -\left(y_i \ell_{j,i} \sigma'(\langle \mathbf{w}_{j,r}, y_i \mathbf{v} \rangle) \cdot \mathbf{v} + \ell_{j,i} \sigma'(\langle \mathbf{w}_{j,r}, \boldsymbol{\xi}_i \rangle) \cdot \boldsymbol{\xi}_i\right),$$

*where $\ell_{j,i} := \mathbb{1}_{y_i = j} - \text{logit}_j(F, \mathbf{x}_i)$ and $\text{logit}_j(F, \mathbf{x}_i) = \frac{e^{F_j(\mathbf{W}, \mathbf{x}_i)}}{\sum_{k \in \{-1,1\}} e^{F_k(\mathbf{W}, \mathbf{x}_i)}}$.*

*Proof of Lemma A.5.* For $j = y_i$,

$$\nabla_{\mathbf{w}_{j,r}} L_i(\mathbf{W})$$
$$= -\frac{e^{F_j(\mathbf{W}, \mathbf{x}_i)} + e^{F_{-j}(\mathbf{W}, \mathbf{x}_i)}}{e^{F_j(\mathbf{W}, \mathbf{x}_i)}} \cdot \frac{e^{F_j(\mathbf{W}, \mathbf{x}_i) + F_{-j}(\mathbf{W}, \mathbf{x}_i)} \cdot \nabla_{\mathbf{w}_{j,r}} F_j(\mathbf{W}, \mathbf{x}_i)}{\left(e^{F_j(\mathbf{W}, \mathbf{x}_i)} + e^{F_{-j}(\mathbf{W}, \mathbf{x}_i)}\right)^2}$$
$$= -\nabla_{\mathbf{w}_{j,r}} F_j(\mathbf{W}, \mathbf{x}_i) \cdot \frac{e^{F_{-j}(\mathbf{W}, \mathbf{x}_i)}}{e^{F_j(\mathbf{W}, \mathbf{x}_i)} + e^{F_{-j}(\mathbf{W}, \mathbf{x}_i)}}$$
$$= -\nabla_{\mathbf{w}_{j,r}} F_j(\mathbf{W}, \mathbf{x}_i) \cdot \left(1 - \frac{e^{F_j(\mathbf{W}, \mathbf{x}_i)}}{e^{F_j(\mathbf{W}, \mathbf{x}_i)} + e^{F_{-j}(\mathbf{W}, \mathbf{x}_i)}}\right)$$
$$= -\nabla_{\mathbf{w}_{j,r}} F_j(\mathbf{W}, \mathbf{x}_i) \cdot \ell_{j,i}$$
$$= -\left(y_i \ell_{j,i} \sigma'(\langle \mathbf{w}_{j,r}, y_i \mathbf{v} \rangle) \cdot \mathbf{v} + \ell_{j,i} \sigma'(\langle \mathbf{w}_{j,r}, \boldsymbol{\xi}_i \rangle) \cdot \boldsymbol{\xi}_i\right).$$

For $j \neq y_i$,

$$\nabla_{\mathbf{w}_{j,r}} L_i(\mathbf{W})$$
$$= \frac{e^{F_j(\mathbf{W}, \mathbf{x}_i)} + e^{F_{-j}(\mathbf{W}, \mathbf{x}_i)}}{e^{F_{-j}(\mathbf{W}, \mathbf{x}_i)}} \cdot \frac{e^{F_j(\mathbf{W}, \mathbf{x}_i) + F_{-j}(\mathbf{W}, \mathbf{x}_i)} \cdot \nabla_{\mathbf{w}_{j,r}} F_j(\mathbf{W}, \mathbf{x}_i)}{\left(e^{F_j(\mathbf{W}, \mathbf{x}_i)} + e^{F_{-j}(\mathbf{W}, \mathbf{x}_i)}\right)^2}$$
$$= \nabla_{\mathbf{w}_{j,r}} F_j(\mathbf{W}, \mathbf{x}_i) \cdot \frac{e^{F_j(\mathbf{W}, \mathbf{x}_i)}}{e^{F_j(\mathbf{W}, \mathbf{x}_i)} + e^{F_{-j}(\mathbf{W}, \mathbf{x}_i)}}$$
$$= -\nabla_{\mathbf{w}_{j,r}} F_j(\mathbf{W}, \mathbf{x}_i) \cdot \left(0 - \frac{e^{F_j(\mathbf{W}, \mathbf{x}_i)}}{e^{F_j(\mathbf{W}, \mathbf{x}_i)} + e^{F_{-j}(\mathbf{W}, \mathbf{x}_i)}}\right)$$

$$= -\nabla_{\mathbf{w}_{j,r}} F_j(\mathbf{W}, \mathbf{x}_i) \cdot \ell_{j,i}$$
$$= -\left(y_i \ell_{j,i} \sigma'(\langle \mathbf{w}_{j,r}, y_i \mathbf{v} \rangle) \cdot \mathbf{v} + \ell_{j,i} \sigma'(\langle \mathbf{w}_{j,r}, \boldsymbol{\xi}_i \rangle) \cdot \boldsymbol{\xi}_i\right).$$

$\square$

Based on the definition of $\ell_{j,i}$, we have a useful lemma as follow:

**Lemma A.6.** *Given data* $(\mathbf{x}_i, y_i)$ *generated from data model 3.1, define* $\ell_{j,i} = \mathbb{1}_{y_i=j} - \mathrm{logit}_j(F, \mathbf{x}_i)$ *and* $\mathrm{logit}_j(F, \mathbf{x}_i) = \frac{e^{F_j(\mathbf{W}, \mathbf{x}_i)}}{\sum_{k \in \{-1,1\}} e^{F_k(\mathbf{W}, \mathbf{x}_i)}}$, *we have*

$$\mathrm{sgn}(y_i \ell_{j,i}) = \mathrm{sgn}(j).$$

*Proof of Lemma A.6.* For $j = y_i$,

$$\begin{aligned}
\mathrm{sgn}(y_i \ell_{j,i}) \cdot \mathrm{sgn}(j) &= \mathrm{sgn}(\ell_{j,i}) \\
&= \mathrm{sgn}(1 - \mathrm{logit}_j(F, \mathbf{x}_i)) && (\mathrm{logit}_j(F, \mathbf{x}_i) \in (0,1)) \\
&= 1.
\end{aligned}$$

For $j \neq y_i$,

$$\begin{aligned}
\mathrm{sgn}(y_i \ell_{j,i}) \cdot \mathrm{sgn}(j) &= -\mathrm{sgn}(\ell_{j,i}) \\
&= -\mathrm{sgn}(-\mathrm{logit}_j(F, \mathbf{x}_i)) \\
&= \mathrm{sgn}(\mathrm{logit}_j(F, \mathbf{x}_i)) && (\mathrm{logit}_j(F, \mathbf{x}_i) \in (0,1)) \\
&= 1.
\end{aligned}$$

$\square$

Now we calculate the gradient of loss (3.1) and loss (3.2), responding to stochastic Adam and stochastic AdamW respectively. Here we slightly abuse the notation. We use $g_{t,j,r}^{(t)}$ to represent the stochastic gradient with respect to $\mathbf{w}_{j,r}^{(t)}$ at the $t$-th iteration. The subscript $t$ of $g_{t,j,r}^{(t)}$ represents the batch at the $t$-th iteration and the superscript $t$ of $g_{t,j,r}^{(t)}$ represents the weight matrix $\mathbf{W}^{(t)}$ at the $t$-th iteration.

**Lemma A.7** (Gradient of Stochastic Adam). *Consider the CNN model in Definition 3.2. Let* $\{(\mathbf{x}_i, y_i)\}_{i=1}^n$ *be the training dataset generated from data model in Definition 3.1. Using stochastic Adam to train the neural network, at the $t$-th iteration with batch data index set* $\mathcal{I}_t$ *of size B, the stochastic gradient of the loss* (3.1) *with respect to* $\mathbf{w}_{j,r}^{(t)}$ *is as follows:*

$$g_{t,j,r}^{(t)} = -\frac{1}{B}\left[\sum_{i \in \mathcal{I}_t} y_i \ell_{j,i}^{(t)} \sigma'(\langle \mathbf{w}_{j,r}^{(t)}, y_i \mathbf{v} \rangle) \cdot \mathbf{v} + \sum_{i \in \mathcal{I}_t} \ell_{j,i}^{(t)} \sigma'(\langle \mathbf{w}_{j,r}^{(t)}, \boldsymbol{\xi}_i \rangle) \cdot \boldsymbol{\xi}_i\right] + \lambda \mathbf{w}_{j,r}^{(t)}.$$

*More specific, for the $k$-th coordinate, we have*

- $k = 1$:

$$g_{t,j,r}^{(t)}[1] = -\frac{1}{B}\left[\sum_{i \in \mathcal{I}_t} y_i \ell_{j,i}^{(t)} \sigma'(\langle \mathbf{w}_{j,r}^{(t)}, y_i \mathbf{v} \rangle) - \alpha \sum_{i \in \mathcal{I}_t} y_i \ell_{j,i}^{(t)} \sigma'(\langle \mathbf{w}_{j,r}^{(t)}, \boldsymbol{\xi}_i \rangle)\right] + \lambda \mathbf{w}_{j,r}^{(t)}[1].$$

- $k \in \mathcal{B}_i$, $i \in \mathcal{I}_t$:

$$g_{t,j,r}^{(t)}[k] = -\frac{1}{B} \ell_{j,i}^{(t)} \sigma'(\langle \mathbf{w}_{j,r}^{(t)}, \boldsymbol{\xi}_i \rangle) \cdot \boldsymbol{\xi}_i[k] + \lambda \mathbf{w}_{j,r}^{(t)}[k].$$

- $k \neq 1$ *and* $k \notin \cup_{i \in \mathcal{I}_t} \mathcal{B}_i$:

$$g_{t,j,r}^{(t)}[k] = \lambda \mathbf{w}_{j,r}^{(t)}[k].$$

*Proof of Lemma A.7.* The loss of stochastic Adam at the $t$-th iteration with batch data index set $\mathcal{I}_t$ is

$$L(\mathbf{W}^{(t)}) = \frac{1}{B} \sum_{i \in \mathcal{I}_t} L_i(\mathbf{W}^{(t)}) + \frac{\lambda}{2} \|\mathbf{W}^{(t)}\|_F^2.$$

By Lemma A.5, we have

$$
\begin{aligned}
g_{t,j,r}^{(t)} &= \nabla_{\mathbf{w}_{j,r}^{(t)}} L(\mathbf{W}^{(t)}) \\
&= \frac{1}{B} \sum_{i \in \mathcal{I}_t} \nabla_{\mathbf{w}_{j,r}^{(t)}} L_i(\mathbf{W}^{(t)}) + \lambda \mathbf{w}_{j,r}^{(t)} \\
&= -\frac{1}{B} \left[ \sum_{i \in \mathcal{I}_t} y_i \ell_{j,i}^{(t)} \sigma'(\langle \mathbf{w}_{j,r}^{(t)}, y_i \mathbf{v} \rangle) \cdot \mathbf{v} + \sum_{i \in \mathcal{I}_t} \ell_{j,i}^{(t)} \sigma'(\langle \mathbf{w}_{j,r}^{(t)}, \boldsymbol{\xi}_i \rangle) \cdot \boldsymbol{\xi}_i \right] + \lambda \mathbf{w}_{j,r}^{(t)}.
\end{aligned}
$$

For the $k$-th coordinate, if $k = 1$, we know $\mathbf{v} = [1, 0, \ldots, 0]^\top$ and $\boldsymbol{\xi}_i[1] = -\alpha y_i$. So we have

$$g_{t,j,r}^{(t)}[1] = -\frac{1}{B} \left[ \sum_{i \in \mathcal{I}_t} y_i \ell_{j,i}^{(t)} \sigma'(\langle \mathbf{w}_{j,r}^{(t)}, y_i \mathbf{v} \rangle) - \alpha \sum_{i \in \mathcal{I}_t} y_i \ell_{j,i}^{(t)} \sigma'(\langle \mathbf{w}_{j,r}^{(t)}, \boldsymbol{\xi}_i \rangle) \right] + \lambda \mathbf{w}_{j,r}^{(t)}[1].$$

If $k \in \mathcal{B}_i$, $i \in \mathcal{I}_t$, by Lemma A.4, we know $\mathcal{B}_i \cap \mathcal{B}_j = \emptyset$ for $i \neq j$. So we have

$$g_{t,j,r}^{(t)}[k] = -\frac{1}{B} \ell_{j,i}^{(t)} \sigma'(\langle \mathbf{w}_{j,r}^{(t)}, \boldsymbol{\xi}_i \rangle) \cdot \boldsymbol{\xi}_i[k] + \lambda \mathbf{w}_{j,r}^{(t)}[k].$$

If $k \neq 1$ and $k \notin \cup_{i \in \mathcal{I}_t} \mathcal{B}_i$, it is obvious that

$$g_{t,j,r}^{(t)}[k] = \lambda \mathbf{w}_{j,r}^{(t)}[k].$$

$\square$

**Lemma A.8** (Gradient of Stochastic AdamW). *Consider the CNN model defined in Eq. 3.2. Let $\{(\mathbf{x}_i, y_i) : i \in [n]\}$ be the training dataset generated from data model 3.1. Use stochastic AdamW training the neural network, at the $t$-th iteration with batch data index set $\mathcal{I}_t$ of size $B$, the stochastic gradient of the Loss defined in Eq. (3.2) with respect to $\mathbf{w}_{j,r}^{(t)}$ is as follows:*

$$g_{t,j,r}^{(t)} = -\frac{1}{B} \left[ \sum_{i \in \mathcal{I}_t} y_i \ell_{j,i}^{(t)} \sigma'(\langle \mathbf{w}_{j,r}^{(t)}, y_i \mathbf{v} \rangle) \cdot \mathbf{v} + \sum_{i \in \mathcal{I}_t} \ell_{j,i}^{(t)} \sigma'(\langle \mathbf{w}_{j,r}^{(t)}, \boldsymbol{\xi}_i \rangle) \cdot \boldsymbol{\xi}_i \right].$$

*More specific, for the $k$-th coordinate, we have*

- $k = 1$:

$$g_{t,j,r}^{(t)}[1] = -\frac{1}{B} \left[ \sum_{i \in \mathcal{I}_t} y_i \ell_{j,i}^{(t)} \sigma'(\langle \mathbf{w}_{j,r}^{(t)}, y_i \mathbf{v} \rangle) - \alpha \sum_{i \in \mathcal{I}_t} y_i \ell_{j,i}^{(t)} \sigma'(\langle \mathbf{w}_{j,r}^{(t)}, \boldsymbol{\xi}_i \rangle) \right].$$

- $k \in \mathcal{B}_i$, $i \in \mathcal{I}_t$:

$$g_{t,j,r}^{(t)}[k] = -\frac{1}{B} \ell_{j,i}^{(t)} \sigma'(\langle \mathbf{w}_{j,r}^{(t)}, \boldsymbol{\xi}_i \rangle) \cdot \boldsymbol{\xi}_i[k].$$

- $k \neq 1$ and $k \notin \cup_{i \in \mathcal{I}_t} \mathcal{B}_i$:

$$g_{t,j,r}^{(t)}[k] = 0.$$

*Proof of Lemma A.8.* The loss of stochastic AdamW at the $t$-th iteration with batch data index set $\mathcal{I}_t$ is

$$L(\mathbf{W}^{(t)}) = \frac{1}{B} \sum_{i \in \mathcal{I}_t} L_i(\mathbf{W}^{(t)}).$$

By Lemma A.5, we have

$$g_{t,j,r}^{(t)} = \nabla_{\mathbf{w}_{j,r}^{(t)}} L(\mathbf{W}^{(t)})$$

$$= \frac{1}{B} \sum_{i \in \mathcal{I}_t} \nabla_{\mathbf{w}_{j,r}^{(t)}} L_i(\mathbf{W}^{(t)})$$

$$= -\frac{1}{B} \left[ \sum_{i \in \mathcal{I}_t} y_i \ell_{j,i}^{(t)} \sigma'(\langle \mathbf{w}_{j,r}^{(t)}, y_i \mathbf{v} \rangle) \cdot \mathbf{v} + \sum_{i \in \mathcal{I}_t} \ell_{j,i}^{(t)} \sigma'(\langle \mathbf{w}_{j,r}^{(t)}, \boldsymbol{\xi}_i \rangle) \cdot \boldsymbol{\xi}_i \right].$$

For the $k$-th coordinate, if $k = 1$, we know $\mathbf{v} = [1, 0, \ldots, 0]^\top$ and $\boldsymbol{\xi}_i[1] = -\alpha y_i$. So we have

$$g_{t,j,r}^{(t)}[1] = -\frac{1}{B} \left[ \sum_{i \in \mathcal{I}_t} y_i \ell_{j,i}^{(t)} \sigma'(\langle \mathbf{w}_{j,r}^{(t)}, y_i \mathbf{v} \rangle) - \alpha \sum_{i \in \mathcal{I}_t} y_i \ell_{j,i}^{(t)} \sigma'(\langle \mathbf{w}_{j,r}^{(t)}, \boldsymbol{\xi}_i \rangle) \right].$$

If $k \in \mathcal{B}_i$, $i \in \mathcal{I}_t$, by Lemma A.4, we know $\mathcal{B}_i \cap \mathcal{B}_j = \emptyset$ for $i \neq j$. So we have

$$g_{t,j,r}^{(t)}[k] = -\frac{1}{B} \ell_{j,i}^{(t)} \sigma'(\langle \mathbf{w}_{j,r}^{(t)}, \boldsymbol{\xi}_i \rangle) \cdot \boldsymbol{\xi}_i[k].$$

If $k \neq 1$ and $k \notin \cup_{i \in \mathcal{I}_t} \mathcal{B}_i$, it is obvious that

$$g_{t,j,r}^{(t)}[k] = 0.$$

$\square$

## B  Proof Sketch

In this section, we mainly outline the proof sketch for the main results in Section 4. Following the two-stage analysis framework of Cao et al. (2022); Zou et al. (2023b), we decompose the proof into two distinct stages:

**Stage I: Pattern Learning.** During the initial phase of training, the effect of regularization is negligible. The model operates in an underfitting regime, where it rapidly learns dominant patterns in the training data, leading to improved empirical performance on test error.

**Stage II: Regularization.** As training progresses, the model's classification accuracy on the training set approaches convergence, resulting in diminished gradient magnitudes. At this stage, regularization dominates the optimization dynamics, steering the model converge to a local minima. Due to the nonconvex nature of the loss landscape, the model retains the patterns acquired during the pattern learning stage.

Furthermore, motivated by the behavioral similarity between Adam and SignGD when the learning rate is sufficiently small or $\beta_1, \beta_2$ approach zero (Balles and Hennig, 2018; Bernstein et al., 2018), we present results for SignSGD and SignSGDW (SignSGD with decoupled weight decay). We subsequently extend these results to stochastic Adam and AdamW, which provided in Appendix C. The update rules for SignSGD are given as follows:

$$\text{(SignSGD)} \quad \mathbf{w}_{j,r}^{(t+1)} = \mathbf{w}_{j,r}^{(t)} - \eta \cdot \text{sgn}(g_{t,j,r}^{(t)}), \tag{B.1}$$

where $g_{t,j,r}^{(t)}$ in (B.1) is stochastic gradient of (3.1). The updata rules for SignSGDW are given as follows:

$$\text{(SignSGDW)} \quad \mathbf{w}_{j,r}^{(t+1)} = (1 - \lambda\eta)\mathbf{w}_{j,r}^{(t)} - \eta \cdot \text{sgn}(g_{t,j,r}^{(t)}), \tag{B.2}$$

where $g_{t,j,r}^{(t)}$ in (B.2) is stochastic gradient of (3.2), and $\lambda$ is the weight decay parameter.

Next, following the framework of feature learning (Allen-Zhu and Li, 2020; Cao et al., 2022; Zou et al., 2023b; Han et al., 2025a), we primarily focus on two key quantities: 1) **Feature Learning** $\langle \mathbf{w}_{j,r}, j\mathbf{v} \rangle$: This term captures the alignment between the learned weight vector $\mathbf{w}_{j,r}$ and the true feature direction $j\mathbf{v}$, reflecting the model's ability to extract meaningful latent structures from the data. 2) **Noise Memorization** $\langle \mathbf{w}_{y_i,r}, \boldsymbol{\xi}_i \rangle$: This term measures the correlation between $\mathbf{w}_{y_i,r}$ and the noise patch $\boldsymbol{\xi}_i$ of individual samples, characterizing the extent to which the model overfits to stochastic perturbations or idiosyncrasies in the training dataset. This decomposition allows us to separately analyze the model's generalization behavior (driven by feature learning) and its memorization capacity (influenced by noise fitting).

## B.1 Proof Sketch for Stochastic Adam

We present the dynamics of feature learning $\langle \mathbf{w}_{j,r}^{(t)}, j\mathbf{v} \rangle$ and noise memorization $\langle \mathbf{w}_{y_i,r}^{(t)}, \boldsymbol{\xi}_i \rangle$ for SignSGD as follows. The details of calculation are provided in Appendix A.

$$
\begin{aligned}
&\left\langle \mathbf{w}_{j,r}^{(t+1)}, j\mathbf{v} \right\rangle \\
&= \left\langle \mathbf{w}_{j,r}^{(t)}, j\mathbf{v} \right\rangle - \eta \cdot \left\langle \mathrm{sgn}\left(g_{t,j,r}^{(t)}\right), j\mathbf{v} \right\rangle \\
&= \left\langle \mathbf{w}_{j,r}^{(t)}, j\mathbf{v} \right\rangle + j\eta \cdot \mathrm{sgn}\left( \sum_{i \in \mathcal{I}_t} y_i \ell_{j,i}^{(t)} \left[ \sigma'(\langle \mathbf{w}_{j,r}^{(t)}, y_i\mathbf{v}\rangle) - \alpha\sigma'(\langle \mathbf{w}_{j,r}^{(t)}, \boldsymbol{\xi}_i\rangle) \right] - B\lambda \mathbf{w}_{j,r}^{(t)}[1] \right), \quad \text{(B.3)}
\end{aligned}
$$

$$
\begin{aligned}
&\left\langle \mathbf{w}_{y_i,r}^{(t+1)}, \boldsymbol{\xi}_i \right\rangle \\
&= \left\langle \mathbf{w}_{y_i,r}^{(t)}, \boldsymbol{\xi}_i \right\rangle - \eta \cdot \left\langle \mathrm{sgn}\left(g_{t,y_i,r}^{(t)}\right), \boldsymbol{\xi}_i \right\rangle \\
&= \left\langle \mathbf{w}_{y_i,r}^{(t)}, \boldsymbol{\xi}_i \right\rangle + \eta \cdot \sum_{k \in \mathcal{B}_i} \left\langle \mathrm{sgn}\left( \ell_{y_i,i}^{(t)} \sigma'(\langle \mathbf{w}_{y_i,r}^{(t)}, \boldsymbol{\xi}_i\rangle)\boldsymbol{\xi}_i[k] - n\lambda \mathbf{w}_{y_i,r}^{(t)}[k] \right), \boldsymbol{\xi}_i[k] \right\rangle \\
&\quad - \alpha y_i \eta \cdot \mathrm{sgn}\left( \sum_{j \in \mathcal{I}_t} y_j \ell_{y_i,j}^{(t)} \left[ \sigma'(\langle \mathbf{w}_{y_i,r}^{(t)}, y_j\mathbf{v}\rangle) - \alpha\sigma'(\langle \mathbf{w}_{y_i,r}^{(t)}, \boldsymbol{\xi}_j\rangle) \right] - B\lambda \mathbf{w}_{y_i,r}^{(t)}[1] \right), \quad \text{(B.4)}
\end{aligned}
$$

where $\ell_{j,i}^{(t)} := \mathbb{1}_{y_i=j} - \mathrm{logit}_j(F, \mathbf{x}_i)$ and $\mathrm{logit}_j(F, \mathbf{x}_i) = \frac{e^{F_j(\mathbf{W}, \mathbf{x}_i)}}{\sum_{k \in \{-1,1\}} e^{F_k(\mathbf{W}, \mathbf{x}_i)}}$.

### B.1.1 Proof Sketch for Theorem 4.1

In this section, we present the proof sketch for Theorem 4.1. We consider $\frac{n}{B} = \Theta(1)$, which is the large-batch setting.

**Lemma B.1.** *Given the training dataset $\mathcal{S}$, if $\frac{n}{B} = \Theta(1)$, $\eta = 1/\mathrm{poly}(d)$ and $0 < \lambda = o(\sigma_0^{q-2}\sigma_p/n)$, then for any $t \leq T_0$ with $T_0 = \widetilde{O}(\frac{1}{\eta s \sigma_p})$ and any $i \in [n]$,*

$$
\langle \mathbf{w}_{j,r}^{(t+1)}, j\mathbf{v} \rangle \leq \langle \mathbf{w}_{j,r}^{(t)}, j\mathbf{v} \rangle + \Theta(\eta), \quad \langle \mathbf{w}_{y_i,r}^{(t+1)}, \boldsymbol{\xi}_i \rangle = \langle \mathbf{w}_{y_i,r}^{(t)}, \boldsymbol{\xi}_i \rangle + \widetilde{\Theta}(\eta s \sigma_p).
$$

We note that the above lemma is equivalent to Lemma 5.2 in (Zou et al., 2023b), which enables us to directly extend their results for full-batch Adam to the large-batch regime. The remainder of the proof proceeds in the same way, as the core theoretical framework remains invariant under the batch size scaling $\frac{n}{B} = \Theta(1)$. Recall that the condition $s\sigma_p = \omega(1)$ implies that noise memorization outpaces feature learning and we have $\ell_{j,r} = \Theta(1)$ throughout **Stage I** since the outputs are small. As a result, after a certain number of iterations, the direction of feature learning is reversed, as indicated by the update rule in Equation B.3. Specifically, the noise-driven term $\alpha\sigma'(\langle \mathbf{w}_{j,r}^{(t)}, \boldsymbol{\xi}_i\rangle)$ becomes dominant, satisfying $\alpha\sigma'(\langle \mathbf{w}_{j,r}^{(t)}, \boldsymbol{\xi}_i\rangle) \gg \sigma'(\langle \mathbf{w}_{j,r}^{(t)}, y_i\mathbf{v}\rangle) + n\lambda|\mathbf{w}_{j,r}^{(t)}[1]|$. By the end of **Stage I**, the model's feature learning direction has been inverted, while noise memorization reaches a quasi-stationary state. In the subsequent regularization phase, weight decay drives the model toward convergence. However, it lacks the capacity to eliminate the memorized noise. Consequently, the model fits the feature noise $-\alpha y\mathbf{v}$ and converges to a local minimum that preserves the patterns acquired in **Stage I**, ultimately leading to poor generalization performance.

We observe that under large-batch setting, the optimization dynamics of Adam closely resemble those of the full-batch setting. This similarity arises because the algorithm traverses the entire dataset within few iterations, resulting in nearly identical momentum estimates and, consequently, comparable training dynamics between large-batch and full-batch regimes.

We next consider the mini-batch setting, which yields conclusions that differ fundamentally from those in the large-batch setting.

### B.1.2 Proof Sketch for Theorem 4.2

In this section, we present the proof sketch for Theorem 4.2. We consider $\frac{n}{B} \geq \Theta(\log \epsilon^{-1})$, which is the mini-batch setting.

**Lemma B.2** (Stage I). *Given the training dataset $\mathcal{S}$, if $\frac{n}{B} \geq \Theta(\log \epsilon^{-1})$, $\eta = 1/\text{poly}(d)$ and $0 < \lambda = o(\sigma_0^{q-2}\sigma_p/n)$, then for any $t \leq T_0$ with $T_0 = \widetilde{O}(\frac{B}{\eta n})$ and any $i \in [n]$,*

$$\langle \mathbf{w}_{j,r}^{((t+1)\cdot\frac{n}{B})}, j \cdot \mathbf{v}\rangle = \langle \mathbf{w}_{j,r}^{(t\cdot\frac{n}{B})}, j \cdot \mathbf{v}\rangle + \Theta(\eta \cdot \frac{n}{B}), \quad \langle \mathbf{w}_{j,r}^{(t)}, \boldsymbol{\xi}_i\rangle \leq \widetilde{\Theta}(\eta s \sigma_p).$$

Compared to Lemma B.1, Lemma B.2 reveals fundamentally different optimization dynamics in the mini-batch regime. In **Stage I**, feature learning advances steadily—since $\ell_{j,r} = \Theta(1)$—while noise memorization remains at its initialization scale. This divergence arises because mini-batch sampling requires many more iterations to traverse the dataset: dense, shared features receive consistent gradient updates and resist weight decay, whereas sparse, uncorrelated noise is continuously attenuated. As features strengthen, network outputs grow, the loss gradients diminish, and weight decay takes over, marking the transition into **Stage II**. We now show that the structures acquired in **Stage I** persist throughout this regularization phase.

**Lemma B.3** (Stage II). *Suppose the same conditions hold as in Lemma B.2. For $t > T_0$, $j \in \{\pm1\}$, $r \in [m]$, $i \in [n]$, let $r^* = \text{argmax}_{r\in[m]}\langle \mathbf{w}_{j,r}^{(t)}, j\mathbf{v}\rangle$, then $\langle \mathbf{w}_{j,r^*}^{(t)}, j\mathbf{v}\rangle = \widetilde{\Theta}(1)$ and $\langle \mathbf{w}_{y_i,r}^{(t)}, \boldsymbol{\xi}_i\rangle \leq \widetilde{\Theta}(\eta s \sigma_p)$.*

This lemma follows because, once feature learning has increased accuracy and reduced gradients, weight decay takes effect but cannot reverse the established feature alignment; finally the model converges to a local minimum that preserves the patterns learned in **Stage I**.

**Lemma B.4** (Convergence). *Suppose the same conditions hold as in Lemma B.2 and B.3, if the step size $\eta = O(d^{-\frac{1}{2}})$, then for any t,*

$$\mathbb{E}\left[L(\mathbf{W}^{(t+1)}) - L(\mathbf{W}^{(t)})\right] \leq -\eta\|\nabla L(\mathbf{W}^{(t)})\|_1 + \widetilde{\Theta}(\eta^2 d).$$

Combining Lemma B.3 and B.4, we observe that the model successfully learns the true features and eventually converges to a local minimum with infinitesimal learning rate $\eta$ and $T = \text{poly}(n)/\eta$, retaining strong generalization performance.

### B.1.3  Proof Sketch for Corollary 4.3

We next show that if the weight decay parameter satisfies $\lambda = \omega(\sigma_0^{q-2})$, then the learning dynamics of Adam are effectively suppressed. This implies that the *effective* weight decay for Adam is of the order $\sigma_0^{q-2}$, which is significantly smaller than that required for AdamW, as will be discussed later.

Corollary 4.3 formalizes this observation by showing that if the Adam weight decay parameter satisfies $\lambda = \omega(\sigma_0^{q-2})$, the training process becomes stagnant and remains near the initialization. This corollary follows directly from the proof sketches of both large-batch and mini-batch Adam. By Lemma A.3, we know that at initialization:

$$|\langle \mathbf{w}_{j,r}^{(0)}, \mathbf{v}\rangle| = \widetilde{\Theta}(\sigma_0), \quad |\langle \mathbf{w}_{j,r}^{(0)}, \boldsymbol{\xi}_i\rangle| = \widetilde{\Theta}(s^{1/2}\sigma_p\sigma_0 + \alpha\sigma_0) = \widetilde{\Theta}(s^{1/2}\sigma_p\sigma_0), \quad \mathbf{w}_{j,r}^{(0)}[k] = \widetilde{\Theta}(\sigma_0).$$

Then, from the update rules given in Equations (B.3) and (B.4), we observe that the updates are dominated by the weight decay term, i.e.,

$$\lambda|\mathbf{w}_{j,r}^{(0)}[1]| \gg \sigma'(\langle \mathbf{w}_{j,r}^{(0)}, y_i\mathbf{v}\rangle) + \alpha\sigma'(\langle \mathbf{w}_{j,r}^{(0)}, \boldsymbol{\xi}_i\rangle),$$

and

$$\lambda|\mathbf{w}_{y_i,r}^{(0)}[k]| \gg \sigma'(\langle \mathbf{w}_{y_i,r}^{(0)}, \boldsymbol{\xi}_i\rangle) \cdot \boldsymbol{\xi}_i[k],$$

due to the condition $\lambda = \omega(\sigma_0^{q-2})$. As a result, the learning dynamics are overwhelmed by the regularization term, preventing meaningful updates. Consequently, the model parameters remain close to their initialization throughout training. We formalize this in Lemma B.5.

**Lemma B.5.** *Suppose the same conditions hold as in Lemma B.1 and B.4, if $\lambda = \omega(\sigma_0^{q-2})$, then*

$$\left|\langle \mathbf{w}_{j,r}^{(t)}, j \cdot \mathbf{v}\rangle\right| \leq \widetilde{\Theta}(\sigma_0),$$

$$\left|\langle \mathbf{w}_{j,r}^{(t)}, \boldsymbol{\xi}_i\rangle\right| \leq \widetilde{\Theta}(s^{1/2}\sigma_p\sigma_0).$$

## B.2 Proof Sketch for Stochastic AdamW

Motivated by the similarity between Adam and SignSGD, and to better illustrate the core idea, we present the dynamics of feature learning $\langle \mathbf{w}_{j,r}^{(t)}, j\mathbf{v} \rangle$ and noise memorization $\langle \mathbf{w}_{y_i,r}^{(t)}, \boldsymbol{\xi}_i \rangle$ under SignSGDW. The detailed derivation of update formula is deferred to Appendix A. However, we emphasize that there are key differences between AdamW and SignSGDW.

In SignSGDW, due to the presence of the sign operator, the weight decay affects $\langle \mathbf{w}_{j,r}^{(t)}, j\mathbf{v} \rangle$ only after it grows beyond a certain threshold, and similarly for $\langle \mathbf{w}_{y_i,r}^{(t)}, \boldsymbol{\xi}_i \rangle$. In contrast, for AdamW, weight decay becomes effective once $\langle \mathbf{w}_{j,r}^{(t)}, j\mathbf{v} \rangle$ or $\langle \mathbf{w}_{y_i,r}^{(t)}, \boldsymbol{\xi}_i \rangle$ reaches a level where the gradient magnitudes become sufficiently small. At this point, the update is normalized by the stability constant $\epsilon$ and dominated by the weight decay term, which causes both $\langle \mathbf{w}_{j,r}^{(t)}, j\mathbf{v} \rangle$ and $\langle \mathbf{w}_{y_i,r}^{(t)}, \boldsymbol{\xi}_i \rangle$ to cease increasing. Besides, as the lemmas in this section are simplified instances of those presented in Section C.2, we omit them for brevity. For more details, refer to Section C.2.

$$
\left\langle \mathbf{w}_{j,r}^{(t+1)}, j\mathbf{v} \right\rangle
$$
$$
= (1 - \lambda\eta) \left\langle \mathbf{w}_{j,r}^{(t)}, j\mathbf{v} \right\rangle - \eta \cdot \left\langle \operatorname{sgn}\left(g_{t,j,r}^{(t)}\right), j\mathbf{v} \right\rangle
$$
$$
= (1 - \lambda\eta) \left\langle \mathbf{w}_{j,r}^{(t)}, j\mathbf{v} \right\rangle + j\eta \cdot \operatorname{sgn}\left( \sum_{i \in \mathcal{I}_t} y_i \ell_{j,i}^{(t)} \left[ \sigma'(\langle \mathbf{w}_{j,r}^{(t)}, y_i \mathbf{v} \rangle) - \alpha\sigma'(\langle \mathbf{w}_{j,r}^{(t)}, \boldsymbol{\xi}_i \rangle) \right] \right), \quad \text{(B.5)}
$$
$$
\left\langle \mathbf{w}_{y_i,r}^{(t+1)}, \boldsymbol{\xi}_i \right\rangle
$$
$$
= (1 - \lambda\eta) \left\langle \mathbf{w}_{y_i,r}^{(t)}, \boldsymbol{\xi}_i \right\rangle - \eta \cdot \left\langle \operatorname{sgn}\left(g_{t,y_i,r}^{(t)}\right), \boldsymbol{\xi}_i \right\rangle
$$
$$
= (1 - \lambda\eta) \left\langle \mathbf{w}_{y_i,r}^{(t)}, \boldsymbol{\xi}_i \right\rangle + \eta \cdot \sum_{k \in \mathcal{B}_i} \left\langle \operatorname{sgn}\left( \ell_{y_i,i}^{(t)} \sigma'(\langle \mathbf{w}_{y_i,r}^{(t)}, \boldsymbol{\xi}_i \rangle) \boldsymbol{\xi}_i[k] \right), \boldsymbol{\xi}_i[k] \right\rangle
$$
$$
- \alpha y_i \eta \cdot \operatorname{sgn}\left( \sum_{i=1}^{n} y_i \ell_{y_i,i}^{(t)} \left[ \sigma'(\langle \mathbf{w}_{y_i,r}^{(t)}, y_i \mathbf{v} \rangle) - \alpha\sigma'(\langle \mathbf{w}_{y_i,r}^{(t)}, \boldsymbol{\xi}_i \rangle) \right] \right), \quad \text{(B.6)}
$$

where $\ell_{j,i}^{(t)} := \mathbb{1}_{y_i=j} - \operatorname{logit}_j(F, \mathbf{x}_i)$ and $\operatorname{logit}_j(F, \mathbf{x}_i) = \frac{e^{F_j(\mathbf{W}, \mathbf{x}_i)}}{\sum_{k \in \{-1,1\}} e^{F_k(\mathbf{W}, \mathbf{x}_i)}}$.

Moreover, we should note that the duration of **Stage I** under SignSGDW differs markedly from that under SignSGD, owing to the decoupled weight decay mechanism. During **Stage I**, model parameters grow unchecked by gradient-based regularization, allowing features to accumulate strength until the decoupled weight decay term begins to exert significant influence. Once this threshold is reached, training transitions into **Stage II**, in which weight decay counteracts further parameter growth and stabilizes the weight norms.

### B.2.1 Proof Sketch for Theorem 4.4

In this section, we present the proof sketch for Theorem 4.4. We consider $\frac{n}{B} = \Theta(1)$ or $\frac{n}{B} = o(s\sigma_p)$, which is the large-batch setting.

The following Lemma B.6 characterizes the duration of **Stage I** in the large-batch SignSGDW setting and provides upper bounds on feature learning and noise memorization.

**Lemma B.6** (Stage I, pattern learning). *Given the training dataset $\mathcal{S}$, if $\frac{n}{B} = \Theta(1)$ or $\frac{n}{B} = o(s\sigma_p)$, $\eta = 1/\operatorname{poly}(d)$, $\lambda = \widetilde{\Omega}(\frac{B^2}{n} \wedge 1)$ and $\lambda = \widetilde{O}(1)$, then for any $t \leq T_0$ with $T_0 = \widetilde{O}(\frac{B}{\lambda\eta n})$,*

$$
\langle \mathbf{w}_{j,r}^{((t+1)\cdot\frac{n}{B})}, j \cdot \mathbf{v} \rangle \leq \langle \mathbf{w}_{j,r}^{(t\cdot\frac{n}{B})}, j \cdot \mathbf{v} \rangle + \Theta(\eta \cdot \frac{n}{B})
$$
$$
\langle \mathbf{w}_{y_i,r}^{((t+1)\cdot\frac{n}{B})}, \boldsymbol{\xi}_i \rangle = \langle \mathbf{w}_{y_i,r}^{(t\cdot\frac{n}{B})}, \boldsymbol{\xi}_i \rangle + \widetilde{\Theta}(\eta s\sigma_p).
$$

Since in the large-batch regime $\frac{n}{B} = o(s\sigma_p)$, Lemma B.6 implies that noise memorization accumulates faster than feature learning. At the beginning of Stage I, feature gradients dominate because

$\sigma'(\langle \mathbf{w}_{y_i,r}^{(t)}, y_i \mathbf{v} \rangle) \gg \alpha \sigma'(\langle \mathbf{w}_{y_i,r}^{(t)}, \boldsymbol{\xi}_i \rangle)$, given $\alpha = o(1)$ and negligible weight decay influence. After a certain number of epochs, the noise term grows until $\alpha \sigma'(\langle \mathbf{w}_{y_i,r}^{(t)}, \boldsymbol{\xi}_i \rangle) \gg \sigma'(\langle \mathbf{w}_{y_i,r}^{(t)}, y_i \mathbf{v} \rangle)$, at which point feature learning reverses and eventually flips direction. Lemma B.7 below provides a precise description of this transition.

**Lemma B.7** (Stage I, fitting feature noise). *Suppose the same conditions hold as in Lemma B.6, if $\alpha \geq \widetilde{\Theta}\left( (\frac{B}{n} s \sigma_p)^{1-q} \right)$, then for any $t \in [T_r, T_0]$ with $T_r = \widetilde{O}\left( \frac{\sigma_0}{\eta s \sigma_p \alpha^{1/(q-1)}} \right) \leq T_0$,*

$$\langle \mathbf{w}_{j,r}^{((t+1)\cdot \frac{n}{B})}, j \cdot \mathbf{v} \rangle = \langle \mathbf{w}_{j,r}^{(t \cdot \frac{n}{B})}, j \cdot \mathbf{v} \rangle - \Theta(\eta \cdot \frac{n}{B}).$$

*and at epoch $T_0$, we have (a) $\mathbf{w}_{j,r}^{(T_0 \cdot \frac{n}{B})}[1] = -\text{sgn}(j) \cdot \widetilde{\Omega}(1/\lambda)$; (b) $\mathbf{w}_{j,r}^{(T_0 \cdot \frac{n}{B})}[k] = \text{sgn}(\boldsymbol{\xi}_i[k]) \cdot \widetilde{\Omega}(\frac{B}{n\lambda})$ or $\pm \widetilde{O}(\eta)$ for $k \in \mathcal{B}_i$ with $y_i = j$; (c) $\mathbf{w}_{j,r}^{(T_0 \cdot \frac{n}{B})}[k] = \pm \widetilde{O}(\eta)$ otherwise.*

Lemma B.7 implies that, by the end of **Stage I**, the model has fitted the training noise. The following Lemma B.8 shows that these pattern persist throughout **Stage II**, ultimately leading to poor generalization.

**Lemma B.8** (Stage II, preserve the noise). *Suppose the same conditions hold as in Lemma B.6 and B.7, for $t > T_0$, $j \in \{\pm 1\}$, $r \in [m]$, $i \in [n]$, let $r^* = \arg\max_{r \in [m]} \langle \mathbf{w}_{y_i,r}^{(t)}, \boldsymbol{\xi}_i \rangle$, then $\langle \mathbf{w}_{j,r}^{(t)}, j \cdot \mathbf{v} \rangle = -\widetilde{\Theta}(1/\lambda)$ and $\langle \mathbf{w}_{y_i,r^*}^{(t)}, \boldsymbol{\xi}_i \rangle = \widetilde{\Theta}(\frac{Bs\sigma_p}{n\lambda})$*

The following Lemma B.9 prove the convergence under certain conditions.

**Lemma B.9** (Convergence). *Suppose the same conditions hold as in Lemma B.6, B.7 and B.8, if the step size satisfies $\eta = O(d^{-1/2})$, then for any $t$,*

$$\mathbb{E}\left[ L(\mathbf{W}^{(t+1)}) - L(\mathbf{W}^{(t)}) \right] \leq -\eta \|\nabla L(\mathbf{W}^{(t)})\|_1 + \widetilde{\Theta}(\eta^{3/2} d).$$

Combining Lemmas B.8 and B.9, we observe that, with an infinitesimal learning rate $\eta$ and $T = \text{poly}(n)/\eta$, the model ultimately fits the feature noise and converges to a local minimum, resulting in poor generalization performance.

### B.2.2  Proof Sketch for Theorem 4.5

In this section, we present the proof sketch for Theorem 4.5. We consider $\frac{n}{B} \geq \Theta(n^{1/2} \vee \log \epsilon^{-1})$ and $\frac{n}{B} = \omega(s\sigma_p)$, which is the mini-batch setting.

The following Lemma B.10 characterizes the duration of **Stage I** in the mini-batch SignSGDW setting and provides upper bounds on feature learning and noise memorization.

**Lemma B.10** (Stage I). *Given the training dataset $\mathcal{S}$, if $\frac{n}{B} \geq \Theta(n^{1/2} \vee \log \epsilon^{-1})$ and $\frac{n}{B} = \omega(s\sigma_p)$, $\eta = 1/\text{poly}(d)$, $\lambda = \widetilde{\Omega}(\frac{B^2}{n} \wedge 1)$ and $\lambda = \widetilde{O}(1)$, then for any $t \leq T_0$ with $T_0 = \widetilde{O}(\frac{B}{\lambda \eta n})$,*

$$\langle \mathbf{w}_{j,r}^{((t+1)\cdot \frac{n}{B})}, j\mathbf{v} \rangle = \langle \mathbf{w}_{j,r}^{(t \cdot \frac{n}{B})}, j\mathbf{v} \rangle + \Theta(\eta \cdot \frac{n}{B}), \quad \langle \mathbf{w}_{y_i,r}^{((t+1)\cdot \frac{n}{B})}, \boldsymbol{\xi}_i \rangle \leq \langle \mathbf{w}_{y_i,r}^{(t \cdot \frac{n}{B})}, \boldsymbol{\xi}_i \rangle + \widetilde{\Theta}(\eta s \sigma_p).$$

In the mini-batch regime, $\frac{n}{B} = \omega(s\sigma_p)$, so feature learning outpaces noise memorization—unlike in the large-batch case. Consequently, noise cannot reverse the feature learning; instead, features are learned continuously until decoupled weight decay intervenes. Noise memorization also grows until this point, but because noise is both sparse and independent, it accrues only during a few iterations per epoch and is concurrently suppressed by weight decay. Hence, both feature learning and noise memorization reach their peak at the end of **Stage I**, after which weight decay governs **Stage II**. The next Lemma B.11 formalizes this behavior.

**Lemma B.11** (Stage II). *Suppose the same conditions hold as in Lemma B.10, for $t > T_0$, $j \in \{\pm 1\}$, $r \in [m]$, $i \in [n]$, let $r^* = \arg\max_{r \in [m]} \langle \mathbf{w}_{j,r}^{(t)}, j\mathbf{v} \rangle$, then $\langle \mathbf{w}_{j,r^*}^{(t)}, j\mathbf{v} \rangle = \widetilde{\Theta}(1/\lambda)$ and $\langle \mathbf{w}_{y_i,r}^{(t)}, \boldsymbol{\xi}_i \rangle \leq \widetilde{\Theta}(\frac{Bs\sigma_p}{n\lambda})$.*

The following lemma establishes convergence under the specified conditions.

**Lemma B.12** (Convergence). *Suppose the same conditions hold as in Lemma B.10 and B.11, if the step size satisfies $\eta = O(d^{-1/2})$, then for any $t$,*

$$\mathbb{E}\left[ L(\mathbf{W}^{(t+1)}) - L(\mathbf{W}^{(t)}) \right] \leq -\eta \|\nabla L(\mathbf{W}^{(t)})\|_1 + \widetilde{\Theta}(\eta^2 d).$$

### B.2.3 Proof Sketch for Corollary 4.6

By Conditions A.1 and A.2, along with Definition 3.2, we know that $d = \mathrm{poly}(n)$, and hence

$$\sigma_0^{q-2} = \Theta\left(\frac{1}{d^{(q-2)/4}}\right), \quad \text{with } q \geq 3.$$

This directly implies that the effective weight decay parameter for Adam satisfies

$$\lambda_{\mathrm{Adam}} \sim \sigma_0^{q-2} \ll \min\left\{\frac{B^2}{n}, 1\right\} \sim \lambda_{\mathrm{AdamW}}.$$

This completes the proof.

## C Proofs

First we give a general upper bound of the moving average in stochastic Adam and stochastic AdamW.

**Lemma C.1.** *Let $\mathbf{m}_{j,r}^{(t)}$ be the first momentum estimate, $\mathbf{v}_{j,r}^{(t)}$ be the second momentum estimate at the $t$-th iterate in the update rule of stochastic Adam or stochastic AdamW. Then for all $j \in \{\pm 1\}$, $r \in [m]$ and $k \in [d]$, if $\beta_2 > \beta_1^2$, $\beta_1, \beta_2 \in [0, 1)$, we have*

$$\left|\frac{\mathbf{m}_{j,r}^{(t)}[k]}{\sqrt{\mathbf{v}_{j,r}^{(t)}[k]} + \epsilon}\right| \leq \Theta(1).$$

*Proof of Lemma C.1.* Let us expand the moment estimates

$$\mathbf{m}_{j,r}^{(t)}[k] = \beta_1 \mathbf{m}_{j,r}^{(t-1)}[k] + (1 - \beta_1) \cdot g_{t,j,r}^{(t)}[k]$$

$$= \sum_{\tau=0}^{t-1} \beta_1^\tau (1 - \beta_1) \cdot g_{t-\tau,j,r}^{(t-\tau)}[k],$$

$$\mathbf{v}_{j,r}^{(t)}[k] = \beta_2 \mathbf{v}_{j,r}^{(t-1)}[k] + (1 - \beta_2) \cdot g_{t,j,r}^{(t)}[k]^2$$

$$= \sum_{\tau=0}^{t-1} \beta_2^\tau (1 - \beta_2) \cdot g_{t-\tau,j,r}^{(t-\tau)}[k]^2.$$

Let $\{z_t^2 : z_t^2 = \frac{[\beta_1^t(1-\beta_1)]^2}{\beta_2^t(1-\beta_2)}\}$ be a convergent series since $\beta_2 > \beta_1^2$. Then we have

$$\mathbf{m}_{j,r}^{(t)}[k]^2$$

$$= \left(\sum_{\tau=0}^{t-1} \beta_1^\tau (1 - \beta_1) \cdot g_{t-\tau,j,r}^{(t-\tau)}[k]\right)^2$$

$$= \left(\sum_{\tau=0}^{t-1} \frac{\beta_1^\tau (1 - \beta_1)}{z_\tau} \cdot g_{t-\tau,j,r}^{(t-\tau)}[k] \cdot z_\tau\right)^2$$

$$\leq \left(\sum_{\tau=0}^{t-1} \frac{[\beta_1^\tau (1 - \beta_1)]^2}{z_\tau^2} \cdot g_{t-\tau,j,r}^{(t-\tau)}[k]^2\right) \cdot \left(\sum_{\tau=0}^{t-1} z_\tau^2\right)$$

$$= \sum_{\tau=0}^{t-1} \beta_2^\tau (1 - \beta_2) \cdot g_{t-\tau,j,r}^{(t-\tau)}[k]^2 \cdot \Theta(1)$$

$$= \mathbf{v}_{j,r}^{(t)}[k] \cdot \Theta(1),$$

where the first inequality we use Cauchy-Schwartz inequality and the third equality we use the fact that $z_t^2 = \frac{[\beta_1^t(1-\beta_1)]^2}{\beta_2^t(1-\beta_2)}$ is a convergent series. So we have

$$\left|\frac{\mathbf{m}_{j,r}^{(t)}[k]}{\sqrt{\mathbf{v}_{j,r}^{(t)}[k]} + \epsilon}\right| \leq \frac{\left|\mathbf{m}_{j,r}^{(t)}[k]\right|}{\sqrt{\mathbf{v}_{j,r}^{(t)}[k]}} \leq \Theta(1).$$

$\square$

### C.1 Proof of Stochastic Adam

First, we try to approximate the update of stochastic Adam to sign update since the similar performance between Adam and SignGD (Bernstein et al., 2018; Balles and Hennig, 2018; Zou et al., 2023b; Xie and Li, 2024; Li et al., 2025).

**Lemma C.2.** *Consider the update of stochastic Adam in* (3.5). *Let* $\mathbf{W}^{(t)}$ *be the weight at the $t$-th iteration. Suppose that* $\langle \mathbf{w}_{j,r}^{(t)}, y_i \mathbf{v} \rangle, \langle \mathbf{w}_{j,r}^{(t)}, \boldsymbol{\xi}_i \rangle = \widetilde{\Theta}(1)$ *for all* $j \in \{\pm 1\}$, $r \in [m]$, $i \in [n]$ *and* $\beta_1^2 < \beta_2$. *We have the approximate update rule for each coordinate weight as follows:*

- *For $k = 1$, we have either* $|g_{t,j,r}^{(t)}[1]| \leq \widetilde{\Theta}(\eta)$ *or*

$$\frac{\mathbf{m}_{j,r}^{(t)}[k]}{\sqrt{\mathbf{v}_{j,r}^{(t)}[k]} + \epsilon} = \mathrm{sgn}\big(g_{t,j,r}^{(t)}[k]\big) \cdot \Theta(1).$$

- *For every $k \in \mathcal{B}_i$, $i \in \mathcal{I}_{t-\tau}$, $\tau \in \mathcal{T}_k := \{\tau_0 + i \cdot \frac{n}{B} : i \in \{0\} \cup [\frac{\bar{\tau}}{n/B} - 1], \tau_0 < \frac{n}{B}\}$, where $\tau_0$ represents the number of iterations away from the current iteration $t$, coordinate $k$ is affected by $\boldsymbol{\xi}_i$ sampled at the iteration $t - \tau_0$ since the moving average, and we define $\bar{\tau} = \Theta(\log(\lambda\eta)^{-1})$*

  - *If $\frac{n}{B} = \Theta(1)$, for any $\tau_0 < \frac{n}{B}$, we have either*

  $$\left| g_{t-\tau_0,j,r}^{(t)}[k] \right| \leq \widetilde{\Theta}\left( B^{-1}\eta s\sigma_p|\ell_{j,i}^{(t)}| + |\lambda\mathbf{w}_{j,r}^{(t)}[k]| \right)$$

  *or*

  $$\frac{\mathbf{m}_{j,r}^{(t)}[k]}{\sqrt{\mathbf{v}_{j,r}^{(t)}[k]} + \epsilon} = \mathrm{sgn}\left( g_{t-\tau_0,j,r}^{(t)}[k] \right) \cdot \Theta(1).$$

  - *If $\frac{n}{B} \geq \Theta(\log(\lambda\eta)^{-1}) = \widetilde{\Theta}(1)$, for $\tau_0 = \Theta(1)$ such that $\beta_1^{\tau_0} = \Theta(1)$, we have either*

  $$\left| g_{t-\tau_0,j,r}^{(t)}[k] \right| \leq \widetilde{\Theta}\left( B^{-1}\eta s\sigma_p|\ell_{j,i}^{(t)}| + |\lambda\mathbf{w}_{j,r}^{(t)}[k]| \right)$$

  *or*

  $$\frac{\mathbf{m}_{j,r}^{(t)}[k]}{\sqrt{\mathbf{v}_{j,r}^{(t)}[k]} + \epsilon} = \mathrm{sgn}\left( g_{t-\tau_0,j,r}^{(t)}[k] \right) \cdot \Theta(1).$$

  *For $\tau_0 \geq \Theta(\log(\lambda\eta)^{-1})$ such that $\beta_2^{\frac{\tau_0}{2}} \leq \Theta(\lambda\eta)$, we have either*

  $$\left| \mathbf{w}_{j,r}^{(t)}[k] \right| \leq \widetilde{\Theta}(\eta)$$

  *or*

  $$\frac{\mathbf{m}_{j,r}^{(t)}[k]}{\sqrt{\mathbf{v}_{j,r}^{(t)}[k]} + \epsilon} = \mathrm{sgn}\left( \mathbf{w}_{j,r}^{(t)}[k] \right) \cdot \Theta(1).$$

- *For the remaining coordinates $k \neq 1$ and $k \notin \mathcal{B}_i$, $i \in \mathcal{I}_{t-\tau_0}$, $\tau_0 \in \{0\} \cup [\bar{\tau}]$, where $\tau_0$ represents the number of iterations away from the current iteration $t$, coordinate $k$ is affected by $\boldsymbol{\xi}_i$ sampled at the iteration $t - \tau_0$ since the moving average, and we define $\bar{\tau} = \log(\lambda\eta)^{-1}$. Then we have either* $|g_{t,j,r}^{(t)}[k]| \leq \widetilde{\Theta}(\lambda\eta)$ *or*

$$\frac{\mathbf{m}_{j,r}^{(t)}[k]}{\sqrt{\mathbf{v}_{j,r}^{(t)}[k]} + \epsilon} = \mathrm{sgn}\left( g_{t,j,r}^{(t)}[k] \right) \cdot \Theta(1).$$

*Proof of Lemma C.2.* Let first focus on the first momentum estimate,

$$\mathbf{m}_{j,r}^{(t)}[k] = \beta_1 \mathbf{m}_{j,r}^{(t-1)}[k] + (1-\beta_1) \cdot g_{t,j,r}^{(t)}[k]$$

$$= \sum_{\tau=0}^{t-1} \beta_1^\tau (1-\beta_1) \cdot g_{t-\tau,j,r}^{(t-\tau)}[k]$$

$$= \sum_{\tau=0}^{\bar{\tau}} \beta_1^\tau (1-\beta_1) \cdot g_{t-\tau,j,r}^{(t-\tau)}[k] + \sum_{\tau=\bar{\tau}+1}^{t-1} \beta_1^\tau (1-\beta_1) \cdot g_{t-\tau,j,r}^{(t-\tau)}[k]$$

$$= \sum_{\tau=0}^{\bar{\tau}} \beta_1^\tau (1-\beta_1) \cdot g_{t-\tau,j,r}^{(t-\tau)}[k] \pm \widetilde{O}(\lambda\eta),$$

where the last equality we select $\bar{\tau} = \Theta(\log(\lambda\eta)^{-1})$ such that $\sum_{\tau=\bar{\tau}+1}^{t} \beta_1^\tau (1-\beta_1) = O(\lambda\eta)$ and $|g_{t-\tau,j,r}^{(t-\tau)}[k]| = \widetilde{O}(1)$ for all $k \in [d]$ by Lemma A.7, since the facts that $\langle \mathbf{w}_{j,r}^{(t)}, y_i \mathbf{v} \rangle, \langle \mathbf{w}_{j,r}^{(t)}, \boldsymbol{\xi}_i \rangle = \widetilde{O}(1)$.

Similarly, for the second momentum estimate,

$$\mathbf{v}_{j,r}^{(t)}[k] = \sum_{\tau=0}^{t-1} \beta_2^\tau (1-\beta_2) \cdot g_{t-\tau,j,r}^{(t-\tau)}[k]^2$$

$$= \sum_{\tau=0}^{\bar{\tau}} \beta_2^\tau (1-\beta_2) \cdot g_{t-\tau,j,r}^{(t-\tau)}[k]^2 + \sum_{\tau=\bar{\tau}+1}^{t-1} \beta_2^\tau (1-\beta_2) \cdot g_{t-\tau,j,r}^{(t-\tau)}[k]^2$$

$$= \sum_{\tau=0}^{\bar{\tau}} \beta_2^\tau (1-\beta_2) \cdot g_{t-\tau,j,r}^{(t-\tau)}[k]^2 \pm \widetilde{O}(\lambda\eta).$$

Here we use the same $\bar{\tau}$ because we can always reselect $\bar{\tau}$ of smaller one to larger one, and the absolute value of the tail will not increase. Then we have

$$\frac{\mathbf{m}_{j,r}^{(t)}[k]}{\sqrt{\mathbf{v}_{j,r}^{(t)}[k]} + \epsilon} = \frac{\sum_{\tau=0}^{\bar{\tau}} \beta_1^\tau (1-\beta_1) \cdot g_{t-\tau,j,r}^{(t-\tau)}[k] \pm \widetilde{O}(\lambda\eta)}{\sqrt{\sum_{\tau=0}^{\bar{\tau}} \beta_2^\tau (1-\beta_2) \cdot g_{t-\tau,j,r}^{(t-\tau)}[k]^2 \pm \widetilde{O}(\lambda\eta)}}, \tag{C.1}$$

since $\epsilon = \Theta(\lambda\eta)$. Now we want to use sign update to approximate (C.1). First, we should note that once the signs of $g_{t-\tau,j,r}^{(t-\tau)}[k]$ for $\tau \in [0,\bar{\tau}]$ aligned, (C.1) can be approximated as $\text{sgn}(g_{t,j,r}^{(t)}[k]) \cdot \widetilde{\Theta}(1)$, since

$$\sqrt{\sum_{\tau=0}^{\bar{\tau}} \beta_2^\tau (1-\beta_2) \cdot g_{t-\tau,j,r}^{(t-\tau)}[k]^2} \le \sum_{\tau=0}^{\bar{\tau}} \beta_2^{\frac{\tau}{2}} (1-\beta_2)^{\frac{1}{2}} \cdot \left| g_{t-\tau,j,r}^{(t-\tau)}[k] \right|,$$

$$\sqrt{\sum_{\tau=0}^{\bar{\tau}} \beta_2^\tau (1-\beta_2) \cdot g_{t-\tau,j,r}^{(t-\tau)}[k]^2} \ge \frac{1}{\bar{\tau}+1} \sum_{\tau=0}^{\bar{\tau}} \beta_2^{\frac{\tau}{2}} (1-\beta_2)^{\frac{1}{2}} \cdot \left| g_{t-\tau,j,r}^{(t-\tau)}[k] \right|.$$

Recall the gradient of stochastic Adam given in Lemma A.7, we want to approximate $g_{t-\tau,j,r}^{(t-\tau)}[k]$ to $g_{t-\tau,j,r}^{(t)}[k]$, such that we can use the current weight to approximate sign update. By Lemma C.1, the upper bound of each coordinate in one step is $\Theta(\eta)$. Then for $\tau \in [t-\bar{\tau}, t]$, we have

$$\left| \langle \mathbf{w}_{j,r}^{(t)}, y_i \mathbf{v} \rangle - \langle \mathbf{w}_{j,r}^{(\tau)}, y_i \mathbf{v} \rangle \right|$$

$$\le \sum_{k=\tau}^{t-1} \left| \langle \mathbf{w}_{j,r}^{(k+1)}, y_i \mathbf{v} \rangle - \langle \mathbf{w}_{j,r}^{(k)}, y_i \mathbf{v} \rangle \right|$$

$$\le \Theta(\eta\bar{\tau}). \tag{C.2}$$

Similarly, we have

$$\left| \langle \mathbf{w}_{j,r}^{(t)}, \boldsymbol{\xi}_i \rangle - \langle \mathbf{w}_{j,r}^{(\tau)}, \boldsymbol{\xi}_i \rangle \right| \le \Theta(\eta\bar{\tau}s\sigma_p), \tag{C.3}$$

$$\left| \mathbf{w}_{j,r}^{(t)}[k] - \mathbf{w}_{j,r}^{(\tau)}[k] \right| \leq \Theta(\eta\bar{\tau}). \tag{C.4}$$

Then recall the predict function

$$F_j(\mathbf{W}^{(t)}, \mathbf{x}_i) = \sum_{r=1}^{m} \left[ \sigma\left(\langle \mathbf{w}_{j,r}^{(t)}, y_i\mathbf{v}\rangle\right) + \sigma\left(\langle \mathbf{w}_{j,r}^{(t)}, \boldsymbol{\xi}_i\rangle\right) \right].$$

We have

$$\left| F_j(\mathbf{W}^{(t)}, \mathbf{x}_i) - F_j(\mathbf{W}^{(\tau)}, \mathbf{x}_i) \right|$$

$$\leq \sum_{r=1}^{m} \left| \sigma\left(\langle \mathbf{w}_{j,r}^{(t)}, y_i\mathbf{v}\rangle\right) - \sigma\left(\langle \mathbf{w}_{j,r}^{(\tau)}, y_i\mathbf{v}\rangle\right) \right| + \sum_{r=1}^{m} \left| \sigma\left(\langle \mathbf{w}_{j,r}^{(t)}, \boldsymbol{\xi}_i\rangle\right) - \sigma\left(\langle \mathbf{w}_{j,r}^{(\tau)}, \boldsymbol{\xi}_i\rangle\right) \right|$$

$$\leq \sum_{r=1}^{m} \widetilde{\Theta}(1) \cdot \left| \langle \mathbf{w}_{j,r}^{(t)}, y_i\mathbf{v}\rangle - \langle \mathbf{w}_{j,r}^{(\tau)}, y_i\mathbf{v}\rangle \right| + \sum_{r=1}^{m} \widetilde{\Theta}(1) \cdot \left| \langle \mathbf{w}_{j,r}^{(t)}, \boldsymbol{\xi}_i\rangle - \langle \mathbf{w}_{j,r}^{(\tau)}, \boldsymbol{\xi}_i\rangle \right|$$

$$\leq \widetilde{\Theta}(m\eta\bar{\tau}s\sigma_p) + \widetilde{\Theta}(m\eta\bar{\tau})$$

$$= \widetilde{\Theta}(\eta\bar{\tau}s\sigma_p), \tag{C.5}$$

where the second inequality we use the convexity of $\sigma(\cdot)$ and the facts that $|\langle \mathbf{w}_{j,r}^{(t)}, y_i\mathbf{v}\rangle| = \widetilde{\Theta}(1)$ and $|\langle \mathbf{w}_{j,r}^{(t)}, \boldsymbol{\xi}_i\rangle| = \widetilde{\Theta}(1)$. The last inequality we use $m = \widetilde{\Theta}(1)$ and $s\sigma_p = \omega(1)$.

Then we can approximate $\ell_{j,r}^{(\tau)}$ to $\ell_{j,r}^{(t)}$ in the gradient A.7.

$$\ell_{j,i}^{(\tau)} = \frac{e^{F_{-j}(\mathbf{W}^{(\tau)}, \mathbf{x}_i)}}{\sum_{k \in \{-1,1\}} e^{F_k(\mathbf{W}^{(\tau)}, \mathbf{x}_i)}}$$

$$= \frac{e^{F_{-j}(\mathbf{W}^{(t)}, \mathbf{x}_i) \pm \widetilde{\Theta}(\eta\bar{\tau}s\sigma_p)}}{e^{F_j(\mathbf{W}^{(t)}, \mathbf{x}_i) \pm \widetilde{\Theta}(\eta\bar{\tau}s\sigma_p)} + e^{F_{-j}(\mathbf{W}^{(t)}, \mathbf{x}_i) \pm \widetilde{\Theta}(\eta\bar{\tau}s\sigma_p)}}$$

$$= \operatorname{sgn}(\ell_{j,i}^{(t)}) \cdot \Theta(|\ell_{j,i}^{(t)}|), \qquad (y_i = j)$$

$$\ell_{j,i}^{(\tau)} = \frac{-e^{F_j(\mathbf{W}^{(\tau)}, \mathbf{x}_i)}}{\sum_{k \in \{-1,1\}} e^{F_k(\mathbf{W}^{(\tau)}, \mathbf{x}_i)}}$$

$$= \frac{-e^{F_j(\mathbf{W}^{(t)}, \mathbf{x}_i) \pm \widetilde{\Theta}(\eta\bar{\tau}s\sigma_p)}}{e^{F_j(\mathbf{W}^{(t)}, \mathbf{x}_i) \pm \widetilde{\Theta}(\eta\bar{\tau}s\sigma_p)} + e^{F_{-j}(\mathbf{W}^{(t)}, \mathbf{x}_i) \pm \widetilde{\Theta}(\eta\bar{\tau}s\sigma_p)}}$$

$$= \operatorname{sgn}(\ell_{j,i}^{(t)}) \cdot \Theta(|\ell_{j,i}^{(t)}|), \qquad (y_i \neq j)$$

where we use the fact that $\widetilde{\Theta}(\eta\bar{\tau}s\sigma_p) = o(1)$ and (C.5). So we have

$$\ell_{j,i}^{(\tau)} = \operatorname{sgn}(\ell_{j,i}^{(t)}) \cdot \Theta(|\ell_{j,i}^{(t)}|),$$

for all $\tau \in [t-\bar{\tau}, t]$. Further, by (C.2), (C.3) and the facts that $|\langle \mathbf{w}_{j,r}^{(t)}, y_i\mathbf{v}\rangle| = \widetilde{O}(1)$ and $|\langle \mathbf{w}_{j,r}^{(t)}, \boldsymbol{\xi}_i\rangle| = \widetilde{O}(1)$, recall $\sigma(x) = \max(0, x)^q$, we have

$$\ell_{j,i}^{(\tau)} \sigma'(\langle \mathbf{w}_{j,r}^{(\tau)}, y_i\mathbf{v}\rangle)$$

$$\leq \ell_{j,i}^{(\tau)} \sigma'(\langle \mathbf{w}_{j,r}^{(t)}, y_i\mathbf{v}\rangle) + |\ell_{j,i}^{(\tau)}| \cdot \widetilde{\Theta}(\eta\bar{\tau})$$

$$= \operatorname{sgn}(\ell_{j,i}^{(t)}) \cdot \Theta(|\ell_{j,i}^{(t)}|) \cdot \sigma'(\langle \mathbf{w}_{j,r}^{(t)}, y_i\mathbf{v}\rangle) + \Theta(|\ell_{j,i}^{(t)}|) \cdot \widetilde{\Theta}(\eta\bar{\tau}),$$

$$\ell_{j,i}^{(\tau)} \sigma'(\langle \mathbf{w}_{j,r}^{(\tau)}, y_i\mathbf{v}\rangle)$$

$$\geq \ell_{j,i}^{(\tau)} \sigma'(\langle \mathbf{w}_{j,r}^{(t)}, y_i\mathbf{v}\rangle) - |\ell_{j,i}^{(\tau)}| \cdot \widetilde{\Theta}(\eta\bar{\tau})$$

$$= \operatorname{sgn}(\ell_{j,i}^{(t)}) \cdot \Theta(|\ell_{j,i}^{(t)}|) \cdot \sigma'(\langle \mathbf{w}_{j,r}^{(t)}, y_i\mathbf{v}\rangle) - \Theta(|\ell_{j,i}^{(t)}|) \cdot \widetilde{\Theta}(\eta\bar{\tau}).$$

So we conclude that

$$\ell_{j,i}^{(\tau)}\sigma'(\langle \mathbf{w}_{j,r}^{(\tau)}, y_i\mathbf{v}\rangle) = \mathrm{sgn}(\ell_{j,i}^{(t)}) \cdot \Theta(|\ell_{j,i}^{(t)}|) \cdot \sigma'(\langle \mathbf{w}_{j,r}^{(t)}, y_i\mathbf{v}\rangle) \pm \Theta(|\ell_{j,i}^{(t)}|) \cdot \widetilde{\Theta}(\eta\bar{\tau}). \tag{C.6}$$

Similarly, we have

$$\ell_{j,i}^{(\tau)}\sigma'(\langle \mathbf{w}_{j,r}^{(\tau)}, \boldsymbol{\xi}_i\rangle) = \mathrm{sgn}(\ell_{j,i}^{(t)}) \cdot \Theta(|\ell_{j,i}^{(t)}|) \cdot \sigma'(\langle \mathbf{w}_{j,r}^{(t)}, \boldsymbol{\xi}_i\rangle) \pm \Theta(|\ell_{j,i}^{(t)}|) \cdot \widetilde{\Theta}(\eta\bar{\tau}s\sigma_p). \tag{C.7}$$

Now, we have all the tools we need to approximate $g_{t-\tau,j,r}^{(t-\tau)}[k]$ to $g_{t-\tau,j,r}^{(t)}[k]$. Recall Lemma A.7, substitute (C.4), (C.6) and (C.7) into $g_{t-\tau,j,r}^{(t-\tau)}[k]$, we have

- For $k = 1$,

$$g_{t-\tau,j,r}^{(t-\tau)}[1] = \Theta\left(g_{t-\tau,j,r}^{(t)}[1]\right) \pm \Theta\left(\frac{1}{B}\sum_{i\in\mathcal{I}_{t-\tau}}|\ell_{j,i}^{(t)}|\right) \cdot \widetilde{O}(\eta\bar{\tau}) \pm \Theta(\lambda\eta\bar{\tau}). \tag{C.8}$$

- For all $k \in \mathcal{B}_i, i \in \mathcal{I}_{t-\tau}$,

$$g_{t-\tau,j,r}^{(t-\tau)}[k] = \Theta\left(g_{t-\tau,j,r}^{(t)}[k]\right) \pm \Theta\left(\frac{|\ell_{j,i}^{(t)}|}{B}\right) \cdot \widetilde{O}(\eta\bar{\tau}s\sigma_p) \pm \Theta(\lambda\eta\bar{\tau}). \tag{C.9}$$

- For $k \neq 1$ and $k \notin \mathcal{B}_i, i \in \mathcal{I}_{t-\tau}$,

$$g_{t-\tau,j,r}^{(t-\tau)}[k] = \Theta\left(g_{t-\tau,j,r}^{(t)}[k]\right) \pm \Theta(\lambda\eta\bar{\tau}). \tag{C.10}$$

Plugging (C.8), (C.9) and (C.10) into (C.1), with facts that $\bar{\tau} = \widetilde{\Theta}(1)$, $\lambda = o(1)$, $|\langle \mathbf{w}_{j,r}^{(t)}, \boldsymbol{\xi}_i\rangle| = \widetilde{\Theta}(1)$, $|\langle \mathbf{w}_{j,r}^{(t)}, y_i\mathbf{v}\rangle| = \widetilde{\Theta}(1)$, $|\ell_{j,i}^{(t)}| = \Theta(1)$, $\epsilon = \Theta(\lambda\eta)$ and Lemma A.6, we have

- For $k = 1$,

$$
\begin{aligned}
&\frac{\mathbf{m}_{j,r}^{(t)}[1]}{\sqrt{\mathbf{v}_{j,r}^{(t)}[1] + \epsilon}} \\
&= \frac{\sum_{\tau=0}^{\bar{\tau}} \beta_1^\tau(1-\beta_1) \cdot g_{t-\tau,j,r}^{(t-\tau)}[1] \pm \widetilde{O}(\lambda\eta)}{\sqrt{\sum_{\tau=0}^{\bar{\tau}} \beta_2^\tau(1-\beta_2) \cdot g_{t-\tau,j,r}^{(t-\tau)}[1]^2 \pm \widetilde{O}(\lambda\eta)}} \\
&= \frac{\sum_{\tau=0}^{\bar{\tau}} \beta_1^\tau(1-\beta_1) \cdot \left(\Theta\left(g_{t-\tau,j,r}^{(t)}[1]\right) \pm \Theta\left(\frac{1}{B}\sum_{i\in\mathcal{I}_{t-\tau}}|\ell_{j,i}^{(t)}|\right) \cdot \widetilde{O}(\eta\bar{\tau}) \pm \Theta(\lambda\eta\bar{\tau})\right) \pm \widetilde{O}(\lambda\eta)}{\sqrt{\sum_{\tau=0}^{\bar{\tau}} \beta_2^\tau(1-\beta_2) \cdot \left(\Theta\left(g_{t-\tau,j,r}^{(t)}[1]\right) \pm \Theta\left(\frac{1}{B}\sum_{i\in\mathcal{I}_{t-\tau}}|\ell_{j,i}^{(t)}|\right) \cdot \widetilde{O}(\eta\bar{\tau}) \pm \Theta(\lambda\eta\bar{\tau})\right)^2 \pm \widetilde{O}(\lambda\eta)}} \\
&= \frac{\sum_{\tau=0}^{\bar{\tau}} \beta_1^\tau(1-\beta_1) \cdot \left(\Theta\left(g_{t-\tau,j,r}^{(t)}[1]\right) \pm \widetilde{\Theta}(\eta\bar{\tau})\right) \pm \widetilde{O}(\lambda\eta)}{\sqrt{\sum_{\tau=0}^{\bar{\tau}} \beta_2^\tau(1-\beta_2) \cdot \left(\Theta\left(g_{t-\tau,j,r}^{(t)}[1]\right) \pm \widetilde{\Theta}(\eta\bar{\tau})\right)^2 \pm \widetilde{O}(\lambda\eta)}} \\
&= \frac{\Theta\left(g_{t,j,r}^{(t)}[1]\right) \pm \widetilde{\Theta}(\eta\bar{\tau}) \pm \widetilde{O}(\lambda\eta)}{\sqrt{\left(\Theta\left(g_{t,j,r}^{(t)}[1]\right) \pm \widetilde{\Theta}(\eta\bar{\tau})\right)^2 \pm \widetilde{O}(\lambda\eta)}} \\
&= \frac{\Theta\left(g_{t,j,r}^{(t)}[1]\right) \pm \widetilde{\Theta}(\eta)}{\Theta\left(|g_{t,j,r}^{(t)}[1]|\right) \pm \widetilde{\Theta}(\eta)}.
\end{aligned}
$$

- For $k \in \mathcal{B}_i$, $i \in \mathcal{I}_{t-\tau_0}$, $\tau_0 \in \{0\} \cup [\bar{\tau}]$, where $\tau_0$ represents the number of iterations away from the current iteration $t$, coordinate $k$ is affected by $\boldsymbol{\xi}_i$ sampled at the iteration $t - \tau_0$ since the moving average. We note that if the number of iteration in one epoch $\frac{n}{B}$ is less than $\bar{\tau}$, the moving average will use some sample $\mathbf{x}$ multiply times. We denote $\mathcal{T}_k := \{\tau_0 + i \cdot \frac{n}{B} : i \in \{0\} \cup [\frac{\bar{\tau}}{n/B} - 1], \tau_0 \le \frac{n}{B}\}$ as the timestamp set involved using noise $\boldsymbol{\xi}_i$ (i.e., $i \in \mathcal{I}_{t-\tau}$ for any $\tau \in \mathcal{T}_k$), and $k \in \mathcal{B}_i$, $\tau_0 \le \frac{n}{B}$.

  If $\frac{n}{B} > \bar{\tau}$, in this case we have $\mathcal{T}_k := \{\tau_0\}$ for any $k \in \mathcal{B}_i$, and $\boldsymbol{\xi}_i$ was used in iteration $t - \tau_0$. Then we have

  $$\frac{\mathbf{m}_{j,r}^{(t)}[k]}{\sqrt{\mathbf{v}_{j,r}^{(t)}[k] + \epsilon}}$$

  $$= \frac{\sum_{\tau=0}^{\bar{\tau}} \beta_1^\tau (1-\beta_1) \cdot g_{t-\tau,j,r}^{(t-\tau)}[k] \pm \widetilde{O}(\lambda\eta)}{\sqrt{\sum_{\tau=0}^{\bar{\tau}} \beta_2^\tau (1-\beta_2) \cdot g_{t-\tau,j,r}^{(t-\tau)}[k]^2 \pm \widetilde{O}(\lambda\eta)}}$$

  $$= \frac{\sum_{\tau \in \mathcal{T}_k} \beta_1^\tau (1-\beta_1) \cdot g_{t-\tau,j,r}^{(t-\tau)}[k] + \sum_{\tau \in \{0\} \cup [\bar{\tau}] \setminus \mathcal{T}_k} \beta_1^\tau (1-\beta_1) \cdot g_{t-\tau,j,r}^{(t-\tau)}[k] \pm \widetilde{O}(\lambda\eta)}{\sqrt{\sum_{\tau=0}^{\bar{\tau}} \beta_2^\tau (1-\beta_2) \cdot g_{t-\tau,j,r}^{(t-\tau)}[k]^2 \pm \widetilde{O}(\lambda\eta)}}$$

  $$= \frac{\sum_{\tau \in \mathcal{T}_k} \beta_1^\tau \cdot \Theta(g_{t-\tau_0,j,r}^{(t-\tau)}[k]) + (\Theta(1) - \sum_{\tau \in \mathcal{T}_k} \beta_1^\tau) \cdot \Theta(\widetilde{g}) \pm \widetilde{O}(\lambda\eta)}{\sqrt{\sum_{\tau=0}^{\bar{\tau}} \beta_2^\tau (1-\beta_2) \cdot g_{t-\tau,j,r}^{(t-\tau)}[k]^2 \pm \widetilde{O}(\lambda\eta)}}$$

  $$= \frac{\beta_1^{\tau_0} \left( \Theta\left( g_{t-\tau_0,j,r}^{(t)}[k] \right) \pm \Theta\left( \frac{|\ell_{j,i}^{(t)}|}{B} \right) \cdot \widetilde{O}(\eta\bar{\tau}s\sigma_p) \pm \Theta(\lambda\eta\bar{\tau}) - \Theta(\lambda\mathbf{w}_{j,r}^{(t)}[k]) \right) + \Theta(\widetilde{g})}{\sqrt{\sum_{\tau=0}^{\bar{\tau}} \beta_2^\tau (1-\beta_2) \cdot g_{t-\tau,j,r}^{(t-\tau)}[k]^2 \pm \widetilde{O}(\lambda\eta)}},$$

  where the third equality we denote $g_{t-\tau,j,r}^{(t-\tau)}[k] = \Theta(g_{t-\tau,j,r}^{(t)}[k]) \pm \Theta(\lambda\eta\bar{\tau}) = \Theta(\lambda\mathbf{w}_{j,r}^{(t)}[k]) \pm \Theta(\lambda\eta\bar{\tau})$ as $\widetilde{g}$ for $\tau \in \{0\} \cup [\bar{\tau}] \setminus \mathcal{T}_k$. For the denominator, we have

  $$\sqrt{\sum_{\tau=0}^{\bar{\tau}} \beta_2^\tau (1-\beta_2) \cdot g_{t-\tau,j,r}^{(t-\tau)}[k]^2}$$

  $$\le \sum_{\tau=0}^{\bar{\tau}} \beta_2^{\frac{\tau}{2}} (1-\beta_2)^{\frac{1}{2}} \cdot \left| g_{t-\tau,j,r}^{(t-\tau)}[k] \right|$$

  $$= \beta_2^{\frac{\tau_0}{2}} \left( \Theta\left( \left| g_{t-\tau_0,j,r}^{(t)}[k] \right| \right) \pm \Theta\left( \frac{|\ell_{j,i}^{(t)}|}{B} \right) \cdot \widetilde{O}(\eta\tau_0 s\sigma_p) \pm \Theta(\lambda\eta\bar{\tau}) - \Theta(|\lambda\mathbf{w}_{j,r}^{(t)}[k]|) \right) + \Theta(|\widetilde{g}|)$$

  $$\sqrt{\sum_{\tau=0}^{\bar{\tau}} \beta_2^\tau (1-\beta_2) \cdot g_{t-\tau,j,r}^{(t-\tau)}[k]^2}$$

  $$\ge \frac{1}{\sqrt{\bar{\tau}+1}} \sum_{\tau=0}^{\bar{\tau}} \beta_2^{\frac{\tau}{2}} (1-\beta_2)^{\frac{1}{2}} \cdot \left| g_{t-\tau,j,r}^{(t-\tau)}[k] \right|$$

  $$= \frac{1}{\sqrt{\bar{\tau}+1}} \beta_2^{\frac{\tau_0}{2}} \left( \Theta\left( \left| g_{t-\tau_0,j,r}^{(t)}[k] \right| \right) \pm \Theta\left( \frac{|\ell_{j,i}^{(t)}|}{B} \right) \cdot \widetilde{O}(\eta\tau_0 s\sigma_p) \pm \Theta(\lambda\eta\bar{\tau}) - \Theta(|\lambda\mathbf{w}_{j,r}^{(t)}[k]|) \right) + \Theta(|\widetilde{g}|).$$

  So we have

  $$\frac{\mathbf{m}_{j,r}^{(t)}[k]}{\sqrt{\mathbf{v}_{j,r}^{(t)}[k] + \epsilon}}$$

  $$= \frac{\beta_1^{\tau_0} \left( \widetilde{\Theta}\left( g_{t-\tau_0,j,r}^{(t)}[k] \right) \pm \widetilde{\Theta}\left( \frac{\eta s\sigma_p |\ell_{j,i}^{(t)}|}{B} \right) \pm \widetilde{\Theta}(\lambda\eta) - \widetilde{\Theta}(\lambda\mathbf{w}_{j,r}^{(t)}[k]) \right) + \widetilde{\Theta}(\lambda\mathbf{w}_{j,r}^{(t)}[k]) \pm \widetilde{\Theta}(\lambda\eta)}{\beta_2^{\frac{\tau_0}{2}} \left( \Theta\left( |g_{t-\tau_0,j,r}^{(t)}[k]| \right) \pm \widetilde{\Theta}\left( \frac{\eta s\sigma_p |\ell_{j,i}^{(t)}|}{B} \right) \pm \widetilde{\Theta}(\lambda\eta) - \Theta(|\lambda\mathbf{w}_{j,r}^{(t)}[k]|) \right) + \widetilde{\Theta}(|\lambda\mathbf{w}_{j,r}^{(t)}[k]|) \pm \widetilde{\Theta}(\lambda\eta)},$$

where we use the fact that $\epsilon = \Theta(\lambda\eta)$. Now we handle $\beta_1^{\tau_0}$ and $\beta_2^{\frac{\tau_0}{2}}$ with more care. First we have $\beta_1^{\tau_0} < \beta_2^{\frac{\tau_0}{2}}$, $|g_{t-\tau_0,j,r}^{(t)}[k]| = \widetilde{O}(1)$ and $\tau_0 < \frac{n}{B}$.

Then if $\frac{n}{B} = \Theta(1)$, we have $\beta_1^{\tau_0} = \beta_2^{\frac{\tau_0}{2}} = \Theta(1)$, then

$$\frac{\mathbf{m}_{j,r}^{(t)}[k]}{\sqrt{\mathbf{v}_{j,r}^{(t)}[k]} + \epsilon}$$

$$= \frac{\widetilde{\Theta}\left(g_{t-\tau_0,j,r}^{(t)}[k]\right) \pm \widetilde{\Theta}\left(\frac{\eta s\sigma_p|\ell_{j,i}^{(t)}|}{B}\right) \pm \widetilde{\Theta}(\lambda\eta) \pm \widetilde{\Theta}(\lambda\mathbf{w}_{j,r}^{(t)}[k])}{\Theta\left(|g_{t-\tau_0,j,r}^{(t)}[k]|\right) \pm \widetilde{\Theta}\left(\frac{\eta s\sigma_p|\ell_{j,i}^{(t)}|}{B}\right) \pm \widetilde{\Theta}(\lambda\eta) \pm \Theta(|\lambda\mathbf{w}_{j,r}^{(t)}[k]|)}$$

$$= \frac{\widetilde{\Theta}\left(-\frac{1}{B}\ell_{j,i}^{(t)}\sigma'(\langle\mathbf{w}_{j,r}^{(t)},\boldsymbol{\xi}_i\rangle)\cdot\boldsymbol{\xi}_i[k]\right) \pm \widetilde{\Theta}\left(\frac{\eta s\sigma_p|\ell_{j,i}^{(t)}|}{B}\right) \pm \widetilde{\Theta}(\lambda\eta) \pm \widetilde{\Theta}(\lambda\mathbf{w}_{j,r}^{(t)}[k])}{\Theta\left(\left|-\frac{1}{B}\ell_{j,i}^{(t)}\sigma'(\langle\mathbf{w}_{j,r}^{(t)},\boldsymbol{\xi}_i\rangle)\cdot\boldsymbol{\xi}_i[k]\right|\right) \pm \widetilde{\Theta}\left(\frac{\eta s\sigma_p|\ell_{j,i}^{(t)}|}{B}\right) \pm \widetilde{\Theta}(\lambda\eta) \pm \Theta(|\lambda\mathbf{w}_{j,r}^{(t)}[k]|)}.$$

If $\frac{n}{B} \geq \Theta(\log(\lambda\eta)^{-1}) = \widetilde{\Theta}(1)$ such that $\beta_2^{\frac{n}{2B}} \leq \Theta(\lambda\eta)$, then for $\tau_0 = \Theta(1)$

$$\frac{\mathbf{m}_{j,r}^{(t)}[k]}{\sqrt{\mathbf{v}_{j,r}^{(t)}[k]} + \epsilon}$$

$$= \frac{\widetilde{\Theta}\left(-\frac{1}{B}\ell_{j,i}^{(t)}\sigma'(\langle\mathbf{w}_{j,r}^{(t)},\boldsymbol{\xi}_i\rangle)\cdot\boldsymbol{\xi}_i[k]\right) \pm \widetilde{\Theta}\left(\frac{\eta s\sigma_p|\ell_{j,i}^{(t)}|}{B}\right) \pm \widetilde{\Theta}(\lambda\eta) \pm \widetilde{\Theta}(\lambda\mathbf{w}_{j,r}^{(t)}[k])}{\Theta\left(\left|-\frac{1}{B}\ell_{j,i}^{(t)}\sigma'(\langle\mathbf{w}_{j,r}^{(t)},\boldsymbol{\xi}_i\rangle)\cdot\boldsymbol{\xi}_i[k]\right|\right) \pm \widetilde{\Theta}\left(\frac{\eta s\sigma_p|\ell_{j,i}^{(t)}|}{B}\right) \pm \widetilde{\Theta}(\lambda\eta) \pm \Theta(|\lambda\mathbf{w}_{j,r}^{(t)}[k]|)}.$$

For $\tau_0 < \frac{n}{B}$, $\tau_0 \geq \Theta(\log(\lambda\eta)^{-1})$ such that $\beta_2^{\frac{\tau_0}{2}} \leq \Theta(\lambda\eta)$, we have

$$\frac{\mathbf{m}_{j,r}^{(t)}[k]}{\sqrt{\mathbf{v}_{j,r}^{(t)}[k]} + \epsilon} = \frac{\widetilde{\Theta}(\lambda\mathbf{w}_{j,r}^{(t)}[k]) \pm \widetilde{\Theta}(\lambda\eta)}{\widetilde{\Theta}(|\lambda\mathbf{w}_{j,r}^{(t)}[k]|) \pm \widetilde{\Theta}(\lambda\eta)},$$

since $|g_{t-\tau_0,j,r}^{(t)}[k]| = \widetilde{O}(1)$, $\lambda\eta = o(1)$ and $\eta s\sigma_p = o(1)$. We claim that the intersection (gap) of $\Theta(\log(\lambda\eta)^{-1})$ and $\Theta(1)$ is very small for $\Theta(\log(\lambda\eta)^{-1})$, that is, considering the intersection part $c\log(\lambda\eta)^{-1}$ for a sufficiently small constant $c > 0$, the impact of the intersection (gap) is very small, since $c\log(\lambda\eta)^{-1} = o(\log(\lambda\eta)^{-1})$. Therefore, for most of $\tau_0$

$$\frac{\mathbf{m}_{j,r}^{(t)}[k]}{\sqrt{\mathbf{v}_{j,r}^{(t)}[k]} + \epsilon} = \frac{\widetilde{\Theta}(\lambda\mathbf{w}_{j,r}^{(t)}[k]) \pm \widetilde{\Theta}(\lambda\eta)}{\widetilde{\Theta}(|\lambda\mathbf{w}_{j,r}^{(t)}[k]|) \pm \widetilde{\Theta}(\lambda\eta)}.$$

- For $k \neq 1$ and $k \notin \mathcal{B}_i, i \in \mathcal{I}_{t-\tau}, \tau \in [0, \bar{\tau}]$, recall $g_{t-\tau,j,r}^{(t)}[k] = \lambda\mathbf{w}_{j,r}^{(t)}[k]$, we have

$$\frac{\mathbf{m}_{j,r}^{(t)}[k]}{\sqrt{\mathbf{v}_{j,r}^{(t)}[k]} + \epsilon}$$

$$= \frac{\sum_{\tau=0}^{\bar{\tau}} \beta_1^\tau(1-\beta_1)\cdot g_{t-\tau,j,r}^{(t-\tau)}[k] \pm \widetilde{O}(\lambda\eta)}{\sqrt{\sum_{\tau=0}^{\bar{\tau}} \beta_2^\tau(1-\beta_2)\cdot g_{t-\tau,j,r}^{(t-\tau)}[k]^2} \pm \widetilde{O}(\lambda\eta)}$$

$$= \frac{\sum_{\tau=0}^{\bar{\tau}} \beta_1^\tau(1-\beta_1)\cdot\left(\Theta\left(g_{t-\tau,j,r}^{(t)}[k]\right) \pm \Theta(\lambda\eta\bar{\tau})\right) \pm \widetilde{O}(\lambda\eta)}{\sqrt{\sum_{\tau=0}^{\bar{\tau}} \beta_2^\tau(1-\beta_2)\cdot\left(\Theta\left(g_{t-\tau,j,r}^{(t)}[k]\right) \pm \Theta(\lambda\eta\bar{\tau})\right)^2} \pm \widetilde{O}(\lambda\eta)}$$

$$= \frac{\Theta(g_{t,j,r}^{(t)}[k]) \pm \widetilde{\Theta}(\lambda\eta)}{\Theta(|g_{t,j,r}^{(t)}[k]|) \pm \widetilde{\Theta}(\lambda\eta)}$$

$$= \frac{\Theta(\mathbf{w}_{j,r}^{(t)}[k]) \pm \widetilde{\Theta}(\eta)}{\Theta(|\mathbf{w}_{j,r}^{(t)}[k]|) \pm \widetilde{\Theta}(\eta)}.$$

This completes the proof. $\qquad\qquad\square$

### C.1.1 Proof of Theorem 4.1

**Lemma C.3** (Stage I). *Given the training dataset $\mathcal{S}$, if $\frac{n}{B} = \Theta(1)$, $\eta = 1/\mathrm{poly}(d)$ and $0 < \lambda = o(\sigma_0^{q-2}\sigma_p/n)$, then for any $t \leq T_0$ with $T_0 = \widetilde{O}(\frac{1}{\eta s \sigma_p})$ and any $i \in [n]$,*

$$\langle \mathbf{w}_{j,r}^{(t+1)}, j\mathbf{v} \rangle \leq \langle \mathbf{w}_{j,r}^{(t)}, j\mathbf{v} \rangle + \Theta(\eta),$$

$$\langle \mathbf{w}_{y_i,r}^{(t+1)}, \boldsymbol{\xi}_i \rangle = \langle \mathbf{w}_{y_i,r}^{(t)}, \boldsymbol{\xi}_i \rangle + \widetilde{\Theta}(\eta s \sigma_p).$$

*Proof of Lemma C.3.* By Lemma C.1, we have

$$\langle \mathbf{w}_{j,r}^{(t+1)}, j\mathbf{v} \rangle \leq \langle \mathbf{w}_{j,r}^{(t)}, j\mathbf{v} \rangle - \eta \left\langle \frac{\mathbf{m}_{j,r}^{(t)}}{\sqrt{\mathbf{v}_{j,r}^{(t)}} + \epsilon}, j\mathbf{v} \right\rangle$$

$$\leq \langle \mathbf{w}_{j,r}^{(t)}, j\mathbf{v} \rangle + \Theta(\eta).$$

Then, we prove $\langle \mathbf{w}_{y_i,r}^{(t+1)}, \boldsymbol{\xi}_i \rangle = \langle \mathbf{w}_{y_i,r}^{(t)}, \boldsymbol{\xi}_i \rangle + \widetilde{\Theta}(\eta s \sigma_p)$ by induction. By Lemma A.3, we have

$$|\langle \mathbf{w}_{j,r}^{(0)}, \mathbf{v} \rangle| = \widetilde{\Theta}(\sigma_0), \quad |\langle \mathbf{w}_{j,r}^{(0)}, \boldsymbol{\xi}_i \rangle| = \widetilde{\Theta}(s^{1/2}\sigma_p\sigma_0), \quad \mathbf{w}_{j,r}^{(0)}[k] = \widetilde{\Theta}(\sigma_0),$$

which imply that $|\ell_{j,i}^{(0)}| = \Theta(1)$. Additionally, we have $\eta s = o(\sigma_0^{q-1})$ and $0 < \lambda = o(\sigma_0^{q-2}\sigma_p/n)$. For a sufficiently large fraction of $k \in \mathcal{B}_i$ (e.g., 0.99), we have $|B^{-1}\sigma_0^{q-1}\boldsymbol{\xi}_i[k]| \geq \widetilde{\Theta}(\eta B^{-1}s\sigma_p|\ell_{j,i}^{(0)}| + \lambda|\mathbf{w}_{j,r}^{(0)}[k]|)$ for $i \in \mathcal{I}_\tau$. Therefore, by Lemma C.2 and A.6, we have

$$\mathrm{sgn}\left( -\frac{1}{B}\ell_{y_i,i}^{(0)}\sigma'(\langle \mathbf{w}_{y_i,r}^{(0)}, \boldsymbol{\xi}_i \rangle)\boldsymbol{\xi}_i[k] \right)$$

$$= -\mathrm{sgn}\left( \ell_{y_i,i}^{(0)}\sigma'(\langle \mathbf{w}_{y_i,r}^{(0)}, \boldsymbol{\xi}_i \rangle)\boldsymbol{\xi}_i[k] \right)$$

$$= -\mathrm{sgn}(\boldsymbol{\xi}_i[k]). \tag{C.11}$$

Recall $\frac{n}{B} = \Theta(1)$. By Lemma C.2 we have the following update according to (B.4), (C.11) and Lemma C.1.

$$\langle \mathbf{w}_{y_i,r}^{(1)}, \boldsymbol{\xi}_i \rangle$$

$$= \langle \mathbf{w}_{y_i,r}^{(0)}, \boldsymbol{\xi}_i \rangle - \eta \cdot \left\langle \frac{\mathbf{m}_{j,r}^{(0)}}{\sqrt{\mathbf{v}_{j,r}^{(0)}} + \epsilon}, j\mathbf{v} \right\rangle$$

$$\geq \langle \mathbf{w}_{y_i,r}^{(0)}, \boldsymbol{\xi}_i \rangle + \Theta(\eta) \cdot \sum_{k \in \mathcal{B}_i} \langle \mathrm{sgn}(\boldsymbol{\xi}_i[k]), \boldsymbol{\xi}_i \rangle - O(\eta\alpha) - O(\eta s \sigma_p)$$

$$= \langle \mathbf{w}_{y_i,r}^{(0)}, \boldsymbol{\xi}_i \rangle + \widetilde{\Theta}(\eta s \sigma_p).$$

For general $t \leq T_0$, assuming $\langle \mathbf{w}_{y_i,r}^{(t)}, \boldsymbol{\xi}_i \rangle \geq \langle \mathbf{w}_{y_i,r}^{(t-1)}, \boldsymbol{\xi}_i \rangle + \widetilde{\Theta}(\eta s \sigma_p)$. Then we have

$$\langle \mathbf{w}_{y_i,r}^{(t)}, \boldsymbol{\xi}_i \rangle = \langle \mathbf{w}_{y_i,r}^{(0)}, \boldsymbol{\xi}_i \rangle + \widetilde{\Theta}(t\eta s \sigma_p) = \widetilde{\Theta}(s^{1/2}\sigma_p\sigma_0 + t\eta s \sigma_p) \leq \widetilde{\Theta}(1).$$

By Lemma C.1, we have

$$|\mathbf{w}_{j,r}^{(t)}[k]| \leq |\mathbf{w}_{j,r}^{(t-1)}[k]| + \Theta(\eta) \leq \widetilde{\Theta}(\sigma_0 + t\eta) \leq \widetilde{\Theta}(1).$$

So we have $|\ell_{j,i}^{(t)}| = \Theta(1)$. Besides, we still establish the condition $|B^{-1}(\mathbf{w}_{j,r}^{(0)}[k]+t\eta s\sigma_p)^{q-1}\boldsymbol{\xi}_i[k]| \geq \widetilde{\Theta}(\eta B^{-1}s\sigma_p|\ell_{j,i}^{(0)}|+\lambda|\mathbf{w}_{j,r}^{(0)}[k]+t\eta|)$ since $0 < \lambda = o(\sigma_0^{q-2}\sigma_p/n)$. Then we have (C.11) for $t$. Follow the same proof above, we have

$$\langle \mathbf{w}_{y_i,r}^{(t+1)}, \boldsymbol{\xi}_i \rangle$$
$$= \langle \mathbf{w}_{y_i,r}^{(t)}, \boldsymbol{\xi}_i \rangle - \eta \cdot \left\langle \frac{\mathbf{m}_{j,r}^{(t)}}{\sqrt{\mathbf{v}_{j,r}^{(t)}} + \epsilon}, j\mathbf{v} \right\rangle$$
$$\geq \langle \mathbf{w}_{y_i,r}^{(t)}, \boldsymbol{\xi}_i \rangle + \Theta(\eta) \cdot \sum_{k \in \mathcal{B}_i} \langle \mathrm{sgn}(\boldsymbol{\xi}_i[k]), \boldsymbol{\xi}_i \rangle - O(\eta\alpha) - O(\eta s\sigma_p)$$
$$= \langle \mathbf{w}_{y_i,r}^{(t)}, \boldsymbol{\xi}_i \rangle + \widetilde{\Theta}(\eta s\sigma_p).$$

The term $O(\eta s\sigma_p)$ in the above inequality arises because, for coordinates that $|\boldsymbol{\xi}_i[k]| \leq O(\sigma_p)$, we cannot exploit sign information. Instead, we directly apply Lemma C.1. This completes the proof. $\qquad\square$

Lemma C.3 coincides with Lemma A.3 of (Zou et al., 2023b), allowing us to transfer their full-batch Adam analysis to the large-batch case under $\frac{n}{B} = \Theta(1)$. Therefore, the remaining proofs are omitted for brevity, as they coincide with those in Zou et al. (2023b).

### C.1.2   Proof of Theorem 4.2

**Lemma C.4** (StageI, nearly zero noise memorization). *Given the training dataset $\mathcal{S}$, if $\frac{n}{B} \geq \Theta(\log \epsilon^{-1})$, $\eta = 1/\mathrm{poly}(d)$ and $0 < \lambda = o(\sigma_0^{q-2}\sigma_p/n)$, then for any $t \leq T_0$ with $T_0 = \widetilde{O}(\frac{1}{\eta})$ and any $i \in [n]$, suppose $\langle \mathbf{w}_{y_i,r}^{(t)}, y_i \cdot \mathbf{v} \rangle > -\widetilde{\Theta}(\sigma_0)$, then*
$$\left\langle \mathbf{w}_{j,r}^{(t)}, \boldsymbol{\xi}_i \right\rangle \leq \widetilde{\Theta}(\sqrt{s}\sigma_p\sigma_0 + \alpha).$$

*Proof of Lemma C.4.* By Lemma A.3, at initialization
$$\left\langle \mathbf{w}_{j,r}^{(0)}, \boldsymbol{\xi}_i \right\rangle \leq \widetilde{\Theta}(\sqrt{s}\,\sigma_p\sigma_0 + \alpha\sigma_0) \leq \widetilde{\Theta}(\sqrt{s}\,\sigma_p\sigma_0),$$
since $\alpha = o(1)$. Lemma C.2 ensures that stochastic updates slow down noise memorization—allowing $\langle \mathbf{w}_{j,r}^{(t)}, \boldsymbol{\xi}_i \rangle$ to grow for only
$$o\big(\log(\lambda\eta)^{-1}\big)$$
iterations after $\boldsymbol{\xi}_i$ is sampled—while in the remaining
$$\frac{n}{B} - o\big(\log(\lambda\eta)^{-1}\big) \gg o\big(\log(\lambda\eta)^{-1}\big)$$
iterations, weight decay dominates. In particular, whenever $|\mathbf{w}_{j,r}^{(t)}[k]| \geq \widetilde{\Theta}(\eta)$ we have
$$\mathbf{w}_{j,r}^{(t+1)}[k] = \mathbf{w}_{j,r}^{(t)}[k] - \mathrm{sgn}\big(\mathbf{w}_{j,r}^{(t)}[k]\big) \cdot \Theta(\eta).$$

Concretely, if $\boldsymbol{\xi}_i$ is sampled at iteration $\tau_1$ of the first epoch, then
$$\left\langle \mathbf{w}_{j,r}^{(\tau_1+1)}, \boldsymbol{\xi}_i \right\rangle \leq \left\langle \mathbf{w}_{j,r}^{(\tau_1)}, \boldsymbol{\xi}_i \right\rangle + \widetilde{\Theta}(\eta s\sigma_p) \leq \widetilde{\Theta}(\sqrt{s}\,\sigma_p\sigma_0 + \eta s\sigma_p) \leq \widetilde{\Theta}(\sqrt{s}\,\sigma_p\sigma_0),$$
where we directly bound the update by $\Theta(1)$ according to Lemma C.1. Over the next $o(\log(\lambda\eta)^{-1})$ iterations the noise memorization $\langle \mathbf{w}_{j,r}^{(t)}, \boldsymbol{\xi}_i \rangle$ increases by at most
$$o\big(\log(\lambda\eta)^{-1}\big) \cdot \Theta(\eta s\sigma_p),$$
and thereafter weight decay decreases it in each of the remaining $\frac{n}{B} - o(\log(\lambda\eta)^{-1})$ steps. Hence, we can calculate the maximum value of the noise memorization
$$\left\langle \mathbf{w}_{j,r}^{(n/B)}, \boldsymbol{\xi}_i \right\rangle \leq \max\left\{ \widetilde{\Theta}(\sqrt{s}\,\sigma_p\sigma_0), \ \widetilde{\Theta}(\sqrt{s}\,\sigma_p\sigma_0 + \alpha) + \Theta(\eta s\sigma_p)\big[o(\log(\lambda\eta)^{-1}) - \tfrac{n}{B}\big] \right\}$$
$$\leq \widetilde{\Theta}(\sqrt{s}\,\sigma_p\sigma_0 + \alpha),$$
since $n/B \geq \Theta\big(\log(\lambda\eta)^{-1}\big)$ and $\boldsymbol{\xi}_i[1] = -\alpha y_i$. This is true in every epoch. So we complete the proof. $\qquad\square$

**Lemma C.5** (Stage I, feature learning). *Given the training dataset $\mathcal{S}$, if $\frac{n}{B} \geq \Theta(\log \epsilon^{-1})$, $\eta = 1/\text{poly}(d)$ and $0 < \lambda = o(\sigma_0^{q-2}\sigma_p/n)$, then for any $t \leq T_0$ with $T_0 = \widetilde{O}(\frac{1}{\eta})$ and $j \in \{\pm 1\}$, we have*

$$\langle \mathbf{w}_{j,r}^{(t+1)}, j\mathbf{v} \rangle = \langle \mathbf{w}_{j,r}^{(t)}, j\mathbf{v} \rangle + \Theta(\eta).$$

*Proof of Lemma C.5.* by Lemma A.7, we have

$$\text{sgn}(g_{0,j,r}^{(0)}) = -\text{sgn}\left( \sum_{i \in \mathcal{I}_0} y_i \ell_{j,i}^{(0)} \sigma'(\langle \mathbf{w}_{j,r}^{(0)}, y_i \mathbf{v} \rangle) - \alpha \sum_{i \in \mathcal{I}_0} y_i \ell_{j,i}^{(0)} \sigma'(\langle \mathbf{w}_{j,r}^{(0)}, \boldsymbol{\xi}_i \rangle) - B\lambda \mathbf{w}_{j,r}^{(0)}[1] \right).$$

Then with Lemma A.3 and facts that $\ell_{j,i}^{(0)} = \Theta(1)$, by Lemma A.6, we have

$$y_i \ell_{j,i}^{(0)} \sigma'(\langle \mathbf{w}_{j,r}^{(0)}, y_i \mathbf{v} \rangle) = \text{sgn}(j) \cdot \widetilde{\Theta}(\sigma_0^{q-1}),$$
$$\alpha y_i \ell_{j,i}^{(0)} \sigma'(\langle \mathbf{w}_{j,r}^{(0)}, \boldsymbol{\xi}_i \rangle) = \text{sgn}(j) \cdot \widetilde{\Theta}(\alpha(s^{1/2}\sigma_p\sigma_0)^{q-1}),$$
$$\lambda \mathbf{w}_{j,r}^{(0)}[1] = \pm o(\sigma_0^{q-1}\sigma_p).$$

Substituting them into $g_{0,j,r}^{(0)}$, with $\alpha = o(1)$, $s^{1/2}\sigma_p = \widetilde{O}(1)$, $\lambda = o(\sigma_0^{q-2}\sigma_p/n)$, we get

$$\text{sgn}(g_{0,j,r}^{(0)}) = -\text{sgn}(j \cdot \widetilde{\Theta}(\sigma_0^{q-1})) = -\text{sgn}(j).$$

By Lemma C.2 and $\eta = o(\sigma_0^{q-1})$, we have

$$\langle \mathbf{w}_{j,r}^{(1)}, j\mathbf{v} \rangle = \langle \mathbf{w}_{j,r}^{(0)}, j\mathbf{v} \rangle - \eta \left\langle \frac{\mathbf{m}_{j,r}^{(0)}}{\sqrt{\mathbf{v}_{j,r}^{(0)} + \epsilon}}, j \cdot \mathbf{v} \right\rangle$$

$$= \langle \mathbf{w}_{j,r}^{(0)}, j\mathbf{v} \rangle + j \cdot \text{sgn}(j) \cdot \Theta(\eta)$$

$$= \langle \mathbf{w}_{j,r}^{(0)}, j\mathbf{v} \rangle + \Theta(\eta).$$

Now suppose that the equality holds for iterations $0, \ldots, t$. Then $\langle \mathbf{w}_{j,r}^{(t)}, j \cdot \mathbf{v} \rangle = \widetilde{O}(1)$, $\langle \mathbf{w}_{y_i,r}^{(t)}, \boldsymbol{\xi}_i \rangle = \widetilde{\Theta}(\eta s \sigma_p) = O(1)$. Therefore, $\ell_{j,i}^{(t)} = \Theta(1)$. By Lemma C.4, we have

$$y_i \ell_{j,i}^{(t)} \sigma'(\langle \mathbf{w}_{j,r}^{(t)}, y_i \mathbf{v} \rangle) = \text{sgn}(j) \cdot \widetilde{\Theta}((\sigma_0 + t\eta)^{q-1}),$$
$$\alpha y_i \ell_{j,i}^{(t)} \sigma'(\langle \mathbf{w}_{j,r}^{(t)}, \boldsymbol{\xi}_i \rangle) \leq \text{sgn}(j) \cdot \widetilde{\Theta}(\alpha(s^{1/2}\sigma_p\sigma_0 + t\eta)^{q-1}),$$
$$\lambda \mathbf{w}_{j,r}^{(t)}[1] = \pm o(\sigma_0^{q-2}\sigma_p(\sigma_0 + t\eta)).$$

Substituting them into $g_{t,j,r}^{(t)}$, with $\alpha = o(1)$, $s^{1/2}\sigma_p = \widetilde{O}(1)$, $\lambda = o(\sigma_0^{q-2}\sigma_p/n)$, we get

$$\text{sgn}(g_{t,j,r}^{(t)}) = -\text{sgn}(j \cdot \widetilde{\Theta}(\sigma_0^{q-1})) = -\text{sgn}(j).$$

By Lemma C.2 and $\eta = o(\sigma_0^{q-1})$, we have

$$\langle \mathbf{w}_{j,r}^{(t+1)}, j\mathbf{v} \rangle = \langle \mathbf{w}_{j,r}^{(t)}, j\mathbf{v} \rangle - \eta \left\langle \frac{\mathbf{m}_{j,r}^{(t)}}{\sqrt{\mathbf{v}_{j,r}^{(t)} + \epsilon}}, j \cdot \mathbf{v} \right\rangle$$

$$= \langle \mathbf{w}_{j,r}^{(t)}, j\mathbf{v} \rangle + j \cdot \text{sgn}(j) \cdot \Theta(\eta)$$

$$= \langle \mathbf{w}_{j,r}^{(t)}, j\mathbf{v} \rangle + \Theta(\eta).$$

This completes the proof. $\qquad\square$

**Lemma C.6** (Stage I, general dynamics). *Given the training dataset $\mathcal{S}$, if $\frac{n}{B} \geq \Theta(\log \epsilon^{-1})$, $\eta = 1/\text{poly}(d)$ and $0 < \lambda = o(\sigma_0^{q-2}\sigma_p/n)$, then for any $t \leq T_0$ with $T_0 = \widetilde{O}(\frac{1}{\eta})$ and any $i \in [n]$,*

$$\langle \mathbf{w}_{j,r}^{(t+1)}, j \cdot \mathbf{v} \rangle = \langle \mathbf{w}_{j,r}^{(t)}, j \cdot \mathbf{v} \rangle + \Theta(\eta \cdot \frac{n}{B}),$$

$$\langle \mathbf{w}_{j,r}^{(t+1)}, \boldsymbol{\xi}_i \rangle \leq \widetilde{\Theta}(\sqrt{s}\sigma_p\sigma_0 + \alpha).$$

*Proof of Lemma C.6.* We prove the claim by induction on $t$, using Lemma C.4 and C.5. At $t = 0$, by Lemma A.3, at initialization we have

$$\left|\langle \mathbf{w}_{j,r}^{(0)}, j\mathbf{v}\rangle\right| \leq \widetilde{\Theta}(\sigma_0),$$

and hence

$$\langle \mathbf{w}_{j,r}^{(0)}, j\mathbf{v}\rangle \geq -\widetilde{\Theta}(\sigma_0).$$

Therefore Lemma C.4 holds at $t = 0$. Suppose Lemma C.4 and C.5 for some $t \geq 0$, then

$$\langle \mathbf{w}_{j,r}^{(t+1)}, j\mathbf{v}\rangle \geq -\widetilde{\Theta}(\sigma_0)$$

by exactly the same proof used in the proof of Lemma C.5. This lower bound remains valid at step $t + 1$. Consequently, Lemma C.4 continues to hold, and the induction carries through all iterations. This completes the proof. $\qquad\square$

**Lemma C.7** (Stage II). *Given the training dataset $\mathcal{S}$, if $\frac{n}{B} \geq \Theta(\log \epsilon^{-1})$, $\eta = 1/\mathrm{poly}(d)$ and $0 < \lambda = o(\sigma_0^{q-2}\sigma_p/n)$, then for any $t > T_0$, $j \in \{\pm 1\}$, $r \in [m]$, $i \in [n]$, let $r^* = \mathrm{argmax}_{r\in[m]}\langle \mathbf{w}_{j,r}^{(t)}, j\mathbf{v}\rangle$, then $\langle \mathbf{w}_{j,r^*}^{(t)}, j\mathbf{v}\rangle = \widetilde{\Theta}(1)$ and $\langle \mathbf{w}_{j,r}^{(t)}, \boldsymbol{\xi}_i\rangle \leq \widetilde{\Theta}(\eta s\sigma_p + \alpha)$.*

*Proof of Lemma C.7.* We begin by establishing the bound $\langle \mathbf{w}_{j,r}^{(t)}, \boldsymbol{\xi}_i\rangle \leq \widetilde{\Theta}(\eta s\sigma_p + \alpha)$. According to Lemma C.4, during the first $T_0$ epochs, the number of iterations in which weight decay dominates the update dynamics is at least

$$T_0 \cdot \left(\frac{n}{B} - o\left(\log(\lambda\eta)^{-1}\right)\right).$$

In each such iteration, the contribution from weight decay is lower bounded by $\widetilde{\Theta}(\eta s\sigma_p)$, leading to a cumulative effect of

$$\widetilde{\Theta}\left(T_0 \cdot \left(\frac{n}{B} - o\left(\log(\lambda\eta)^{-1}\right)\right) \cdot \eta s\sigma_p\right) = \widetilde{\Theta}\left(\frac{ns\sigma_p}{B}\right).$$

This term is asymptotically larger than $\widetilde{\Theta}(\sqrt{s}\sigma_p\sigma_0)$, i.e., $ns\sigma_p/B = \omega(\sqrt{s}\sigma_p\sigma_0)$. Therefore, we conclude that over the first $T_0$ epochs, the weight decay effectively suppresses noise memorization, ensuring that $\langle \mathbf{w}_{j,r}^{(t)}, \boldsymbol{\xi}_i\rangle \leq \langle \mathbf{w}_{j,r}^{(t)}, -\alpha y_i \mathbf{v}\rangle + \widetilde{\Theta}(\eta s\sigma_p) \leq \widetilde{\Theta}(\eta s\sigma_p + \alpha)$ holds.

Next, we focus on $\langle \mathbf{w}_{j,r^*}^{(t)}, j\mathbf{v}\rangle = \widetilde{\Theta}(1)$ for $r^* = \mathrm{argmax}_{r\in[m]}\langle \mathbf{w}_{j,r}^{(t)}, j\mathbf{v}\rangle$. By Lemma C.6, we know $\langle \mathbf{w}_{j,r^*}^{(T_0)}, j \cdot \mathbf{v}\rangle = \widetilde{\Theta}(1)$ and $\ell_{j,r}^{(T_0)} = \Theta(1)$. For $t > T_0$, We show if $\langle \mathbf{w}_{j,r^*}^{(t)}, j \cdot \mathbf{v}\rangle \leq \left(\frac{1}{m}\log\left((\lambda)^{-1} - 1\right)\right)^{\frac{1}{q}}$, then for $(\mathbf{x}_i, y_i)$ with $y_i = j$,

$$\ell_{j,i}^{(t)} = \frac{e^{F_{-j}(\mathbf{W}^{(t)}, \mathbf{x}_i)}}{\sum_{j\in\{-1,1\}} e^{F_j(\mathbf{W}^{(t)}, \mathbf{x}_i)}}$$

$$= \frac{1}{1 + \exp\left[\sum_{r=1}^m \sigma(\langle \mathbf{w}_{j,r}^{(t)}, j\mathbf{v}\rangle) + \sigma(\langle \mathbf{w}_{j,r}^{(t)}, \boldsymbol{\xi}_i\rangle) - \sigma(\langle \mathbf{w}_{-j,r}^{(t)}, j\mathbf{v}\rangle) - \sigma(\langle \mathbf{w}_{-j,r}^{(t)}, \boldsymbol{\xi}_i\rangle)\right]}$$

$$\geq \frac{1}{1 + \exp\left[\sum_{r=1}^m \sigma(\langle \mathbf{w}_{j,r}^{(t)}, j\mathbf{v}\rangle)\right]}$$

$$= \frac{1}{1 + \exp\left[m \cdot \left(\frac{1}{m}\log\left((\lambda)^{-1} - 1\right)\right)^{\frac{1}{q}\cdot q}\right]}$$

$$= \Theta(\lambda).$$

where the inequality we use $\langle \mathbf{w}_{j,r}^{(t)}, \boldsymbol{\xi}_i\rangle \leq \widetilde{\Theta}(\eta s\sigma_p + \alpha)$ and $\eta s\sigma_p = o(1)$, $\alpha = o(1)$. Then we have

$$\mathrm{sgn}\left(\sum_{i\in\mathcal{I}_t} y_i\ell_{j,i}^{(t)}\sigma'(\langle \mathbf{w}_{j,r}^{(t)}, y_i\mathbf{v}\rangle) - \alpha\sum_{i\in\mathcal{I}_t} y_i\ell_{j,i}^{(t)}\sigma'(\langle \mathbf{w}_{j,r}^{(t)}, \boldsymbol{\xi}_i\rangle) - B\lambda\mathbf{w}_{j,r}^{(t)}[1]\right)$$

$$= \mathrm{sgn}\left(\mathrm{sgn}(j) \cdot \lambda\left(\frac{1}{m}\log\left((\lambda)^{-1} - 1\right)\right)^{\frac{q-1}{q}} \pm \lambda\left(\frac{1}{m}\log\left((\lambda)^{-1} - 1\right)\right)^{\frac{1}{q}}\right)$$

$$= \text{sgn}(j),$$

where we use Lemma A.6, $\langle \mathbf{w}_{j,r}^{(t)}, \boldsymbol{\xi}_i \rangle \leq \widetilde{\Theta}(\eta s \sigma_p + \alpha)$ and $\alpha = o(1)$. So we have

$$\langle \mathbf{w}_{j,r^*}^{(t+1)}, j \cdot \mathbf{v} \rangle \geq \langle \mathbf{w}_{j,r^*}^{(t)}, j \cdot \mathbf{v} \rangle + \Theta(\eta).$$

If $\langle \mathbf{w}_{j,r^*}^{(t)}, j \cdot \mathbf{v} \rangle \geq \log(\lambda^{-\frac{1}{2}})$, then $\ell_{j,i}^{(t)} = o(\lambda)$, then for $(\mathbf{x}_i, y_i)$ with $y_i = j$,

$$\ell_{j,i}^{(t)} = \frac{e^{F_{-j}(\mathbf{W}^{(t)}, \mathbf{x}_i)}}{\sum_{j \in \{-1,1\}} e^{F_j(\mathbf{W}^{(t)}, \mathbf{x}_i)}}$$

$$= \frac{1}{1 + \exp\left[\sum_{r=1}^{m} \sigma(\langle \mathbf{w}_{j,r}^{(t)}, j\mathbf{v} \rangle) + \sigma(\langle \mathbf{w}_{j,r}^{(t)}, \boldsymbol{\xi}_i \rangle) - \sigma(\langle \mathbf{w}_{-j,r}^{(t)}, j\mathbf{v} \rangle) - \sigma(\langle \mathbf{w}_{-j,r}^{(t)}, \boldsymbol{\xi}_i \rangle)\right]}$$

$$\leq \frac{1}{1 + \exp\left[m(1 - \alpha) \cdot \left(\log(\lambda^{-\frac{1}{2}})\right)^q\right]}$$

$$\leq \frac{1}{\exp\left[\left(\log(\lambda^{-\frac{1}{2}})\right)^q\right]}$$

$$\leq \frac{1}{\lambda^{-\frac{q}{2}}}$$

$$= o\left(\frac{\lambda}{\left(\log(\lambda^{-\frac{1}{2}})\right)^{q-2}}\right).$$

Then we have

$$\text{sgn}\left(\sum_{i \in \mathcal{I}_t} y_i \ell_{j,i}^{(t)} \sigma'(\langle \mathbf{w}_{j,r}^{(t)}, y_i \mathbf{v} \rangle) - \alpha \sum_{i \in \mathcal{I}_t} y_i \ell_{j,i}^{(t)} \sigma'(\langle \mathbf{w}_{j,r}^{(t)}, \boldsymbol{\xi}_i \rangle) - B\lambda \mathbf{w}_{j,r}^{(t)}[1]\right)$$

$$= \text{sgn}\left(\text{sgn}(j) \cdot \left(\log(\lambda^{-\frac{1}{2}})\right)^{q-1} \cdot o\left(\frac{\lambda}{\left(\log(\lambda^{-\frac{1}{2}})\right)^{q-2}}\right) \pm \lambda \log(\lambda^{-\frac{1}{2}})\right)$$

$$= \text{sgn}\left(\text{sgn}(j) \cdot \lambda \cdot o\left(\log(\lambda^{-\frac{1}{2}})\right) \pm \lambda \log(\lambda^{-\frac{1}{2}})\right)$$

$$= \text{sgn}(-\mathbf{w}_{j,r}^{(t)}[1])$$

$$= -\text{sgn}(\mathbf{w}_{j,r}^{(t)}[1]).$$

So we have

$$\langle \mathbf{w}_{j,r^*}^{(t+1)}, j \cdot \mathbf{v} \rangle \geq \langle \mathbf{w}_{j,r^*}^{(t)}, j \cdot \mathbf{v} \rangle - \Theta(\eta).$$

Therefore, $\langle \mathbf{w}_{j,r^*}^{(t)}, j \cdot \mathbf{v} \rangle = \widetilde{\Theta}(1)$ for $t > T_0 = \frac{1}{\eta}$. This completes the proof. $\qquad \square$

**Lemma C.8** (Convergence). *Suppose the same conditions hold as in Lemma C.6 and C.7, if the step size $\eta = O(d^{-\frac{1}{2}})$, then for any $t$,*

$$\mathbb{E}\left[L(\mathbf{W}^{(t+1)}) - L(\mathbf{W}^{(t)})\right] \leq -\eta \|\nabla L(\mathbf{W}^{(t)})\|_1 + \widetilde{\Theta}(\eta^2 d).$$

*Proof of Lemma C.8.* We aim to prove the convergence of the objective function under the Adam optimization algorithm in a non-convex setting. Recall the loss function for each data point $i$ is

$$L_i(\mathbf{W}) = \log\left(1 + \frac{1}{\exp\left(F_{y_i}(\mathbf{W}, \mathbf{x}_i) - F_{-y_i}(\mathbf{W}, \mathbf{x}_i)\right)}\right),$$

where $\mathbf{W}$ represents the parameter matrix, $\mathbf{x}_i$ is the input data, $y_i$ is the true label, and $F_j(\mathbf{W}, \mathbf{x}_i)$ are the logits for class $j$. The total objective is:

$$L(\mathbf{W}) = \frac{1}{n} \sum_{i=1}^{n} L_i(\mathbf{W}) + \lambda \|\mathbf{W}\|_F^2,$$

with $\lambda = o(1)$ as a small regularization parameter.

Since $L_i(\mathbf{W})$ is non-convex, we exploit its smoothness with respect to the logits $[F_j(\mathbf{W}, \mathbf{x}_i)]_j$. Specifically, $L_i(\mathbf{W})$ is 1-smooth in $[F_j(\mathbf{W}, \mathbf{x}_i)]_j$ due to the properties of the cross-entropy loss. Define:

$$\Delta F_{j,i} = F_j(\mathbf{W}^{(t+1)}, \mathbf{x}_i) - F_j(\mathbf{W}^{(t)}, \mathbf{x}_i).$$

Using the smoothness property, we apply a second-order Taylor-like expansion around $\mathbf{W}^{(t)}$:

$$L_i(\mathbf{W}^{(t+1)}) - L_i(\mathbf{W}^{(t)}) \leq \sum_j \frac{\partial L_i(\mathbf{W}^{(t)})}{\partial F_j(\mathbf{W}^{(t)}, \mathbf{x}_i)} \cdot \Delta F_{j,i} + \sum_j (\Delta F_{j,i})^2. \qquad \text{(C.12)}$$

This upper bound arises because the second derivative of $L_i$ with respect to the logits is bounded by 1, a standard result for cross-entropy loss. The logits are defined as: $F_j(\mathbf{W}^{(t)}, \mathbf{x}_i) = \sum_{r=1}^{m}[\sigma(\langle \mathbf{w}_{j,r}^{(t)}, y_i\mathbf{v} \rangle) + \sigma(\langle \mathbf{w}_{j,r}^{(t)}, \boldsymbol{\xi}_i \rangle)]$, where $\mathbf{w}_{j,r}^{(t)}$ the $r$-th neuron in $j$-th output of $\mathbf{W}^{(t)}$, $\sigma(z) = [z]_+^q$ is a smooth activation function (e.g., with $q \geq 3$). By Lemma C.7 and C.1, we have $\langle \mathbf{w}_{j,r}^{(t)}, \mathbf{v} \rangle \leq \widetilde{\Theta}(1)$ and $\langle \mathbf{w}_{j,r}^{(t)}, \boldsymbol{\xi}_i \rangle \leq \widetilde{\Theta}(1)$, ensuring the local smoothness of $\sigma$ remains $\widetilde{O}(1)$ between $\langle \mathbf{w}_{j,r}^{(t+1)}, y_i\mathbf{v} \rangle$ and $\langle \mathbf{w}_{j,r}^{(t)}, y_i\mathbf{v} \rangle$ (similar for $\langle \mathbf{w}_{j,r}^{(t)}, \boldsymbol{\xi}_i \rangle$). Then with Taylor expansion, we have

$$\left| \sigma(\langle \mathbf{w}_{j,r}^{(t+1)}, y_i\mathbf{v} \rangle) - \sigma(\langle \mathbf{w}_{j,r}^{(t)}, y_i\mathbf{v} \rangle) - \langle \nabla_{\mathbf{w}_{j,r}} \sigma(\langle \mathbf{w}_{j,r}^{(t)}, y_i\mathbf{v} \rangle), \mathbf{w}_{j,r}^{(t+1)} - \mathbf{w}_{j,r}^{(t)} \rangle \right|$$

$$\leq \widetilde{\Theta}(1) \cdot \left\| \mathbf{w}_{j,r}^{(t+1)} - \mathbf{w}_{j,r}^{(t)} \right\|_2^2$$

$$= \widetilde{\Theta}(1) \cdot \left\| \eta \cdot \frac{\mathbf{m}_{j,r}^{(t)}}{\sqrt{\mathbf{v}_{j,r}^{(t)}} + \epsilon} \right\|_2^2$$

$$\leq \widetilde{\Theta}(\eta^2 d), \qquad \text{(C.13)}$$

where the last inequality we use Lemma C.1. Similarly, we have

$$\left| \sigma(\langle \mathbf{w}_{j,r}^{(t+1)}, \boldsymbol{\xi}_i \rangle) - \sigma(\langle \mathbf{w}_{j,r}^{(t)}, \boldsymbol{\xi}_i \rangle) - \langle \nabla_{\mathbf{w}_{j,r}} \sigma(\langle \mathbf{w}_{j,r}^{(t)}, \boldsymbol{\xi}_i \rangle), \mathbf{w}_{j,r}^{(t+1)} - \mathbf{w}_{j,r}^{(t)} \rangle \right|$$

$$\leq \widetilde{\Theta}(1) \cdot \left\| \mathbf{w}_{j,r}^{(t+1)} - \mathbf{w}_{j,r}^{(t)} \right\|_2^2$$

$$= \widetilde{\Theta}(1) \cdot \left\| \eta \cdot \frac{\mathbf{m}_{j,r}^{(t)}}{\sqrt{\mathbf{v}_{j,r}^{(t)}} + \epsilon} \right\|_2^2$$

$$\leq \widetilde{\Theta}(\eta^2 d). \qquad \text{(C.14)}$$

Summing over $r$ (with $m = \widetilde{\Theta}(1)$), we get

$$|\Delta F_{j,i}| \leq \left| \langle \nabla_{\mathbf{W}} F_j(\mathbf{W}^{(t)}, \mathbf{x}_i), \mathbf{W}^{(t+1)} - \mathbf{W}^{(t)} \rangle \right| + \widetilde{\Theta}(\eta^2 d). \qquad \text{(C.15)}$$

Additionally, $\|\nabla_{\mathbf{W}} F_j(\mathbf{W}^{(t)}, \mathbf{x}_i)\|_F \leq \widetilde{\Theta}(1)$ since $m = \widetilde{\Theta}(1)$, $\langle \mathbf{w}_{j,r}^{(t)}, y_i\mathbf{v} \rangle \leq \widetilde{\Theta}(1)$, $\langle \mathbf{w}_{j,r}^{(t)}, \boldsymbol{\xi}_i \rangle \leq \widetilde{\Theta}(1)$. So we have

$$|\Delta F_{j,i}| \leq \widetilde{\Theta}(\eta s \sigma_p + \eta \alpha + \eta + \eta^2 d) \leq \widetilde{\Theta}(\eta s \sigma_p + \eta^2 d). \qquad \text{(C.16)}$$

Substitute (C.15) and (C.16) into (C.12):

$$L_i(\mathbf{W}^{(t+1)}) - L_i(\mathbf{W}^{(t)}) \leq \langle \nabla_{\mathbf{W}} L_i(\mathbf{W}^{(t)}), \mathbf{W}^{(t+1)} - \mathbf{W}^{(t)} \rangle + \widetilde{\Theta}(\eta^2 d). \qquad \text{(C.17)}$$

For the full objective:

$$L(\mathbf{W}^{(t+1)}) - L(\mathbf{W}^{(t)}) = \frac{1}{n}\sum_{i=1}^{n}[L_i(\mathbf{W}^{(t+1)}) - L_i(\mathbf{W}^{(t)})] + \lambda(\|\mathbf{W}^{(t+1)}\|_F^2 - \|\mathbf{W}^{(t)}\|_F^2).$$

(C.18)

Since $\lambda\|\mathbf{W}\|_F^2$ is $2\lambda$-smooth and $\lambda = o(1)$, the regularization term contributes:

$$\lambda(\|\mathbf{W}^{(t+1)}\|_F^2 - \|\mathbf{W}^{(t)}\|_F^2) \le 2\lambda\langle\mathbf{W}^{(t)}, \mathbf{W}^{(t+1)} - \mathbf{W}^{(t)}\rangle + \lambda\widetilde{\Theta}(\eta^2 d),\qquad(\text{C.19})$$

where the quadratic term is absorbed into $\widetilde{\Theta}(\eta^2 d)$. Substitute (C.17) and (C.19) into (C.18), we have

$$L(\mathbf{W}^{(t+1)}) - L(\mathbf{W}^{(t)}) \le \langle\nabla L(\mathbf{W}^{(t)}), \mathbf{W}^{(t+1)} - \mathbf{W}^{(t)}\rangle + \widetilde{\Theta}(\eta^2 d).\qquad(\text{C.20})$$

Take expectation for the stochastic gradient of both side in (C.20),

$$\mathbb{E}\left[L(\mathbf{W}^{(t+1)}) - L(\mathbf{W}^{(t)})\right]$$

$$\le \mathbb{E}\left[\langle\nabla L(\mathbf{W}^{(t)}), \mathbf{W}^{(t+1)} - \mathbf{W}^{(t)}\rangle\right] + \widetilde{\Theta}(\eta^2 d)$$

$$\le -\eta\cdot\mathbb{E}\left[\sum_{j\in\{\pm1\}}\sum_{r\in[m]}\left\|g_{t,j,r}^{(t)}\right\|_1\right] + \widetilde{\Theta}(d\cdot\eta^2) + \widetilde{\Theta}(ns\cdot\eta^2 s\sigma_p) + \widetilde{\Theta}(\eta^2 d)$$

$$\le -\eta\cdot\sum_{j\in\{\pm1\}}\sum_{r\in[m]}\left\|\mathbb{E}\left[g_{t,j,r}^{(t)}\right]\right\|_1 + \widetilde{\Theta}(\eta^2 d)$$

$$= -\eta\|\nabla L(\mathbf{W}^{(t)})\|_1 + \widetilde{\Theta}(\eta^2 d),$$

where we use Lemma C.2 that the update aligns with the gradient's sign for large gradient and the fact that $ns^2\sigma_p = O(d)$ and Jensen's inequality. This completes the proof. $\qquad\square$

**Lemma C.9** (Generalization Performance of Stochastic Adam). *Suppose the same conditions hold as in Lemma C.8. We have the following results for $T = \frac{\text{poly}(n)}{\eta}$, with training dataset $\mathcal{S}$*

- *The training error is zero:* $\text{err}_{\mathcal{S}}(\mathbf{W}^{(T)}) = 0$.

- *The test error is near-zero:* $\text{err}_{\mathcal{D}}(\mathbf{W}^{(T)}) = o(1)$.

*Proof of Lemma C.9.* By Lemma C.7, we have

$$\langle\mathbf{w}_{j,r^*}^{(T)}, j\mathbf{v}\rangle = \widetilde{\Theta}(1), \quad \langle\mathbf{w}_{j,r}^{(T)}, \boldsymbol{\xi}_i\rangle = \widetilde{O}(\eta s\sigma_p + \alpha).$$

Recall $F_j(\mathbf{W}, \mathbf{x})$ in Definition 3.2, with $\eta s\sigma_p = o(1)$, $\alpha = o(1)$, we directly have

$$\text{err}_{\mathcal{S}}(\mathbf{W}^{(T)}) = \mathbb{E}_{(\mathbf{x},y)\sim\mathcal{S}}\mathbb{1}\left[F_y(\mathbf{W}^{(T)}, \mathbf{x}) \le F_{-y}(\mathbf{W}^{(T)}, \mathbf{x})\right] = 0,$$

since $F_{y_i}(\mathbf{W}^{(T)}, \mathbf{x}_i) = \widetilde{\Omega}(1)$, while $F_{-y_i}(\mathbf{W}^{(T)}, \mathbf{x}) \le \widetilde{\Theta}(\eta s\sigma_p + \alpha)$. Besides, for test data $(\mathbf{x}, y) \sim \mathcal{D}$ with $\mathbf{x} = [y\mathbf{v}^\top, \boldsymbol{\xi}^\top]^\top$, it is clear that with high probability $\langle\mathbf{w}_{y,r^*}^{(T)}, y\mathbf{v}\rangle = \widetilde{\Theta}(1)$ and $[\langle\mathbf{w}_{y,r}^{(T)}, \boldsymbol{\xi}\rangle]_+ \le \widetilde{\Theta}(\eta s\sigma_p + \alpha)$, then similar as training error, we have

$$F_y(\mathbf{W}^{(T)}, \mathbf{x}) \ge \sigma(\langle\mathbf{w}_{y,r^*}^{(T)}, y\mathbf{v}\rangle) = \widetilde{\Omega}(1),$$

while

$$F_{-y}(\mathbf{W}^*, \mathbf{x}) = \sum_{r=1}^{m}\left[\sigma(\langle\mathbf{w}_{-y,r}^*, y\mathbf{v}\rangle) + \sigma(\langle\mathbf{w}_{-y,r}^*, \boldsymbol{\xi}\rangle)\right] \le \widetilde{\Theta}(\eta s\sigma_p + \alpha).$$

Therefore, we have

$$\text{err}_{\mathcal{D}}(\mathbf{W}^{(T)}) = \mathbb{E}_{(\mathbf{x},y)\sim\mathcal{D}}\mathbb{1}\left[F_y(\mathbf{W}^{(T)}, \mathbf{x}) \le F_{-y}(\mathbf{W}^{(T)}, \mathbf{x})\right] = o(1).$$

This implies that mini-batch Adam can achieve nearly zero test error. This completes the proof. $\quad\square$

### C.1.3 Proof of Corollary 4.3

Corollary 4.3 follows directly from Lemma C.10.

**Lemma C.10.** *Suppose the same conditions hold as in Lemma C.3 and C.6, if $\lambda = \omega(\sigma_0^{q-2})$, then*

$$\langle \mathbf{w}_{j,r}^{(t)}, j\mathbf{v} \rangle \leq \widetilde{\Theta}(\sigma_0),$$
$$\langle \mathbf{w}_{j,r}^{(t)}, \boldsymbol{\xi}_i \rangle \leq \widetilde{\Theta}(s^{1/2}\sigma_p\sigma_0).$$

*Proof of Lemma C.10.* This corollary is an immediate consequence of Lemmas C.3 and C.6. In particular, Lemma A.3 guarantees that at $t = 0$

$$|\langle \mathbf{w}_{j,r}^{(0)}, \mathbf{v} \rangle| = \widetilde{\Theta}(\sigma_0), \quad |\langle \mathbf{w}_{j,r}^{(0)}, \boldsymbol{\xi}_i \rangle| = \widetilde{\Theta}(s^{1/2}\,\sigma_p\,\sigma_0), \quad \mathbf{w}_{j,r}^{(0)}[k] = \widetilde{\Theta}(\sigma_0).$$

Since $\lambda = \omega(\sigma_0^{q-2})$, Lemma A.7 implies that, at initialization, the weight decay regularization term overwhelmingly dominates the gradient:

$$\lambda \left| \mathbf{w}_{j,r}^{(0)}[1] \right| \gg \sigma'\big(\langle \mathbf{w}_{j,r}^{(0)}, y_i\mathbf{v} \rangle\big) + \alpha\,\sigma'\big(\langle \mathbf{w}_{j,r}^{(0)}, \boldsymbol{\xi}_i \rangle\big),$$
$$\lambda \left| \mathbf{w}_{y_i,r}^{(0)}[k] \right| \gg \sigma'\big(\langle \mathbf{w}_{y_i,r}^{(0)}, \boldsymbol{\xi}_i \rangle\big)\,\boldsymbol{\xi}_i[k].$$

Hence, by Lemma C.2, the updates remain in the regularization-dominated regime, and no coordinate ever grows beyond its $\widetilde{\Theta}(\sigma_0)$ scale throughout training. This completes the proof. $\qquad\square$

### C.2 Proof of Stochastic AdamW

**Lemma C.11.** *Consider the update of stochastic AdamW in (3.6). Let $\mathbf{W}^{(t)}$ be the weight at the $t$-th iteration. Suppose that $\langle \mathbf{w}_{j,r}^{(t)}, y_i\mathbf{v} \rangle, \langle \mathbf{w}_{j,r}^{(t)}, \boldsymbol{\xi}_i \rangle = \widetilde{\Theta}(1)$ for all $j \in \{\pm 1\}$, $r \in [m]$, $i \in [n]$ and $\beta_1^2 < \beta_2$. We have the approximate update rule for each coordinate weight as follows:*

- *For $k = 1$, we have either $|g_{t,j,r}^{(t)}[1]| \leq \widetilde{\Theta}(\eta)$ or*

$$\frac{\mathbf{m}_{j,r}^{(t)}[k]}{\sqrt{\mathbf{v}_{j,r}^{(t)}[k]} + \epsilon} = \mathrm{sgn}\big(g_{t,j,r}^{(t)}[k]\big) \cdot \Theta(1).$$

- *For every $k \in \mathcal{B}_i$, $i \in \mathcal{I}_{t-\tau}$, $\tau \in \mathcal{T}_k := \{\tau_0 + i \cdot \frac{n}{B} : i \in \{0\} \cup [\frac{\bar{\tau}}{n/B}], \tau_0 < \frac{n}{B}\}$, where $\tau_0$ represents the number of iterations away from the current iteration $t$, coordinate $k$ is affected by $\boldsymbol{\xi}_i$ sampled at the iteration $t - \tau_0$ since the moving average, and we define $\bar{\tau} = \Theta(\log(\lambda\eta)^{-1})$*

  - *If $\frac{n}{B} \leq \Theta(1)$, for any $\tau_0 < \frac{n}{B}$, we have either*

  $$\left| g_{t-\tau_0,j,r}^{(t)}[k] \right| \leq \widetilde{\Theta}\left( B^{-1}\eta s\sigma_p |\ell_{j,i}^{(t)}| \right)$$

  *or*

  $$\frac{\mathbf{m}_{j,r}^{(t)}[k]}{\sqrt{\mathbf{v}_{j,r}^{(t)}[k]} + \epsilon} = \mathrm{sgn}\left( g_{t-\tau_0,j,r}^{(t)}[k] \right) \cdot \Theta(1).$$

  - *If $\frac{n}{B} \geq \Theta(\log(\lambda\eta)^{-1}) = \widetilde{\Theta}(1)$, for $\tau_0 \leq \Theta(\log(\lambda^{-1}s\sigma_p))$ such that $\beta_1^{\tau_0} \geq \Theta(\frac{\lambda}{s\sigma_p})$, we have either*

  $$\left| g_{t-\tau_0,j,r}^{(t)}[k] \right| \leq \widetilde{\Theta}\left( B^{-1}\eta s\sigma_p |\ell_{j,i}^{(t)}| \right)$$

  *or*

  $$\frac{\mathbf{m}_{j,r}^{(t)}[k]}{\sqrt{\mathbf{v}_{j,r}^{(t)}[k]} + \epsilon} = \mathrm{sgn}\left( g_{t-\tau_0,j,r}^{(t)}[k] \right) \cdot \Theta(1).$$

  *For $\tau_0 \geq \Theta(\log(\lambda\eta)^{-1})$ such that $\beta_2^{\frac{\tau_0}{2}} \leq \Theta(\lambda^2\eta^2)$, we have*

  $$\frac{\mathbf{m}_{j,r}^{(t)}[k]}{\sqrt{\mathbf{v}_{j,r}^{(t)}[k]} + \epsilon} = \pm\widetilde{\Theta}(\lambda\eta) = o(1).$$

- *For the remaining coordinates $k \neq 1$ and $k \notin \mathcal{B}_i$, $i \in \mathcal{I}_{t-\tau_0}$, $\tau_0 \in \{0\} \cup [\bar{\tau}]$, where $\tau_0$ represents the number of iterations away from the current iteration $t$, coordinate $k$ is affected by $\boldsymbol{\xi}_i$ sampled at the iteration $t - \tau_0$ since the moving average, and we define $\bar{\tau} = \log(\lambda\eta)^{-1}$. Then we have*

$$\frac{\mathbf{m}_{j,r}^{(t)}[k]}{\sqrt{\mathbf{v}_{j,r}^{(t)}[k] + \epsilon}} = \pm\widetilde{O}(\lambda\eta) = o(1).$$

*Proof of Lemma C.11.* The proof is similar to Lemma C.2. We select $\bar{\tau} = \Theta(\log(\lambda\eta)^{-1})$ such that $\sum_{\tau=\bar{\tau}+1}^{t} \beta_1^\tau (1 - \beta_1) = O(\lambda^2\eta^2)$ and $\sum_{\tau=\bar{\tau}+1}^{t} \beta_1^\tau (1 - \beta_1) \cdot g_{t-\tau,j,r}^{(t-\tau)}[k] = \widetilde{O}(\lambda^2\eta^2)$.

$$\frac{\mathbf{m}_{j,r}^{(t)}[k]}{\sqrt{\mathbf{v}_{j,r}^{(t)}[k] + \epsilon}} = \frac{\sum_{\tau=0}^{\bar{\tau}} \beta_1^\tau (1 - \beta_1) \cdot g_{t-\tau,j,r}^{(t-\tau)}[k] \pm \widetilde{O}(\lambda^2\eta^2)}{\sqrt{\sum_{\tau=\bar{\tau}}^{\bar{\tau}} \beta_2^\tau (1 - \beta_2) \cdot g_{t-\tau,j,r}^{(t-\tau)}[k]^2 + \epsilon \pm \widetilde{O}(\lambda^2\eta^2)}}.$$

Recall the gradient of stochastic AdamW given in Lemma A.8, we want to approximate $g_{t-\tau,j,r}^{(t-\tau)}[k]$ to $g_{t-\tau,j,r}^{(t)}[k]$, such that we can use the current weight to approximate sign update. By Lemma C.1, the upper bound of the normalized moving average of each coordinate in one step is $\Theta(\eta)$ since $\lambda = \widetilde{O}(1)$. Then for $\tau \in [t - \bar{\tau}, t]$, we have

$$\left| \langle \mathbf{w}_{j,r}^{(t)}, y_i\mathbf{v} \rangle - \langle \mathbf{w}_{j,r}^{(\tau)}, y_i\mathbf{v} \rangle \right|$$

$$\leq \sum_{k=\tau}^{t-1} \left| \langle \mathbf{w}_{j,r}^{(k+1)}, y_i\mathbf{v} \rangle - \langle \mathbf{w}_{j,r}^{(k)}, y_i\mathbf{v} \rangle \right|$$

$$\leq \sum_{k=\tau}^{t-1} \eta\lambda \left| \langle \mathbf{w}_{j,r}^{(k)}, y_i\mathbf{v} \rangle \right| + \sum_{k=\tau}^{t-1} \eta \left| \left\langle \frac{\mathbf{m}_{j,r}^{(k)}}{\sqrt{\mathbf{v}_{j,r}^{(k)} + \epsilon}}, y_i\mathbf{v} \right\rangle \right|$$

$$\leq \widetilde{\Theta}(\eta\bar{\tau}). \tag{C.21}$$

The last inequality we use the fact that $\langle \mathbf{w}_{j,r}^{(k)}, y_i\mathbf{v} \rangle = \widetilde{\Theta}(1)$, $\lambda = \widetilde{O}(1)$ and Lemma C.1. Similarly, we have

$$\left| \langle \mathbf{w}_{j,r}^{(t)}, \boldsymbol{\xi}_i \rangle - \langle \mathbf{w}_{j,r}^{(\tau)}, \boldsymbol{\xi}_i \rangle \right| \leq \Theta(\eta\bar{\tau}s\sigma_p). \tag{C.22}$$

Then recall the predict function

$$F_j(\mathbf{W}^{(t)}, \mathbf{x}_i) = \sum_{r=1}^{m} \left[ \sigma\left( \langle \mathbf{w}_{j,r}^{(t)}, y_i\mathbf{v} \rangle \right) + \sigma\left( \langle \mathbf{w}_{j,r}^{(t)}, \boldsymbol{\xi}_i \rangle \right) \right].$$

We have

$$\left| F_j(\mathbf{W}^{(t)}, \mathbf{x}_i) - F_j(\mathbf{W}^{(\tau)}, \mathbf{x}_i) \right|$$

$$\leq \sum_{r=1}^{m} \left| \sigma\left( \langle \mathbf{w}_{j,r}^{(t)}, y_i\mathbf{v} \rangle \right) - \sigma\left( \langle \mathbf{w}_{j,r}^{(\tau)}, y_i\mathbf{v} \rangle \right) \right| + \sum_{r=1}^{m} \left| \sigma\left( \langle \mathbf{w}_{j,r}^{(t)}, \boldsymbol{\xi}_i \rangle \right) - \sigma\left( \langle \mathbf{w}_{j,r}^{(\tau)}, \boldsymbol{\xi}_i \rangle \right) \right|$$

$$\leq \sum_{r=1}^{m} \widetilde{\Theta}(1) \cdot \left| \langle \mathbf{w}_{j,r}^{(t)}, y_i\mathbf{v} \rangle - \langle \mathbf{w}_{j,r}^{(\tau)}, y_i\mathbf{v} \rangle \right| + \sum_{r=1}^{m} \widetilde{\Theta}(1) \cdot \left| \langle \mathbf{w}_{j,r}^{(t)}, \boldsymbol{\xi}_i \rangle - \langle \mathbf{w}_{j,r}^{(\tau)}, \boldsymbol{\xi}_i \rangle \right|$$

$$\leq \widetilde{\Theta}(m\eta\bar{\tau}s\sigma_p) + \widetilde{\Theta}(m\eta\bar{\tau})$$

$$= \widetilde{\Theta}(\eta\bar{\tau}s\sigma_p), \tag{C.23}$$

where the second inequality we use the convexity of $\sigma(\cdot)$ and the facts that $|\langle \mathbf{w}_{j,r}^{(t)}, y_i\mathbf{v} \rangle| = \widetilde{\Theta}(1)$ and $|\langle \mathbf{w}_{j,r}^{(t)}, \boldsymbol{\xi}_i \rangle| = \widetilde{\Theta}(1)$, the last inequality we use $m = \widetilde{\Theta}(1)$ and $s\sigma_p = \omega(1)$.

Then we can approximate $\ell_{j,r}^{(\tau)}$ to $\ell_{j,r}^{(\tau)}$ in the gradient A.8.

$$\ell_{j,i}^{(\tau)} = \frac{e^{F_{-j}(\mathbf{W}^{(\tau)}, \mathbf{x}_i)}}{\sum_{k \in \{-1,1\}} e^{F_k(\mathbf{W}^{(\tau)}, \mathbf{x}_i)}}$$

$$= \frac{e^{F_{-j}(\mathbf{W}^{(t)}, \mathbf{x}_i) \pm \widetilde{\Theta}(\eta \bar{\tau} s \sigma_p)}}{e^{F_j(\mathbf{W}^{(t)}, \mathbf{x}_i) \pm \widetilde{\Theta}(\eta \bar{\tau} s \sigma_p)} + e^{F_{-j}(\mathbf{W}^{(t)}, \mathbf{x}_i) \pm \widetilde{\Theta}(\eta \bar{\tau} s \sigma_p)}}$$

$$= \mathrm{sgn}(\ell_{j,i}^{(t)}) \cdot \Theta(|\ell_{j,i}^{(t)}|), \qquad\qquad (y_i = j)$$

$$\ell_{j,i}^{(\tau)} = \frac{-e^{F_j(\mathbf{W}^{(\tau)}, \mathbf{x}_i)}}{\sum_{k \in \{-1,1\}} e^{F_k(\mathbf{W}^{(\tau)}, \mathbf{x}_i)}}$$

$$= \frac{-e^{F_j(\mathbf{W}^{(t)}, \mathbf{x}_i) \pm \widetilde{\Theta}(\eta \bar{\tau} s \sigma_p)}}{e^{F_j(\mathbf{W}^{(t)}, \mathbf{x}_i) \pm \widetilde{\Theta}(\eta \bar{\tau} s \sigma_p)} + e^{F_{-j}(\mathbf{W}^{(t)}, \mathbf{x}_i) \pm \widetilde{\Theta}(\eta \bar{\tau} s \sigma_p)}}$$

$$= \mathrm{sgn}(\ell_{j,i}^{(t)}) \cdot \Theta(|\ell_{j,i}^{(t)}|), \qquad\qquad (y_i \neq j)$$

where we use the fact that $\widetilde{\Theta}(\eta \bar{\tau} s \sigma_p) = o(1)$ and (C.23). So we have

$$\ell_{j,i}^{(\tau)} = \mathrm{sgn}(\ell_{j,i}^{(t)}) \cdot \Theta(|\ell_{j,i}^{(t)}|),$$

for all $\tau \in [t - \bar{\tau}, t]$. Further, by (C.21), (C.22) and the facts that $|\langle \mathbf{w}_{j,r}^{(t)}, y_i \mathbf{v} \rangle| = \widetilde{O}(1)$ and $|\langle \mathbf{w}_{j,r}^{(t)}, \boldsymbol{\xi}_i \rangle| = \widetilde{O}(1)$, recall $\sigma(x) = \max(0, x)^q$, we have

$$\ell_{j,i}^{(\tau)} \sigma'(\langle \mathbf{w}_{j,r}^{(\tau)}, y_i \mathbf{v} \rangle)$$
$$\leq \ell_{j,i}^{(\tau)} \sigma'(\langle \mathbf{w}_{j,r}^{(t)}, y_i \mathbf{v} \rangle) + |\ell_{j,i}^{(\tau)}| \cdot \widetilde{\Theta}(\eta \bar{\tau})$$
$$= \mathrm{sgn}(\ell_{j,i}^{(t)}) \cdot \Theta(|\ell_{j,i}^{(t)}|) \cdot \sigma'(\langle \mathbf{w}_{j,r}^{(t)}, y_i \mathbf{v} \rangle) + \Theta(|\ell_{j,i}^{(t)}|) \cdot \widetilde{\Theta}(\eta \bar{\tau}),$$
$$\ell_{j,i}^{(\tau)} \sigma'(\langle \mathbf{w}_{j,r}^{(\tau)}, y_i \mathbf{v} \rangle)$$
$$\geq \ell_{j,i}^{(\tau)} \sigma'(\langle \mathbf{w}_{j,r}^{(t)}, y_i \mathbf{v} \rangle) - |\ell_{j,i}^{(\tau)}| \cdot \widetilde{\Theta}(\eta \bar{\tau})$$
$$= \mathrm{sgn}(\ell_{j,i}^{(t)}) \cdot \Theta(|\ell_{j,i}^{(t)}|) \cdot \sigma'(\langle \mathbf{w}_{j,r}^{(t)}, y_i \mathbf{v} \rangle) - \Theta(|\ell_{j,i}^{(t)}|) \cdot \widetilde{\Theta}(\eta \bar{\tau}).$$

So we conclude that

$$\ell_{j,i}^{(\tau)} \sigma'(\langle \mathbf{w}_{j,r}^{(\tau)}, y_i \mathbf{v} \rangle) = \mathrm{sgn}(\ell_{j,i}^{(t)}) \cdot \Theta(|\ell_{j,i}^{(t)}|) \cdot \sigma'(\langle \mathbf{w}_{j,r}^{(t)}, y_i \mathbf{v} \rangle) \pm \Theta(|\ell_{j,i}^{(t)}|) \cdot \widetilde{\Theta}(\eta \bar{\tau}). \qquad (\text{C.24})$$

Similarly, we have

$$\ell_{j,i}^{(\tau)} \sigma'(\langle \mathbf{w}_{j,r}^{(\tau)}, \boldsymbol{\xi}_i \rangle) = \mathrm{sgn}(\ell_{j,i}^{(t)}) \cdot \Theta(|\ell_{j,i}^{(t)}|) \cdot \sigma'(\langle \mathbf{w}_{j,r}^{(t)}, \boldsymbol{\xi}_i \rangle) \pm \Theta(|\ell_{j,i}^{(t)}|) \cdot \widetilde{\Theta}(\eta \bar{\tau} s \sigma_p). \qquad (\text{C.25})$$

Now, we have all the tools we need to approximate $g_{t-\tau,j,r}^{(t-\tau)}[k]$ to $g_{t-\tau,j,r}^{(t)}[k]$. Recall Lemma A.8, substitute (C.24) and (C.25) into $g_{t-\tau,j,r}^{(t-\tau)}[k]$, we have

- For $k = 1$,

$$g_{t-\tau,j,r}^{(t-\tau)}[1] = \Theta\left(g_{t-\tau,j,r}^{(t)}[1]\right) \pm \Theta\left(\frac{1}{B} \sum_{i \in \mathcal{I}_{t-\tau}} |\ell_{j,i}^{(t)}|\right) \cdot \widetilde{O}(\eta \bar{\tau}). \qquad (\text{C.26})$$

- For all $k \in \mathcal{B}_i$, $i \in \mathcal{I}_{t-\tau}$,

$$g_{t-\tau,j,r}^{(t-\tau)}[k] = \Theta\left(g_{t-\tau,j,r}^{(t)}[k]\right) \pm \Theta\left(\frac{|\ell_{j,i}^{(t)}|}{B}\right) \cdot \widetilde{O}(\eta \bar{\tau} s \sigma_p). \qquad (\text{C.27})$$

- For $k \neq 1$ and $k \notin \mathcal{B}_i$, $i \in \mathcal{I}_{t-\tau}$,

$$g_{t-\tau,j,r}^{(t-\tau)}[k] = g_{t-\tau,j,r}^{(t)}[k] = 0. \qquad (\text{C.28})$$

Plugging (C.26), (C.27) and (C.28) into (C.1), with facts that $\bar{\tau} = \widetilde{\Theta}(1)$, $\lambda = o(1)$, $|\langle \mathbf{w}_{j,r}^{(t)}, \boldsymbol{\xi}_i \rangle| = \widetilde{\Theta}(1)$, $|\langle \mathbf{w}_{j,r}^{(t)}, y_i \mathbf{v} \rangle| = \widetilde{\Theta}(1)$, $|\ell_{j,i}^{(t)}| = \Theta(1)$, $\epsilon = \Theta(\lambda \eta)$ and Lemma A.6, we have

- For $k = 1$,

$$\frac{\mathbf{m}_{j,r}^{(t)}[1]}{\sqrt{\mathbf{v}_{j,r}^{(t)}[1] + \epsilon}}$$

$$= \frac{\sum_{\tau=0}^{\bar{\tau}} \beta_1^{\tau}(1 - \beta_1) \cdot g_{t-\tau,j,r}^{(t-\tau)}[1] \pm \widetilde{O}(\lambda^2\eta^2)}{\sqrt{\sum_{\tau=0}^{\bar{\tau}} \beta_2^{\tau}(1 - \beta_2) \cdot g_{t-\tau,j,r}^{(t-\tau)}[1]^2 + \epsilon \pm \widetilde{O}(\lambda^2\eta^2)}}$$

$$= \frac{\sum_{\tau=0}^{\bar{\tau}} \beta_1^{\tau}(1 - \beta_1) \cdot \left( \Theta\left( g_{t-\tau,j,r}^{(t)}[1] \right) \pm \Theta\left( \frac{1}{B} \sum_{i \in \mathcal{I}_{t-\tau}} |\ell_{j,i}^{(t)}| \right) \cdot \widetilde{O}(\eta\bar{\tau}) \right) \pm \widetilde{O}(\lambda^2\eta^2)}{\sqrt{\sum_{\tau=0}^{\bar{\tau}} \beta_2^{\tau}(1 - \beta_2) \cdot \left( \Theta\left( g_{t-\tau,j,r}^{(t)}[1] \right) \pm \Theta\left( \frac{1}{B} \sum_{i \in \mathcal{I}_{t-\tau}} |\ell_{j,i}^{(t)}| \right) \cdot \widetilde{O}(\eta\bar{\tau}) \right)^2 + \epsilon \pm \widetilde{O}(\lambda^2\eta^2)}}$$

$$= \frac{\sum_{\tau=0}^{\bar{\tau}} \beta_1^{\tau}(1 - \beta_1) \cdot \left( \Theta\left( g_{t-\tau,j,r}^{(t)}[1] \right) \pm \widetilde{\Theta}(\eta\bar{\tau}) \right) \pm \widetilde{O}(\lambda^2\eta^2)}{\sqrt{\sum_{\tau=0}^{\bar{\tau}} \beta_2^{\tau}(1 - \beta_2) \cdot \left( \Theta\left( g_{t-\tau,j,r}^{(t)}[1] \right) \pm \widetilde{\Theta}(\eta\bar{\tau}) \right)^2 + \epsilon \pm \widetilde{O}(\lambda^2\eta^2)}}$$

$$= \frac{\Theta\left( g_{t,j,r}^{(t)}[1] \right) \pm \widetilde{\Theta}(\eta\bar{\tau}) \pm \widetilde{O}(\lambda^2\eta^2)}{\sqrt{\left( \Theta\left( g_{t,j,r}^{(t)}[1] \right) \pm \widetilde{\Theta}(\eta\bar{\tau}) \right)^2 + \epsilon \pm \widetilde{O}(\lambda^2\eta^2)}}$$

$$= \frac{\Theta\left( g_{t,j,r}^{(t)}[1] \right) \pm \widetilde{\Theta}(\eta)}{\Theta\left( |g_{t,j,r}^{(t)}[1]| \right) \pm \widetilde{\Theta}(\eta)}.$$

- For $k \in \mathcal{B}_i$, $i \in \mathcal{I}_{t-\tau_0}$, $\tau_0 \in \{0\} \cup [\bar{\tau}]$, where $\tau_0$ represents the number of iterations away from the current iteration $t$, coordinate $k$ is affected by $\boldsymbol{\xi}_i$ sampled at the iteration $t - \tau_0$ since the moving average. We note that if the number of iteration in one epoch $\frac{n}{B}$ is less than $\bar{\tau}$, the moving average will use some sample $\mathbf{x}$ multiply times. We denote $\mathcal{T}_k := \{\tau_0 + i \cdot \frac{n}{B} : i \in [\frac{\bar{\tau}}{n/B} - 1]\}$ as the timestamp set involved using noise $\boldsymbol{\xi}_i$ (i.e., $i \in \mathcal{I}_{t-\tau}$ for any $\tau \in \mathcal{T}_k$), and $k \in \mathcal{B}_i$, $\tau_0 \leq \frac{n}{B}$.

If $\frac{n}{B} > \bar{\tau}$, in this case we have $\mathcal{T}_k := \{\tau_0\}$ for any $k \in \mathcal{B}_i$, and $\boldsymbol{\xi}_i$ was used in iteration $t - \tau_0$. Then we have

$$\frac{\mathbf{m}_{j,r}^{(t)}[k]}{\sqrt{\mathbf{v}_{j,r}^{(t)}[k] + \epsilon}}$$

$$= \frac{\sum_{\tau=0}^{\bar{\tau}} \beta_1^{\tau}(1 - \beta_1) \cdot g_{t-\tau,j,r}^{(t-\tau)}[k] \pm \widetilde{O}(\lambda^2\eta^2)}{\sqrt{\sum_{\tau=0}^{\bar{\tau}} \beta_2^{\tau}(1 - \beta_2) \cdot g_{t-\tau,j,r}^{(t-\tau)}[k]^2 + \epsilon \pm \widetilde{O}(\lambda^2\eta^2)}}$$

$$= \frac{\sum_{\tau \in \mathcal{T}_k} \beta_1^{\tau}(1 - \beta_1) \cdot g_{t-\tau,j,r}^{(t-\tau)}[k] \pm \widetilde{O}(\lambda^2\eta^2)}{\sqrt{\sum_{\tau \in \mathcal{T}_k} \beta_2^{\tau}(1 - \beta_2) \cdot g_{t-\tau,j,r}^{(t-\tau)}[k]^2 + \epsilon \pm \widetilde{O}(\lambda^2\eta^2)}}$$

$$= \frac{\beta_1^{\tau_0} \cdot \Theta(g_{t-\tau_0,j,r}^{(t-\tau_0)}[k]) \pm \widetilde{O}(\lambda^2\eta^2)}{\sqrt{\beta_2^{\tau_0} \cdot \Theta(g_{t-\tau_0,j,r}^{(t-\tau_0)}[k]^2) + \epsilon \pm \widetilde{O}(\lambda^2\eta^2)}}$$

$$= \frac{\beta_1^{\tau_0} \left( \Theta\left( g_{t-\tau_0,j,r}^{(t)}[k] \right) \pm \widetilde{\Theta}\left( \frac{\eta s \sigma_p |\ell_{j,i}^{(t)}|}{B} \right) \right) \pm \widetilde{O}(\lambda^2\eta^2)}{\beta_2^{\frac{\tau_0}{2}} \left( \Theta\left( |g_{t-\tau_0,j,r}^{(t)}[k]| \right) \pm \widetilde{\Theta}\left( \frac{\eta s \sigma_p |\ell_{j,i}^{(t)}|}{B} \right) \right) + \epsilon \pm \widetilde{O}(\lambda^2\eta^2)}.$$

Now we handle $\beta_1^{\tau_0}$ and $\beta_2^{\frac{\tau_0}{2}}$ with more care. First we have $\beta_1^{\tau_0} < \beta_2^{\frac{\tau_0}{2}}$ and $|g_{t-\tau_0,j,r}^{(t)}[k]| = \widetilde{O}(1)$.

Then if $\frac{n}{B} \leq \Theta(1)$, then $\beta_1^{\tau_0} = \Theta(1)$ and $\beta_2^{\frac{\tau_0}{2}} = \Theta(1)$ since $\tau_0 < \frac{n}{B}$. In this case, we have

$$\frac{\mathbf{m}_{j,r}^{(t)}[k]}{\sqrt{\mathbf{v}_{j,r}^{(t)}[k] + \epsilon}} = \frac{\Theta\left(g_{t-\tau_0,j,r}^{(t)}[k]\right) \pm \widetilde{\Theta}\left(\frac{\eta s \sigma_p |\ell_{j,i}^{(t)}|}{B}\right)}{\Theta\left(|g_{t-\tau_0,j,r}^{(t)}[k]|\right) \pm \widetilde{\Theta}\left(\frac{\eta s \sigma_p |\ell_{j,i}^{(t)}|}{B}\right)}.$$

If $\frac{n}{B} \geq \Theta(\log \epsilon^{-1}) = \Theta(\log(\lambda\eta)^{-1}) = \widetilde{\Theta}(1)$ such that $\beta_2^{\frac{n}{2B}} \leq \Theta(\lambda^2\eta^2)$, then for $\tau_0 = O(\log(\lambda^{-1} s \sigma_p))$ such that $\beta_1^{\tau_0} \geq \Theta(\frac{\epsilon}{\eta s \sigma_p}) = \Theta(\frac{\lambda}{s\sigma_p})$, we have

$$\frac{\mathbf{m}_{j,r}^{(t)}[k]}{\sqrt{\mathbf{v}_{j,r}^{(t)}[k] + \epsilon}} = \frac{\Theta\left(g_{t-\tau_0,j,r}^{(t)}[k]\right) \pm \widetilde{\Theta}\left(\frac{\eta s \sigma_p |\ell_{j,i}^{(t)}|}{B}\right)}{\Theta\left(|g_{t-\tau_0,j,r}^{(t)}[k]|\right) \pm \widetilde{\Theta}\left(\frac{\eta s \sigma_p |\ell_{j,i}^{(t)}|}{B}\right)}.$$

For $\tau_0 < \frac{n}{B}$, $\tau_0 \geq \Theta(\log(\lambda\eta)^{-1})$ such that $\beta_2^{\frac{\tau_0}{2}} = O(\lambda^2\eta^2)$, we have

$$\frac{\mathbf{m}_{j,r}^{(t)}[k]}{\sqrt{\mathbf{v}_{j,r}^{(t)}[k] + \epsilon}} = \frac{\pm\widetilde{O}(\lambda^2\eta^2)}{\epsilon \pm \widetilde{O}(\lambda^2\eta^2)} = \pm\widetilde{O}(\lambda\eta) = o(1),$$

since $\epsilon = \Theta(\lambda\eta)$.

- For $k \neq 1$ and $k \notin \mathcal{B}_i, i \in \mathcal{I}_{t-\tau}, \tau \in [0, \bar{\tau}]$, recall $g_{t-\tau,j,r}^{(t)}[k] = 0$, we have

$$\frac{\mathbf{m}_{j,r}^{(t)}[k]}{\sqrt{\mathbf{v}_{j,r}^{(t)}[k] + \epsilon}}$$

$$= \frac{\sum_{\tau=0}^{\bar{\tau}} \beta_1^\tau (1 - \beta_1) \cdot g_{t-\tau,j,r}^{(t-\tau)}[k] \pm \widetilde{O}(\lambda^2\eta^2)}{\sqrt{\sum_{\tau=0}^{\bar{\tau}} \beta_2^\tau (1 - \beta_2) \cdot g_{t-\tau,j,r}^{(t-\tau)}[k]^2 + \epsilon \pm \widetilde{O}(\lambda^2\eta^2)}}$$

$$= \frac{\pm\widetilde{O}(\lambda^2\eta^2)}{\epsilon \pm \widetilde{O}(\lambda^2\eta^2)}$$

$$= \pm\widetilde{O}(\lambda\eta)$$

$$= o(1).$$

This completes the proof. $\qquad\square$

### C.2.1 Proof of Theorem 4.4

**Lemma C.12** (Stage I, pattern learning). *Given the training dataset $\mathcal{S}$, if $\frac{n}{B} = \Theta(1)$ or $\frac{n}{B} = o(s\sigma_p)$, $\eta = 1/\mathrm{poly}(d)$, $\lambda = \widetilde{\Omega}(\frac{B^2}{n} \wedge 1)$ and $\lambda = \widetilde{O}(1)$, then for any $t \leq T_0$ with $T_0 = \widetilde{O}(\frac{1}{\eta s \sigma_p})$,*

$$\langle \mathbf{w}_{j,r}^{((t+1)\cdot\frac{n}{B})}, j \cdot \mathbf{v} \rangle \leq \langle \mathbf{w}_{j,r}^{(t\cdot\frac{n}{B})}, j \cdot \mathbf{v} \rangle + \Theta(\eta \cdot \frac{n}{B})$$

$$\langle \mathbf{w}_{y_i,r}^{((t+1)\cdot\frac{n}{B})}, \boldsymbol{\xi}_i \rangle = \langle \mathbf{w}_{y_i,r}^{(t\cdot\frac{n}{B})}, \boldsymbol{\xi}_i \rangle + \widetilde{\Theta}(\eta s \sigma_p).$$

*Proof of Lemma C.12.* We prove this Lemma by induction. By Lemma C.1,

$$\langle \mathbf{w}_{j,r}^{(\frac{n}{B})}, j \cdot \mathbf{v} \rangle = (1 - \lambda\eta)\langle \mathbf{w}_{j,r}^{(\frac{n}{B}-1)}, j \cdot \mathbf{v} \rangle - \eta \left\langle \frac{\mathbf{m}_{j,r}^{(\frac{n}{B}-1)}}{\sqrt{\mathbf{v}_{j,r}^{(\frac{n}{B}-1)} + \epsilon}}, j\mathbf{v} \right\rangle$$

$$\leq (1 - \lambda\eta)\langle \mathbf{w}_{j,r}^{(\frac{n}{B}-1)}, j \cdot \mathbf{v} \rangle + \Theta(\eta)$$

$$\leq (1-\lambda\eta)^{\frac{n}{B}}\langle \mathbf{w}_{j,r}^{(0)}, j\cdot\mathbf{v}\rangle + \Theta(\eta)\cdot\sum_{k=0}^{\frac{n}{B}-1}(1-\lambda\eta)^k$$

$$= \langle \mathbf{w}_{j,r}^{(0)}, j\cdot\mathbf{v}\rangle - \Theta(\lambda\eta\cdot\frac{n}{B})\cdot\langle\mathbf{w}_{j,r}^{(0)}, j\cdot\mathbf{v}\rangle + \Theta(\eta\cdot\frac{n}{B})$$

$$= \langle \mathbf{w}_{j,r}^{(0)}, j\cdot\mathbf{v}\rangle + \Theta(\eta\cdot\frac{n}{B}),$$

where the second equality we use Taylor expansion and $\lambda\eta\cdot\frac{n}{B} = o(1)$, and the last equality we have $\langle \mathbf{w}_{j,r}^{(0)}, j\cdot\mathbf{v}\rangle = \widetilde{\Theta}(\sigma_0)$ by Lemma A.3. Now suppose the inequality holds for $t = 0,\ldots,t_0$ with $t_0 \leq T_0$. We have

$$\langle \mathbf{w}_{j,r}^{\left(t_0\cdot\frac{n}{B}\right)}, j\cdot\mathbf{v}\rangle \leq \langle \mathbf{w}_{j,r}^{(0)}, j\cdot\mathbf{v}\rangle + \Theta(\eta\cdot\frac{n}{B}\cdot t_0) \leq \widetilde{\Theta}(\sigma_0 + \frac{n}{Bs\sigma_p}) = \widetilde{O}(1),$$

since $\frac{n}{B} = o(s\sigma_p)$. For $t = t_0 + 1$,

$$\langle \mathbf{w}_{j,r}^{\left((t_0+1)\cdot\frac{n}{B}\right)}, j\cdot\mathbf{v}\rangle = (1-\lambda\eta)\langle\mathbf{w}_{j,r}^{\left((t_0+1)\cdot\frac{n}{B}-1\right)}, j\cdot\mathbf{v}\rangle - \eta\left\langle\frac{\mathbf{m}_{j,r}^{\left((t_0+1)\cdot\frac{n}{B}-1\right)}}{\sqrt{\mathbf{v}_{j,r}^{\left((t_0+1)\cdot\frac{n}{B}-1\right)}}+\epsilon}, j\mathbf{v}\right\rangle$$

$$\leq (1-\lambda\eta)\langle\mathbf{w}_{j,r}^{\left((t_0+1)\cdot\frac{n}{B}-1\right)}, j\cdot\mathbf{v}\rangle + \Theta(\eta)$$

$$\leq (1-\lambda\eta)^{\frac{n}{B}}\langle\mathbf{w}_{j,r}^{\left(t_0\cdot\frac{n}{B}\right)}, j\cdot\mathbf{v}\rangle + \Theta(\eta)\cdot\sum_{k=0}^{\frac{n}{B}-1}(1-\lambda\eta)^k$$

$$= \langle\mathbf{w}_{j,r}^{\left(t_0\cdot\frac{n}{B}\right)}, j\cdot\mathbf{v}\rangle - \Theta(\lambda\eta\cdot\frac{n}{B})\cdot\langle\mathbf{w}_{j,r}^{\left(t_0\cdot\frac{n}{B}\right)}, j\cdot\mathbf{v}\rangle + \Theta(\eta\cdot\frac{n}{B})$$

$$\leq \langle\mathbf{w}_{j,r}^{\left(t_0\cdot\frac{n}{B}\right)}, j\cdot\mathbf{v}\rangle + \Theta(\eta\cdot\frac{n}{B}).$$

Hence, we have $\langle\mathbf{w}_{j,r}^{(t)}, j\cdot\mathbf{v}\rangle = \widetilde{O}(1)$. Then, we prove $\langle\mathbf{w}_{y_i,r}^{\left((t+1)\cdot\frac{n}{B}\right)}, \boldsymbol{\xi}_i\rangle = \langle\mathbf{w}_{y_i,r}^{\left(t\cdot\frac{n}{B}\right)}, \boldsymbol{\xi}_i\rangle + \widetilde{\Theta}(\eta s\sigma_p)$ by induction. By Lemma A.3, we have

$$|\langle\mathbf{w}_{j,r}^{(0)}, \boldsymbol{\xi}_i\rangle| = \widetilde{\Theta}(s^{1/2}\sigma_p\sigma_0), \quad \mathbf{w}_{j,r}^{(0)}[k] = \widetilde{\Theta}(\sigma_0),$$

which imply that $|\ell_{j,i}^{(0)}| = \Theta(1)$. Assume that sample $(\mathbf{x}_i, y_i)$ is in batch $\mathcal{I}_\tau$ in the first epoch. Then we have

$$\langle\mathbf{w}_{y_i,r}^{(\tau)}, \boldsymbol{\xi}_i\rangle \geq (1-\lambda\eta)\langle\mathbf{w}_{y_i,r}^{(\tau-1)}, \boldsymbol{\xi}_i\rangle - \Theta(\eta\alpha)$$

$$\geq \langle\mathbf{w}_{y_i,r}^{(0)}, \boldsymbol{\xi}_i\rangle - \widetilde{\Theta}(\lambda\eta\sigma_0 + \eta\alpha)$$

$$= \widetilde{\Theta}(\sigma_0),$$

since $\lambda\eta = o(1)$, $\eta = o(\sigma_0)$, $\alpha = o(1)$ and $s^{1/2}\sigma_p = \widetilde{O}(1)$. Additionally, we have $\eta s = o(\sigma_0^{q-1})$ and $|\boldsymbol{\xi}_i[k]| \geq \widetilde{\Theta}(\sigma_p)$ with high probability. Then $|B^{-1}\sigma_0^{q-1}\boldsymbol{\xi}_i[k]| \geq \widetilde{\Theta}(\eta B^{-1}s\sigma_p|\ell_{j,i}^{(0)}|)$ for $i \in \mathcal{I}_\tau$. Therefore, by Lemma C.11 and A.6, we have

$$\text{sgn}\left(-\frac{1}{B}\ell_{y_i,i}^{(0)}\sigma'(\langle\mathbf{w}_{y_i,r}^{(0)}, \boldsymbol{\xi}_i\rangle)\boldsymbol{\xi}_i[k]\right)$$

$$= -\text{sgn}\left(\ell_{y_i,i}^{(0)}\sigma'(\langle\mathbf{w}_{y_i,r}^{(0)}, \boldsymbol{\xi}_i\rangle)\boldsymbol{\xi}_i[k]\right)$$

$$= -\text{sgn}(\boldsymbol{\xi}_i[k]). \tag{C.29}$$

Then, by Lemma C.11 we have the following update according to (B.6), (C.29) and Lemma C.1.

$$\langle\mathbf{w}_{y_i,r}^{(\tau+1)}, \boldsymbol{\xi}_i\rangle$$

$$= (1-\lambda\eta)\langle\mathbf{w}_{y_i,r}^{(\tau)}, \boldsymbol{\xi}_i\rangle - \eta\cdot\left\langle\frac{\mathbf{m}_{y_i,r}^{(\tau)}}{\sqrt{\mathbf{v}_{y_i,r}^{(\tau)}}+\epsilon}, \boldsymbol{\xi}_i\right\rangle$$

$$\geq \langle \mathbf{w}_{y_i,r}^{(0)}, \boldsymbol{\xi}_i \rangle + \Theta(\eta) \cdot \sum_{k \in \mathcal{B}_i} \langle \mathrm{sgn}(\boldsymbol{\xi}_i[k]), \boldsymbol{\xi}_i \rangle - \widetilde{O}(\lambda \eta \sigma_0) - O(\eta \alpha) - O(\eta s \sigma_p)$$

$$= \langle \mathbf{w}_{y_i,r}^{(0)}, \boldsymbol{\xi}_i \rangle + \widetilde{\Theta}(\eta s \sigma_p).$$

At the end of the first epoch, we have

$$\langle \mathbf{w}_{y_i,r}^{\left(\frac{n}{B}\right)}, \boldsymbol{\xi}_i \rangle$$
$$\geq (1 - \lambda\eta)\langle \mathbf{w}_{y_i,r}^{\left(\frac{n}{B}-1\right)}, \boldsymbol{\xi}_i \rangle - O(\eta\alpha)$$
$$\geq \langle \mathbf{w}_{y_i,r}^{(\tau+1)}, \boldsymbol{\xi}_i \rangle - \widetilde{O}(\lambda\eta^2 s \sigma_p) - O(\eta\alpha)$$
$$\geq \langle \mathbf{w}_{y_i,r}^{(0)}, \boldsymbol{\xi}_i \rangle + \widetilde{\Theta}(\eta s \sigma_p).$$

This completes the base case for $t = 1$. For general $t \leq t_0$ with $t_0 \leq T_0$, assuming $\langle \mathbf{w}_{y_i,r}^{\left(t \cdot \frac{n}{B}\right)}, \boldsymbol{\xi}_i \rangle = \langle \mathbf{w}_{y_i,r}^{\left((t-1)\cdot \frac{n}{B}\right)}, \boldsymbol{\xi}_i \rangle + \widetilde{\Theta}(\eta s \sigma_p)$. Then we have

$$\langle \mathbf{w}_{y_i,r}^{\left(t \cdot \frac{n}{B}\right)}, \boldsymbol{\xi}_i \rangle = \langle \mathbf{w}_{y_i,r}^{\left((t-1)\cdot \frac{n}{B}\right)}, \boldsymbol{\xi}_i \rangle + \widetilde{\Theta}(\eta s \sigma_p)$$
$$= \langle \mathbf{w}_{y_i,r}^{(0)}, \boldsymbol{\xi}_i \rangle + \widetilde{\Theta}(t \eta s \sigma_p)$$
$$= \widetilde{\Theta}(s^{1/2}\sigma_p \sigma_0 + t \eta s \sigma_p)$$
$$\leq \widetilde{\Theta}(1).$$

By Lemma C.1, we have

$$|\mathbf{w}_{j,r}^{\left(t \cdot \frac{n}{B}\right)}[k]| \leq |\mathbf{w}_{j,r}^{\left(t \cdot \frac{n}{B}-1\right)}[k]| + \Theta(\eta)$$
$$\leq |\mathbf{w}_{j,r}^{(0)}[k]| + \Theta(\eta \cdot t \cdot \frac{n}{B})$$
$$\leq \widetilde{\Theta}(\sigma_0 + \frac{n}{Bs\sigma_p})$$
$$\leq \widetilde{\Theta}(1).$$

So we have $|\ell_{j,i}^{\left(t \cdot \frac{n}{B}\right)}| = \Theta(1)$. Follow the same proof above with $t = t_0 + 1$, assuming that sample $(\mathbf{x}_i, y_i)$ is in batch $\mathcal{I}_{t_0 \cdot n/B+\tau}$ in the $t$-th epoch. Then we have

$$\langle \mathbf{w}_{y_i,r}^{\left(t_0 \cdot \frac{n}{B}+\tau\right)}, \boldsymbol{\xi}_i \rangle \geq (1 - \lambda\eta)\langle \mathbf{w}_{y_i,r}^{\left(t_0 \cdot \frac{n}{B}+\tau-1\right)}, \boldsymbol{\xi}_i \rangle - \Theta(\eta\alpha)$$
$$\geq \langle \mathbf{w}_{y_i,r}^{\left(t_0 \cdot \frac{n}{B}\right)}, \boldsymbol{\xi}_i \rangle - \widetilde{\Theta}(\lambda\eta + \eta\alpha)$$
$$= \langle \mathbf{w}_{y_i,r}^{\left(t_0 \cdot \frac{n}{B}\right)}, \boldsymbol{\xi}_i \rangle,$$

since $\eta = o(\sigma_0)$ and $\alpha = o(1)$. Additionally, we have $\eta s = o(\sigma_0^{q-1})$ and $|\boldsymbol{\xi}_i[k]| \geq \widetilde{\Theta}(\sigma_p)$ with high probability. Then $|B^{-1}(\mathbf{w}_{j,r}^{(0)}[k] + t\eta s\sigma_p)^{q-1}\boldsymbol{\xi}_i[k]| \geq \widetilde{\Theta}(\eta B^{-1}s\sigma_p|\ell_{j,i}^{(0)}|)$ for $i \in \mathcal{I}_{t_0 \cdot n/B+\tau}$. Therefore, by Lemma C.11 and A.6, we have

$$\mathrm{sgn}\left(-\frac{1}{B}\ell_{y_i,i}^{\left(t_0 \cdot \frac{n}{B}+\tau\right)}\sigma'(\langle \mathbf{w}_{y_i,r}^{\left(t_0 \cdot \frac{n}{B}+\tau\right)}, \boldsymbol{\xi}_i \rangle)\boldsymbol{\xi}_i[k]\right)$$
$$= -\mathrm{sgn}\left(\ell_{y_i,i}^{\left(t_0 \cdot \frac{n}{B}+\tau\right)}\sigma'(\langle \mathbf{w}_{y_i,r}^{\left(t_0 \cdot \frac{n}{B}+\tau\right)}, \boldsymbol{\xi}_i \rangle)\boldsymbol{\xi}_i[k]\right)$$
$$= -\mathrm{sgn}(\boldsymbol{\xi}_i[k]). \tag{C.30}$$

Then, by Lemma C.11 we have the following update according to (B.6), (C.30) and Lemma C.1.

$$\langle \mathbf{w}_{y_i,r}^{\left(t_0 \cdot \frac{n}{B}+\tau+1\right)}, \boldsymbol{\xi}_i \rangle$$
$$= (1 - \lambda\eta)\langle \mathbf{w}_{y_i,r}^{\left(t_0 \cdot \frac{n}{B}+\tau\right)}, \boldsymbol{\xi}_i \rangle - \eta \cdot \left\langle \frac{\mathbf{m}_{y_i,r}^{\left(t_0 \cdot \frac{n}{B}+\tau\right)}}{\sqrt{\mathbf{v}_{y_i,r}^{\left(t_0 \cdot \frac{n}{B}+\tau\right)}} + \epsilon}, \boldsymbol{\xi}_i \right\rangle$$

$$\geq \langle \mathbf{w}_{y_i,r}^{(t_0 \cdot \frac{n}{B})}, \boldsymbol{\xi}_i \rangle + \Theta(\eta) \cdot \sum_{k \in \mathcal{B}_i} \langle \mathrm{sgn}(\boldsymbol{\xi}_i[k]), \boldsymbol{\xi}_i \rangle - \widetilde{O}(\lambda \eta) - O(\eta \alpha) - O(\eta s \sigma_p)$$

$$= \langle \mathbf{w}_{y_i,r}^{(t_0 \cdot \frac{n}{B})}, \boldsymbol{\xi}_i \rangle + \widetilde{\Theta}(\eta s \sigma_p).$$

At the end of this epoch, we have

$$\langle \mathbf{w}_{y_i,r}^{(t \cdot \frac{n}{B})}, \boldsymbol{\xi}_i \rangle$$
$$\geq (1 - \lambda \eta) \langle \mathbf{w}_{y_i,r}^{(t \cdot \frac{n}{B} - 1)}, \boldsymbol{\xi}_i \rangle - O(\eta \alpha)$$
$$\geq \langle \mathbf{w}_{y_i,r}^{(t_0 \cdot \frac{n}{B} + \tau + 1)}, \boldsymbol{\xi}_i \rangle - \widetilde{O}(\lambda \eta) - O(\eta \alpha)$$
$$\geq \langle \mathbf{w}_{y_i,r}^{(t_0 \cdot \frac{n}{B})}, \boldsymbol{\xi}_i \rangle + \widetilde{\Theta}(\eta s \sigma_p).$$

This completes the proof. $\qquad \square$

Because in the large-batch regime we have $n/B = o(s\sigma_p)$, Lemma C.12 tells us that noise memorization outpaces feature learning. Early in Stage I, feature gradients predominate since $\sigma'(\langle \mathbf{w}_{y_i,r}^{(t)}, y_i \mathbf{v} \rangle) \gg \alpha \sigma'(\langle \mathbf{w}_{y_i,r}^{(t)}, \boldsymbol{\xi}_i \rangle)$, given that $\alpha = o(1)$ and weight decay effect is negligible. After a certain number of epochs, however, the noise component grows until $\alpha \sigma'(\langle \mathbf{w}_{y_i,r}^{(t)}, \boldsymbol{\xi}_i \rangle) \gg \sigma'(\langle \mathbf{w}_{y_i,r}^{(t)}, y_i \mathbf{v} \rangle)$, at which point feature learning slows, then reverses direction entirely. Lemma C.13 below characterizes this transition in detail.

**Lemma C.13** (Stage I, fitting feature noise). *Given the training dataset $\mathcal{S}$, if $\frac{n}{B} = \Theta(1)$ or $\frac{n}{B} = o(s\sigma_p)$, $\eta = 1/\mathrm{poly}(d)$, $\lambda = \widetilde{\Omega}(\frac{B^2}{n} \wedge 1)$ and $\lambda = \widetilde{O}(1)$, then if $\alpha \geq \widetilde{\Theta}\left(\left(\frac{B}{n} s\sigma_p\right)^{1-q}\right)$, for any $t \in [T_r, T_0]$ with $T_r = \widetilde{O}\left(\frac{\sigma_0}{\eta s \sigma_p \alpha^{1/(q-1)}}\right) \leq T_0$,*

$$\langle \mathbf{w}_{j,r}^{((t+1) \cdot \frac{n}{B})}, j \cdot \mathbf{v} \rangle = \langle \mathbf{w}_{j,r}^{(t \cdot \frac{n}{B})}, j \cdot \mathbf{v} \rangle - \Theta(\eta \cdot \frac{n}{B}).$$

*At epoch $T_0$, we have (a) $\mathbf{w}_{j,r}^{(T_0 \cdot \frac{n}{B})}[1] = -\mathrm{sgn}(j) \cdot \widetilde{\Omega}(\frac{n}{B s \sigma_p})$; (b) $\mathbf{w}_{j,r}^{(T_0 \cdot \frac{n}{B})}[k] = \mathrm{sgn}(\boldsymbol{\xi}_i[k]) \cdot \widetilde{\Omega}(\frac{1}{s\sigma_p})$ or $\pm \widetilde{O}(\eta)$ for $k \in \mathcal{B}_i$ with $y_i = j$; (c) $\mathbf{w}_{j,r}^{(T_0 \cdot \frac{n}{B})}[k] = \pm \widetilde{O}(\eta)$ otherwise.*

*Proof of Lemma C.13.* By Lemma C.12, we have

$$\alpha \sigma'\left(\left\langle \mathbf{w}_{y_i,r}^{(T_r \cdot \frac{n}{B})}, \boldsymbol{\xi}_i \right\rangle\right) \geq \alpha \left(\widetilde{\Theta}(s^{1/2}\sigma_p \sigma_0) + T_r \cdot \widetilde{\Theta}(\eta s \sigma_p)\right)^{q-1}$$
$$= \alpha \left(\widetilde{\Theta}(s^{1/2}\sigma_p \sigma_0) + \widetilde{\Theta}(\frac{\sigma_0}{\alpha^{1/(q-1)}})\right)^{q-1}$$
$$\geq \widetilde{\Theta}(\sigma_0^{q-1}),$$
$$\sigma'(\langle \mathbf{w}_{y_i,r}^{(T_r \cdot \frac{n}{B})}, y_i \mathbf{v} \rangle) \leq \left(\widetilde{\Theta}(\sigma_0) + T_r \cdot \Theta(\eta \cdot \frac{n}{B})\right)^{q-1}$$
$$= \left(\widetilde{\Theta}(\sigma_0) + \frac{n\sigma_0}{B s \sigma_p \alpha^{1/(q-1)}}\right)^{q-1}$$
$$\leq \widetilde{\Theta}(\sigma_0^{q-1}).$$

Hence, there exists some constant $C > 0$, for $t \in [T_r \cdot \frac{n}{B}, T_0 \cdot \frac{n}{B}]$,

$$\alpha \sigma'\left(\left\langle \mathbf{w}_{y_i,r}^{(t)}, \boldsymbol{\xi}_i \right\rangle\right) \geq C \cdot \sigma'(\langle \mathbf{w}_{y_i,r}^{(t)}, y_i \mathbf{v} \rangle).$$

Then by Lemma A.8, C.11 and A.6, we have

$$\left\langle \mathbf{w}_{j,r}^{(t+1)}, j\mathbf{v} \right\rangle = (1 - \lambda \eta) \left\langle \mathbf{w}_{j,r}^{(t)}, j\mathbf{v} \right\rangle - \eta \cdot \left\langle \frac{\mathbf{m}_{j,r}^{(t)}}{\sqrt{\mathbf{v}_{j,r}^{(t)} + \epsilon}}, j\mathbf{v} \right\rangle$$

$$\leq \left\langle \mathbf{w}_{j,r}^{(t)}, j\mathbf{v} \right\rangle - \Theta(\eta) \cdot j \cdot \mathrm{sgn}\left( \sum_{i \in \mathcal{I}_t} \alpha y_i \ell_{j,i}^{(t)} \sigma' \left( \left\langle \mathbf{w}_{y_i,r}^{(t)}, \boldsymbol{\xi}_i \right\rangle \right) \right)$$

$$= \left\langle \mathbf{w}_{j,r}^{(t)}, j\mathbf{v} \right\rangle - \Theta(\eta) \cdot j \cdot \mathrm{sgn}(j)$$

$$= \left\langle \mathbf{w}_{j,r}^{(t)}, j\mathbf{v} \right\rangle - \Theta(\eta).$$

So we conclude that

$$\langle \mathbf{w}_{j,r}^{((t+1)\cdot\frac{n}{B})}, j \cdot \mathbf{v} \rangle = \langle \mathbf{w}_{j,r}^{(t\cdot\frac{n}{B})}, j \cdot \mathbf{v} \rangle - \Theta(\eta \cdot \frac{n}{B}).$$

Moreover, at the end of epoch $T_0$,

$$\left\langle \mathbf{w}_{j,r}^{(T_0 \cdot \frac{n}{B})}, j\mathbf{v} \right\rangle = \widetilde{\Theta}(\sigma_0) + T_r \cdot \Theta(\eta \cdot \frac{n}{B}) - (T_0 - T_r) \cdot \Theta(\eta \cdot \frac{n}{B})$$

$$= -\widetilde{\Theta}(\frac{n}{Bs\sigma_p}).$$

Multiply $j$ on both side, we get

$$\mathbf{w}_{j,r}^{(T_0 \cdot \frac{n}{B})}[1] = -\mathrm{sgn}(j) \cdot \widetilde{\Theta}(\frac{n}{Bs\sigma_p}).$$

For $\mathbf{w}_{j,r}^{(T_0 \cdot \frac{n}{B})}[k]$, where $k \in \mathcal{B}_i$ and $i \in [n]$, Lemma C.12 shows that it increases by $\Theta(\eta)$ in the direction of $\mathrm{sgn}(\boldsymbol{\xi}_i[k])$ if $\langle \mathbf{w}_{y_i,r}^{(0)}, \boldsymbol{\xi}_i \rangle > 0$, that is,

$$\mathbf{w}_{j,r}^{(T_0 \cdot \frac{n}{B})}[k] = \mathbf{w}_{j,r}^{(0)}[k] + \mathrm{sgn}(\boldsymbol{\xi}_i[k]) \cdot T_0 \cdot \Theta(\eta)$$

$$= \mathrm{sgn}(\boldsymbol{\xi}_i[k]) \cdot \widetilde{\Theta}\left( \frac{1}{s\sigma_p} \right).$$

Otherwise, weight decay drives $\mathbf{w}_{j,r}^{(t)}[k]$ toward zero if it is initially negative, in this case $\mathbf{w}_{j,r}^{(T_0 \cdot \frac{n}{B})}[k] \in [-\widetilde{\Theta}(\eta), \widetilde{\Theta}(\eta)]$. For the remaining coordinates, Lemma A.8 implies the gradients are zero, so the updates are dominated by weight decay. Given the fact that $T_0\eta = \omega(\sigma_0)$, we have $\mathbf{w}_{j,r}^{(T_0 \cdot \frac{n}{B})}[k] \in [-\widetilde{\Theta}(\eta), \widetilde{\Theta}(\eta)]$. This completes the proof. $\square$

Lemma C.13 implies that, by the end of **Stage I**, the model has fitted the feature noise $-\alpha y\mathbf{v}$. The following Lemma C.14 shows that these pattern persist throughout **Stage II**, ultimately leading to poor generalization.

**Lemma C.14** (Stage II, preserve the noise). *Suppose the same conditions hold as in Lemma C.12 and C.13, for $t > T_0 \cdot \frac{n}{B}$, $j \in \{\pm 1\}$, $r \in [m]$, $i \in [n]$, let $r^* = \mathrm{argmax}_{r\in[m]}\langle \mathbf{w}_{y_i,r}^{(t)}, \boldsymbol{\xi}_i \rangle$, then $\langle \mathbf{w}_{j,r}^{(t)}, j \cdot \mathbf{v} \rangle = -\widetilde{\Theta}(\frac{n}{Bs\sigma_p})$ and $\langle \mathbf{w}_{y_i,r^*}^{(t)}, \boldsymbol{\xi}_i \rangle = \widetilde{\Theta}(1)$.*

*Proof of Lemma C.14.* By Lemma C.12, C.11 and (B.6), we have $\langle \mathbf{w}_{-y_i,r}^{(t)}, \boldsymbol{\xi}_i \rangle \in [-\widetilde{\Theta}(\eta s\sigma_p), \widetilde{\Theta}(\sigma_0)]$. Because if $\langle \mathbf{w}_{-y_i,r}^{(t)}, \boldsymbol{\xi}_i \rangle \geq \widetilde{\Theta}(\sigma_0)$, then we have

$$\langle \mathbf{w}_{-y_i,r}^{(t+\frac{n}{B})}, \boldsymbol{\xi}_i \rangle \leq \langle \mathbf{w}_{-y_i,r}^{(t)}, \boldsymbol{\xi}_i \rangle - \widetilde{\Theta}(\eta s\sigma_p),$$

while if $\langle \mathbf{w}_{-y_i,r}^{(t)}, \boldsymbol{\xi}_i \rangle < 0$, we have

$$\langle \mathbf{w}_{-y_i,r}^{(t+1)}, \boldsymbol{\xi}_i \rangle \geq (1 - \lambda\eta)\langle \mathbf{w}_{-y_i,r}^{(t)}, \boldsymbol{\xi}_i \rangle + \Theta(\eta\alpha).$$

Now, suppose $\langle \mathbf{w}_{y_i,r^*}^{(t)}, \boldsymbol{\xi}_i \rangle \leq \left( \frac{1}{m} \log \left( (\lambda\eta)^{-1} - 1 \right) \right)^{\frac{1}{q}}$, then for $(\mathbf{x}_i, y_i)$ with $y_i = j$,

$$\ell_{j,i}^{(t)} = \frac{e^{F_{-j}(\mathbf{W}^{(t)}, \mathbf{x}_i)}}{\sum_{j \in \{-1,1\}} e^{F_j(\mathbf{W}^{(t)}, \mathbf{x}_i)}}$$

$$= \frac{1}{1 + \exp\left[\sum_{r=1}^{m} \sigma(\langle \mathbf{w}_{j,r}^{(t)}, j\mathbf{v}\rangle) + \sigma(\langle \mathbf{w}_{j,r}^{(t)}, \boldsymbol{\xi}_i\rangle) - \sigma(\langle \mathbf{w}_{-j,r}^{(t)}, j\mathbf{v}\rangle) - \sigma(\langle \mathbf{w}_{-j,r}^{(t)}, \boldsymbol{\xi}_i\rangle)\right]}$$

$$\geq \frac{1}{1 + \exp\left[\sum_{r=1}^{m} \sigma(\langle \mathbf{w}_{j,r}^{(t)}, \boldsymbol{\xi}_i\rangle)\right]}$$

$$\geq \frac{1}{1 + \exp\left[m \cdot \left(\frac{1}{m} \log\left((\lambda\eta)^{-1} - 1\right)\right)^{\frac{1}{q} \cdot q}\right]}$$

$$= \Theta(\lambda\eta),$$

where the inequality we use $\langle \mathbf{w}_{j,r}^{(t)}, j\mathbf{v}\rangle < 0$. Then, in epoch $T_a$, which contains iteration $t$, it follows from Lemmas C.11, C.12 and (B.6) that for all $t_a \in [t, T_a + \frac{n}{B}]$, we have

$$\langle \mathbf{w}_{y_i,r^*}^{(t_a)}, \boldsymbol{\xi}_i\rangle \geq (1 - \lambda\eta)\langle \mathbf{w}_{y_i,r^*}^{(t)}, \boldsymbol{\xi}_i\rangle + \widetilde{\Theta}(\eta s\sigma_p)$$

$$\geq \langle \mathbf{w}_{y_i,r^*}^{(t)}, \boldsymbol{\xi}_i\rangle + \widetilde{\Theta}(\eta s\sigma_p).$$

If $\langle \mathbf{w}_{y_i,r^*}^{(t)}, \boldsymbol{\xi}_i\rangle \geq \left(\log\left((\lambda\eta)^{-2} - 1\right)\right)^{\frac{1}{q}}$, then for $(\mathbf{x}_i, y_i)$ with $y_i = j$,

$$\ell_{j,i}^{(t)} = \frac{e^{F_{-j}(\mathbf{W}^{(t)}, \mathbf{x}_i)}}{\sum_{j\in\{-1,1\}} e^{F_j(\mathbf{W}^{(t)}, \mathbf{x}_i)}}$$

$$= \frac{1}{1 + \exp\left[\sum_{r=1}^{m} \sigma(\langle \mathbf{w}_{j,r}^{(t)}, j\mathbf{v}\rangle) + \sigma(\langle \mathbf{w}_{j,r}^{(t)}, \boldsymbol{\xi}_i\rangle) - \sigma(\langle \mathbf{w}_{-j,r}^{(t)}, j\mathbf{v}\rangle) - \sigma(\langle \mathbf{w}_{-j,r}^{(t)}, \boldsymbol{\xi}_i\rangle)\right]}$$

$$\leq \frac{1}{1 + \exp\left[\sigma(\langle \mathbf{w}_{y_i,r^*}^{(t)}, \boldsymbol{\xi}_i\rangle)\right]}$$

$$\leq \frac{1}{1 + \exp\left[(\log\left((\lambda\eta)^{-2} - 1\right))^{\frac{1}{q} \cdot q}\right]}$$

$$= \Theta(\lambda^2\eta^2),$$

where the inequality we use $\langle \mathbf{w}_{-j,r}^{(t)}, j\mathbf{v}\rangle \leq \widetilde{\Theta}(\frac{n}{Bs\sigma_p})$, $\frac{n}{B} = o(s\sigma_p)$, $\langle \mathbf{w}_{-j,r}^{(t)}, \boldsymbol{\xi}_i\rangle) \leq \widetilde{\Theta}(\sigma_0)$. Then by Lemma C.11, C.12 and (B.6), we have

$$\langle \mathbf{w}_{y_i,r^*}^{(t+1)}, \boldsymbol{\xi}_i\rangle \leq (1 - \lambda\eta)\langle \mathbf{w}_{y_i,r^*}^{(t)}, \boldsymbol{\xi}_i\rangle + \widetilde{\Theta}(\lambda\eta \cdot \eta s\sigma_p)$$

$$\leq \langle \mathbf{w}_{y_i,r^*}^{(t)}, \boldsymbol{\xi}_i\rangle,$$

since $\eta s\sigma_p = o(1)$. For $\langle \mathbf{w}_{j,r}^{(t)}, j\mathbf{v}\rangle = -\widetilde{\Theta}(\frac{n}{Bs\sigma_p})$, the same proof applies, since $s\sigma_p = o(1)$ and $\frac{n}{B} = o(s\sigma_p)$. If $\langle \mathbf{w}_{y_i,r^*}^{(t)}, \boldsymbol{\xi}_i\rangle \geq \left(\log\left((\lambda\eta)^{-2} - 1\right)\right)^{\frac{1}{q}}$, then for $(\mathbf{x}_i, y_i)$ with $y_i = j$,

$$\ell_{j,i}^{(t)} \leq \Theta(\lambda^2\eta^2).$$

Then by Lemma C.11, C.12 and (B.5), we have

$$\langle \mathbf{w}_{j,r}^{(t+1)}, j\mathbf{v}\rangle \geq (1 - \lambda\eta)\langle \mathbf{w}_{j,r}^{(t)}, j\mathbf{v}\rangle - \Theta(\lambda\eta \cdot \eta)$$

$$\geq \langle \mathbf{w}_{j,r}^{(t)}, j\mathbf{v}\rangle + \widetilde{\Theta}(\lambda\eta \cdot \frac{n}{Bs\sigma_p}) - \Theta(\lambda\eta \cdot \eta)$$

$$\geq \langle \mathbf{w}_{j,r}^{(t)}, j\mathbf{v}\rangle,$$

since $\eta = o(\frac{1}{s\sigma_p})$. This completes the proof. $\qquad\square$

**Lemma C.15** (Convergence). *Suppose the same conditions hold as in Lemma C.12, C.13 and C.14, if the step size satisfies $\eta = O(d^{-1/2})$, then for any $t$,*

$$\mathbb{E}\left[L(\mathbf{W}^{(t+1)}) - L(\mathbf{W}^{(t)})\right] \leq -\eta\|\nabla L(\mathbf{W}^{(t)})\|_1 + \widetilde{\Theta}(\eta^2 d).$$

*Proof of Lemma C.15.* The proof is similar to Lemma C.8. We aim to prove the convergence of the objective function under the AdamW optimization algorithm in a non-convex setting. Recall the loss function for each data point $i$ is

$$L_i(\mathbf{W}) = \log\left(1 + \frac{1}{\exp\left(F_{y_i}(\mathbf{W}, \mathbf{x}_i) - F_{-y_i}(\mathbf{W}, \mathbf{x}_i)\right)}\right),$$

where $\mathbf{W}$ represents the parameter matrix, $\mathbf{x}_i$ is the input data, $y_i$ is the true label, and $F_j(\mathbf{W}, \mathbf{x}_i)$ are the logits for class $j$. The total objective is:

$$L(\mathbf{W}) = \frac{1}{n}\sum_{i=1}^n L_i(\mathbf{W})$$

Since $L_i(\mathbf{W})$ is non-convex, we exploit its smoothness with respect to the logits $[F_j(\mathbf{W}, \mathbf{x}_i)]_j$. Specifically, $L_i(\mathbf{W})$ is 1-smooth in $[F_j(\mathbf{W}, \mathbf{x}_i)]_j$ due to the properties of the cross-entropy loss. Define:

$$\Delta F_{j,i} = F_j(\mathbf{W}^{(t+1)}, \mathbf{x}_i) - F_j(\mathbf{W}^{(t)}, \mathbf{x}_i).$$

Using the smoothness property, we apply a second-order Taylor-like expansion around $\mathbf{W}^{(t)}$:

$$L_i(\mathbf{W}^{(t+1)}) - L_i(\mathbf{W}^{(t)}) \leq \sum_j \frac{\partial L_i(\mathbf{W}^{(t)})}{\partial F_j(\mathbf{W}^{(t)}, \mathbf{x}_i)} \cdot \Delta F_{j,i} + \sum_j (\Delta F_{j,i})^2. \tag{C.31}$$

This upper bound arises because the second derivative of $L_i$ with respect to the logits is bounded by 1, a standard result for cross-entropy loss. The logits are defined as: $F_j(\mathbf{W}^{(t)}, \mathbf{x}_i) = \sum_{r=1}^m [\sigma(\langle \mathbf{w}_{j,r}^{(t)}, y_i\mathbf{v}\rangle) + \sigma(\langle \mathbf{w}_{j,r}^{(t)}, \boldsymbol{\xi}_i\rangle)]$, where $\mathbf{w}_{j,r}^{(t)}$ the $r$-th neuron in $j$-th output of $\mathbf{W}^{(t)}$, $\sigma(z) = [z]_+^q$ is a smooth activation function (e.g., with $q \geq 3$). By Lemma C.14 and C.1, we have $\langle \mathbf{w}_{j,r}^{(t)}, \mathbf{v}\rangle \leq \widetilde{\Theta}(1)$ and $\langle \mathbf{w}_{j,r}^{(t)}, \boldsymbol{\xi}_i\rangle \leq \widetilde{\Theta}(1)$, ensuring the local smoothness of $\sigma$ remains $\widetilde{O}(1)$ between $\langle \mathbf{w}_{j,r}^{(t+1)}, y_i\mathbf{v}\rangle$ and $\langle \mathbf{w}_{j,r}^{(t)}, y_i\mathbf{v}\rangle$ (similar for $\langle \mathbf{w}_{j,r}^{(t)}, \boldsymbol{\xi}_i\rangle$). Then with Taylor expansion, we have

$$\left|\sigma(\langle \mathbf{w}_{j,r}^{(t+1)}, y_i\mathbf{v}\rangle) - \sigma(\langle \mathbf{w}_{j,r}^{(t)}, y_i\mathbf{v}\rangle) - \langle \nabla_{\mathbf{w}_{j,r}}\sigma(\langle \mathbf{w}_{j,r}^{(t)}, y_i\mathbf{v}\rangle), \mathbf{w}_{j,r}^{(t+1)} - \mathbf{w}_{j,r}^{(t)}\rangle\right|$$

$$\leq \widetilde{\Theta}(1) \cdot \left\|\mathbf{w}_{j,r}^{(t+1)} - \mathbf{w}_{j,r}^{(t)}\right\|_2^2$$

$$= \widetilde{\Theta}(1) \cdot \left\|\lambda\eta \cdot \mathbf{w}_{j,r}^{(t)} + \eta \cdot \frac{\mathbf{m}_{j,r}^{(t)}}{\sqrt{\mathbf{v}_{j,r}^{(t)}} + \epsilon}\right\|_2^2$$

$$\leq \widetilde{\Theta}(\eta^2 d), \tag{C.32}$$

where the last inequality we use Lemma C.1 and $\|\mathbf{w}_{j,r}^{(t)}\|_2^2 \ll \Theta(d)$ by Lemma C.13 and C.14. Similarly, we have

$$\left|\sigma(\langle \mathbf{w}_{j,r}^{(t+1)}, \boldsymbol{\xi}_i\rangle) - \sigma(\langle \mathbf{w}_{j,r}^{(t)}, \boldsymbol{\xi}_i\rangle) - \langle \nabla_{\mathbf{w}_{j,r}}\sigma(\langle \mathbf{w}_{j,r}^{(t)}, \boldsymbol{\xi}_i\rangle), \mathbf{w}_{j,r}^{(t+1)} - \mathbf{w}_{j,r}^{(t)}\rangle\right|$$

$$\leq \widetilde{\Theta}(1) \cdot \left\|\mathbf{w}_{j,r}^{(t+1)} - \mathbf{w}_{j,r}^{(t)}\right\|_2^2$$

$$\leq \widetilde{\Theta}(\eta^2 d). \tag{C.33}$$

Summing over $r$ (with $m = \widetilde{\Theta}(1)$), we get

$$|\Delta F_{j,i}| \leq \left|\langle \nabla_{\mathbf{W}}F_j(\mathbf{W}^{(t)}, \mathbf{x}_i), \mathbf{W}^{(t+1)} - \mathbf{W}^{(t)}\rangle\right| + \widetilde{\Theta}(\eta^2 d). \tag{C.34}$$

Additionally, $\|\nabla_{\mathbf{W}}F_j(\mathbf{W}^{(t)}, \mathbf{x}_i)\|_F \leq \widetilde{\Theta}(1)$ since $m = \widetilde{\Theta}(1)$, $\langle \mathbf{w}_{j,r}^{(t)}, y_i\mathbf{v}\rangle \leq \widetilde{\Theta}(1)$, $\langle \mathbf{w}_{j,r}^{(t)}, \boldsymbol{\xi}_i\rangle \leq \widetilde{\Theta}(1)$. So we have

$$|\Delta F_{j,i}| \leq \widetilde{\Theta}(\eta s\sigma_p + \eta\alpha + \eta + \eta^2 d) \leq \widetilde{\Theta}(\eta s\sigma_p + \eta^2 d). \tag{C.35}$$

Substitute (C.34) and (C.35) into (C.31):

$$L_i(\mathbf{W}^{(t+1)}) - L_i(\mathbf{W}^{(t)}) \leq \langle \nabla_{\mathbf{W}} L_i(\mathbf{W}^{(t)}), \mathbf{W}^{(t+1)} - \mathbf{W}^{(t)} \rangle + \widetilde{\Theta}(\eta^2 d). \tag{C.36}$$

For the full objective:

$$L(\mathbf{W}^{(t+1)}) - L(\mathbf{W}^{(t)}) = \frac{1}{n}\sum_{i=1}^{n}[L_i(\mathbf{W}^{(t+1)}) - L_i(\mathbf{W}^{(t)})]. \tag{C.37}$$

Substitute (C.36) into (C.37), we have

$$L(\mathbf{W}^{(t+1)}) - L(\mathbf{W}^{(t)}) \leq \langle \nabla L(\mathbf{W}^{(t)}), \mathbf{W}^{(t+1)} - \mathbf{W}^{(t)} \rangle + \widetilde{\Theta}(\eta^2 d). \tag{C.38}$$

Take expectation for the stochastic gradient of both side in (C.38),

$$\mathbb{E}\left[ L(\mathbf{W}^{(t+1)}) - L(\mathbf{W}^{(t)}) \right]$$

$$\leq \mathbb{E}\left[ \langle \nabla L(\mathbf{W}^{(t)}), \mathbf{W}^{(t+1)} - \mathbf{W}^{(t)} \rangle \right] + \widetilde{\Theta}(\eta^2 d)$$

$$\leq -\eta \cdot \mathbb{E}\left[ \sum_{j\in\{\pm 1\}}\sum_{r\in[m]} \left\| g_{t,j,r}^{(t)} \right\|_1 \right] - \lambda\eta \cdot \langle \nabla L(\mathbf{W}^{(t)}), \mathbf{W}^{(t)} \rangle + \widetilde{\Theta}(d\cdot\eta^2) + \widetilde{\Theta}(ns\cdot\eta^2 s\sigma_p) + \widetilde{\Theta}(\eta^2 d)$$

$$\leq -\eta \cdot \sum_{j\in\{\pm 1\}}\sum_{r\in[m]} \left\| \mathbb{E}\left[ g_{t,j,r}^{(t)} \right] \right\|_1 + \widetilde{\Theta}(\frac{\lambda\eta n}{Bs\sigma_p}) \cdot \|\nabla L(\mathbf{W}^{(t)})\|_1 + \widetilde{\Theta}(\eta^2 d)$$

$$\leq -\eta\|\nabla L(\mathbf{W}^{(t)})\|_1 + \widetilde{\Theta}(\eta^2 d),$$

where we use Lemma C.11 that the update aligns with the gradient's sign for large gradient and the fact that $ns^2\sigma_p = O(d)$, and Jensen's inequality, Hölder's inequality and $\frac{n}{B} = o(s\sigma_p)$, $\|\mathbf{W}^{(t)}\|_\infty \leq \widetilde{\Theta}(\frac{n}{Bs\sigma_p})$ by Lemma C.13 and C.14. This completes the proof. $\qquad\square$

**Lemma C.16** (Generalization of Stochastic AdamW, large-batch). *Suppose the same conditions hold as in Lemma C.15. We have the following results for $T = \frac{\text{poly}(n)}{\eta}$, with training dataset $\mathcal{S}$*

- *The training error is zero:* $\text{err}_{\mathcal{S}}(\mathbf{W}^{(T)}) = 0$.

- *The test error is high:* $\text{err}_{\mathcal{D}}(\mathbf{W}^{(T)}) \geq \frac{1}{2} - o(1)$.

*Proof of Lemma C.16.* By Lemma C.14, we have

$$\langle \mathbf{w}_{j,r}^{(T)}, j\mathbf{v} \rangle = -\widetilde{\Theta}(\frac{1}{s\sigma_p}), \quad \langle \mathbf{w}_{y_i,r^*}^{(T)}, \boldsymbol{\xi}_i \rangle = \widetilde{\Theta}(1), \quad \langle \mathbf{w}_{-y_i,r}^{(T)}, \boldsymbol{\xi}_i \rangle \leq \widetilde{\Theta}(\sigma_0).$$

Recall $F_j(\mathbf{W}, \mathbf{x})$ in Definition 3.2, with $\eta s\sigma_p = o(1)$, $\alpha = o(1)$, we directly have

$$\text{err}_{\mathcal{S}}(\mathbf{W}^{(T)}) = \mathbb{E}_{(\mathbf{x},y)\sim\mathcal{S}}\, \mathbb{1}\left[ F_y(\mathbf{W}^{(T)}, \mathbf{x}) \leq F_{-y}(\mathbf{W}^{(T)}, \mathbf{x}) \right] = 0,$$

since $F_{y_i}(\mathbf{W}^{(T)}, \mathbf{x}_i) = \widetilde{\Omega}(1)$, while $F_{-y_i}(\mathbf{W}^{(T)}, \mathbf{x}) \leq \widetilde{\Theta}(\frac{1}{s\sigma_p} + \sigma_0)$ and $s\sigma_p = \omega(1)$. Besides, for test data $(\mathbf{x}, y) \sim \mathcal{D}$ with $\mathbf{x} = [y\mathbf{v}^\top, \boldsymbol{\xi}^\top]^\top$, it is clear that with high probability $\langle \mathbf{w}_{y,r}^{(T)}, y\mathbf{v} \rangle = -\widetilde{\Theta}(\frac{1}{s\sigma_p})$, then similar as training error, we have

$$F_y(\mathbf{W}^{(T)}, \mathbf{x}) = \sum_{r=1}^{m}\left[ \sigma(\langle \mathbf{w}_{y,r}^{(T)}, y\mathbf{v} \rangle) + \sigma(\langle \mathbf{w}_{y,r}^{(T)}, \boldsymbol{\xi} \rangle) \right] = \sum_{r=1}^{m}\left[ \frac{\alpha}{s\sigma_p} + \zeta_{y,r} \right]_+^q,$$

while

$$F_{-y}(\mathbf{W}^{(T)}, \mathbf{x}) = \sum_{r=1}^{m}\left[ \sigma(\langle \mathbf{w}_{-y,r}^{*}, y\mathbf{v} \rangle) + \sigma(\langle \mathbf{w}_{-y,r}^{*}, \boldsymbol{\xi} \rangle) \right] = \sum_{r=1}^{m}\left[ [\widetilde{\Theta}(\frac{1}{s\sigma_p})]^q + [\zeta_{-y,r}]_+^q \right].$$

Here, $\zeta_{y,r}$ and $\zeta_{-y,r}$ are independent and symmetric random variables. Therefore, if the term $\widetilde{\Theta}\left(\frac{1}{s\sigma_p}\right)$ dominates $\zeta_{y,r}$ and $\zeta_{-y,r}$, then it is immediate that $F_y(\mathbf{W}^{(T)}, \mathbf{x}) < F_{-y}(\mathbf{W}^{(T)}, \mathbf{x})$, since $\alpha = o(1)$. This implies that large-batch AdamW yields high test error. On the other hand, if $\widetilde{\Theta}\left(\frac{1}{s\sigma_p}\right)$ is dominated by both $\zeta_{y,r}$ and $\zeta_{-y,r}$, then with probability at least $1/2 - o(1)$, we have $F_y(\mathbf{W}^{(T)}, \mathbf{x}) < F_{-y}(\mathbf{W}^{(T)}, \mathbf{x})$, as $\zeta_{y,r}$ and $\zeta_{-y,r}$ are independent of $\mathbf{v}$. In this case, large-batch AdamW incurs at least $1/2 - o(1)$ test error. Therefore, we conclude:

$$\mathrm{err}_{\mathcal{D}}(\mathbf{W}^{(T)}) = \mathbb{E}_{(\mathbf{x},y)\sim\mathcal{D}} \mathbb{1}\left[F_y(\mathbf{W}^{(T)}, \mathbf{x}) \leq F_{-y}(\mathbf{W}^{(T)}, \mathbf{x})\right] \geq \frac{1}{2} - o(1).$$

This completes the proof. $\qquad\square$

### C.2.2 Proof of Theorem 4.5

**Lemma C.17** (Stage I). *Given the training dataset $\mathcal{S}$, if $\frac{n}{B} \geq \Theta(n^{1/2} \vee \log\epsilon^{-1})$ and $\frac{n}{B} = \omega(s\sigma_p)$, $\eta = 1/\mathrm{poly}(d)$, $\lambda = \widetilde{\Omega}(\frac{B^2}{n} \wedge 1)$ and $\lambda = \widetilde{O}(1)$, then for any $t \leq T_0$ with $T_0 = \widetilde{O}(\frac{B}{\eta n})$,*

$$\langle \mathbf{w}_{j,r}^{\left((t+1)\cdot\frac{n}{B}\right)}, j\mathbf{v} \rangle = \langle \mathbf{w}_{j,r}^{\left(t\cdot\frac{n}{B}\right)}, j\mathbf{v} \rangle + \Theta(\eta \cdot \frac{n}{B}),$$

$$\langle \mathbf{w}_{y_i,r}^{\left((t+1)\cdot\frac{n}{B}\right)}, \boldsymbol{\xi}_i \rangle = \langle \mathbf{w}_{y_i,r}^{\left(t\cdot\frac{n}{B}\right)}, \boldsymbol{\xi}_i \rangle + \widetilde{\Theta}(\eta s\sigma_p).$$

*Proof of Lemma C.17.* We prove this Lemma by induction. First, we prove $\langle \mathbf{w}_{y_i,r}^{\left((t+1)\cdot\frac{n}{B}\right)}, \boldsymbol{\xi}_i \rangle = \langle \mathbf{w}_{y_i,r}^{\left(t\cdot\frac{n}{B}\right)}, \boldsymbol{\xi}_i \rangle + \widetilde{\Theta}(\eta s\sigma_p)$. It is same as Lemma C.12. By Lemma A.3, we have

$$|\langle \mathbf{w}_{j,r}^{(0)}, \boldsymbol{\xi}_i \rangle| = \widetilde{\Theta}(s^{1/2}\sigma_p\sigma_0), \quad \mathbf{w}_{j,r}^{(0)}[k] = \widetilde{\Theta}(\sigma_0),$$

which imply that $|\ell_{j,i}^{(0)}| = \Theta(1)$. Assume that sample $(\mathbf{x}_i, y_i)$ is in batch $\mathcal{I}_\tau$ in the first epoch. Then we have

$$\begin{aligned}
\langle \mathbf{w}_{y_i,r}^{(\tau)}, \boldsymbol{\xi}_i \rangle &\geq (1-\lambda\eta)\langle \mathbf{w}_{y_i,r}^{(\tau-1)}, \boldsymbol{\xi}_i \rangle - \Theta(\eta\alpha) \\
&\geq \langle \mathbf{w}_{y_i,r}^{(0)}, \boldsymbol{\xi}_i \rangle - \widetilde{\Theta}(\lambda\eta\sigma_0 + \eta\alpha) \\
&= \widetilde{\Theta}(\sigma_0),
\end{aligned}$$

since $\lambda\eta = o(1)$, $\eta = o(\sigma_0)$, $\alpha = o(1)$ and $s^{1/2}\sigma_p = \widetilde{O}(1)$. Additionally, we have $\eta s = o(\sigma_0^{q-1})$ and $|\boldsymbol{\xi}_i[k]| \geq \widetilde{\Theta}(\sigma_p)$ with high probability. Then $|B^{-1}\sigma_0^{q-1}\boldsymbol{\xi}_i[k]| \geq \widetilde{\Theta}(\eta B^{-1}s\sigma_p|\ell_{j,i}^{(0)}|)$ for $i \in \mathcal{I}_\tau$. Therefore, by Lemma C.11 and A.6, we have

$$\begin{aligned}
&\mathrm{sgn}\left(-\frac{1}{B}\ell_{y_i,i}^{(0)}\sigma'(\langle \mathbf{w}_{y_i,r}^{(0)}, \boldsymbol{\xi}_i \rangle)\boldsymbol{\xi}_i[k]\right) \\
&= -\mathrm{sgn}\left(\ell_{y_i,i}^{(0)}\sigma'(\langle \mathbf{w}_{y_i,r}^{(0)}, \boldsymbol{\xi}_i \rangle)\boldsymbol{\xi}_i[k]\right) \\
&= -\mathrm{sgn}(\boldsymbol{\xi}_i[k]).
\end{aligned} \tag{C.39}$$

Then, by Lemma C.11 we have the following update according to (B.6), (C.29) and Lemma C.1.

$$\begin{aligned}
&\langle \mathbf{w}_{y_i,r}^{(\tau+1)}, \boldsymbol{\xi}_i \rangle \\
&= (1-\lambda\eta)\langle \mathbf{w}_{y_i,r}^{(\tau)}, \boldsymbol{\xi}_i \rangle - \eta \cdot \left\langle \frac{\mathbf{m}_{y_i,r}^{(\tau)}}{\sqrt{\mathbf{v}_{y_i,r}^{(\tau)} + \epsilon}}, \boldsymbol{\xi}_i \right\rangle \\
&\geq \langle \mathbf{w}_{y_i,r}^{(0)}, \boldsymbol{\xi}_i \rangle + \Theta(\eta) \cdot \sum_{k\in\mathcal{B}_i}\langle \mathrm{sgn}(\boldsymbol{\xi}_i[k]), \boldsymbol{\xi}_i \rangle - \widetilde{O}(\lambda\eta\sigma_0) - O(\eta\alpha) - O(\eta s\sigma_p) \\
&= \langle \mathbf{w}_{y_i,r}^{(0)}, \boldsymbol{\xi}_i \rangle + \widetilde{\Theta}(\eta s\sigma_p).
\end{aligned}$$

At the end of the first epoch, we have

$$\langle \mathbf{w}_{y_i,r}^{\left(\frac{n}{B}\right)}, \boldsymbol{\xi}_i \rangle$$

$$\geq (1-\lambda\eta)\langle \mathbf{w}_{y_i,r}^{(\frac{n}{B}-1)}, \boldsymbol{\xi}_i\rangle - \widetilde{O}(\eta\alpha)$$

$$\geq \langle \mathbf{w}_{y_i,r}^{(\tau+1)}, \boldsymbol{\xi}_i\rangle - \widetilde{O}(\lambda\eta^2 s\sigma_p) - \widetilde{O}(\eta\alpha)$$

$$\geq \langle \mathbf{w}_{y_i,r}^{(0)}, \boldsymbol{\xi}_i\rangle + \widetilde{\Theta}(\eta s\sigma_p).$$

This completes the base case for $t=1$. For general $t \leq t_0$ with $t_0 \leq T_0$, assuming $\langle \mathbf{w}_{y_i,r}^{(t\cdot\frac{n}{B})}, \boldsymbol{\xi}_i\rangle = \langle \mathbf{w}_{y_i,r}^{(t-1)\cdot\frac{n}{B}}, \boldsymbol{\xi}_i\rangle + \widetilde{\Theta}(\eta s\sigma_p)$. Then we have

$$\langle \mathbf{w}_{y_i,r}^{(t\cdot\frac{n}{B})}, \boldsymbol{\xi}_i\rangle = \langle \mathbf{w}_{y_i,r}^{(t-1)\cdot\frac{n}{B}}, \boldsymbol{\xi}_i\rangle + \widetilde{\Theta}(\eta s\sigma_p)$$

$$= \langle \mathbf{w}_{y_i,r}^{(0)}, \boldsymbol{\xi}_i\rangle + \widetilde{\Theta}(t\eta s\sigma_p)$$

$$= \widetilde{\Theta}(s^{1/2}\sigma_p\sigma_0 + t\eta s\sigma_p)$$

$$\leq \widetilde{\Theta}(1).$$

By Lemma C.1, we have

$$|\mathbf{w}_{j,r}^{(t\cdot\frac{n}{B})}[k]| \leq |\mathbf{w}_{j,r}^{(t\cdot\frac{n}{B}-1)}[k]| + \Theta(\eta)$$

$$\leq |\mathbf{w}_{j,r}^{(0)}[k]| + \Theta(\eta \cdot t \cdot \frac{n}{B})$$

$$\leq \widetilde{\Theta}(1).$$

It's also obvious that $\langle \mathbf{w}_{j,r}^{(t\cdot\frac{n}{B})}, j\mathbf{v}\rangle = \widetilde{O}(1)$. So we have $|\ell_{j,i}^{(t\cdot\frac{n}{B})}| = \Theta(1)$. Follow the same proof above with $t = t_0 + 1$, assuming that sample $(\mathbf{x}_i, y_i)$ is in batch $\mathcal{I}_{t_0\cdot n/B+\tau}$ in the $t$-th epoch. Then we have

$$\langle \mathbf{w}_{y_i,r}^{(t_0\cdot\frac{n}{B}+\tau)}, \boldsymbol{\xi}_i\rangle \geq (1-\lambda\eta)\langle \mathbf{w}_{y_i,r}^{(t_0\cdot\frac{n}{B}+\tau-1)}, \boldsymbol{\xi}_i\rangle - \Theta(\eta\alpha)$$

$$\geq \langle \mathbf{w}_{y_i,r}^{(t_0\cdot\frac{n}{B})}, \boldsymbol{\xi}_i\rangle - \widetilde{\Theta}(\lambda\eta + \eta\alpha)$$

$$= \langle \mathbf{w}_{y_i,r}^{(t_0\cdot\frac{n}{B})}, \boldsymbol{\xi}_i\rangle,$$

since $\eta = o(\sigma_0)$ and $\alpha = o(1)$. Additionally, we have $\eta s = o(\sigma_0^{q-1})$ and $|\boldsymbol{\xi}_i[k]| \geq \widetilde{\Theta}(\sigma_p)$ with high probability. Then $|B^{-1}(\mathbf{w}_{j,r}^{(0)}[k] + t\eta s\sigma_p)^{q-1}\boldsymbol{\xi}_i[k]| \geq \widetilde{\Theta}(\eta B^{-1}s\sigma_p|\ell_{j,i}^{(0)}|)$ for $i \in \mathcal{I}_{t_0\cdot n/B+\tau}$. Therefore, by Lemma C.11 and A.6, we have

$$\text{sgn}\left(-\frac{1}{B}\ell_{y_i,i}^{(t_0\cdot\frac{n}{B}+\tau)}\sigma'(\langle \mathbf{w}_{y_i,r}^{(t_0\cdot\frac{n}{B}+\tau)}, \boldsymbol{\xi}_i\rangle)\boldsymbol{\xi}_i[k]\right)$$

$$= -\text{sgn}\left(\ell_{y_i,i}^{(t_0\cdot\frac{n}{B}+\tau)}\sigma'(\langle \mathbf{w}_{y_i,r}^{(t_0\cdot\frac{n}{B}+\tau)}, \boldsymbol{\xi}_i\rangle)\boldsymbol{\xi}_i[k]\right)$$

$$= -\text{sgn}(\boldsymbol{\xi}_i[k]). \tag{C.40}$$

Then, by Lemma C.11 we have the following update according to (B.6), (C.30) and Lemma C.1.

$$\langle \mathbf{w}_{y_i,r}^{(t_0\cdot\frac{n}{B}+\tau+1)}, \boldsymbol{\xi}_i\rangle$$

$$= (1-\lambda\eta)\langle \mathbf{w}_{y_i,r}^{(t_0\cdot\frac{n}{B}+\tau)}, \boldsymbol{\xi}_i\rangle - \eta \cdot \left\langle \frac{\mathbf{m}_{y_i,r}^{(t_0\cdot\frac{n}{B}+\tau)}}{\sqrt{\mathbf{v}_{y_i,r}^{(t_0\cdot\frac{n}{B}+\tau)}} + \epsilon}, \boldsymbol{\xi}_i\right\rangle$$

$$\geq \langle \mathbf{w}_{y_i,r}^{(t_0\cdot\frac{n}{B})}, \boldsymbol{\xi}_i\rangle + \Theta(\eta) \cdot \sum_{k\in\mathcal{B}_i}\langle \text{sgn}(\boldsymbol{\xi}_i[k]), \boldsymbol{\xi}_i\rangle - \widetilde{O}(\lambda\eta) - O(\eta\alpha) - O(\eta s\sigma_p)$$

$$= \langle \mathbf{w}_{y_i,r}^{(t_0\cdot\frac{n}{B})}, \boldsymbol{\xi}_i\rangle + \widetilde{\Theta}(\eta s\sigma_p).$$

At the end of this epoch, we have

$$\langle \mathbf{w}_{y_i,r}^{(t\cdot\frac{n}{B})}, \boldsymbol{\xi}_i\rangle$$

$$\geq (1 - \lambda\eta)\langle \mathbf{w}_{y_i,r}^{\left(t \cdot \frac{n}{B} - 1\right)}, \boldsymbol{\xi}_i \rangle - O(\eta\alpha)$$

$$\geq \langle \mathbf{w}_{y_i,r}^{\left(t_0 \cdot \frac{n}{B} + \tau + 1\right)}, \boldsymbol{\xi}_i \rangle - \widetilde{O}(\lambda\eta) - O(\eta\alpha)$$

$$\geq \langle \mathbf{w}_{y_i,r}^{\left(t_0 \cdot \frac{n}{B}\right)}, \boldsymbol{\xi}_i \rangle + \widetilde{\Theta}(\eta s \sigma_p).$$

Next, we prove $\langle \mathbf{w}_{j,r}^{\left((t+1)\cdot\frac{n}{B}\right)}, j\mathbf{v} \rangle = \langle \mathbf{w}_{j,r}^{\left(t\cdot\frac{n}{B}\right)}, j\mathbf{v} \rangle + \Theta(\eta \cdot \frac{n}{B})$. By Lemma A.3, A.8 and C.11, we have

$$\sigma'(\langle \mathbf{w}_{j,r}^{(0)}, j\mathbf{v} \rangle) - \alpha\sigma'(\langle \mathbf{w}_{j,r}^{(0)}, \boldsymbol{\xi}_i \rangle) \geq \widetilde{\Theta}(\sigma_0^{q-1}),$$

since $\alpha = o(1)$, and we have $\eta = o(\sigma_0^{q-1})$. By Lemma A.6, we have

$$\mathrm{sgn}(g_{0,j,r}^{(0)}) = \mathrm{sgn}(-\mathrm{sgn}(j)) = -\mathrm{sgn}(j).$$

Apply Lemma C.11, we get

$$\langle \mathbf{w}_{j,r}^{(1)}, j \cdot \mathbf{v} \rangle = (1 - \lambda\eta)\langle \mathbf{w}_{j,r}^{(0)}, j \cdot \mathbf{v} \rangle - \Theta(\eta) \cdot \langle \mathrm{sgn}(g_{0,j,r}^{(0)}), j\mathbf{v} \rangle$$

$$= \langle \mathbf{w}_{j,r}^{(0)}, j \cdot \mathbf{v} \rangle + \Theta(\eta),$$

since $\langle \mathbf{w}_{j,r}^{(0)}, j \cdot \mathbf{v} \rangle = \widetilde{\Theta}(\sigma_0)$ and $\lambda = \widetilde{O}(1)$. We have

$$\langle \mathbf{w}_{y_i,r}^{\left((t+1)\cdot\frac{n}{B}\right)}, \boldsymbol{\xi}_i \rangle \geq \langle \mathbf{w}_{y_i,r}^{\left(t\cdot\frac{n}{B}\right)}, \boldsymbol{\xi}_i \rangle + \widetilde{\Theta}(\eta s \sigma_p).$$

So we have $\langle \mathbf{w}_{y_i,r}^{(t)}, \boldsymbol{\xi}_i \rangle \leq \widetilde{\Theta}(s^{1/2}\sigma_p\sigma_0 + \eta s \sigma_p)$ for $t \in [0, \frac{n}{B}]$. Thus, for $t \in [0, \frac{n}{B}]$, we have

$$\sigma'(\langle \mathbf{w}_{j,r}^{(t)}, j\mathbf{v} \rangle) \gg \alpha\sigma'(\langle \mathbf{w}_{j,r}^{(t)}, \boldsymbol{\xi}_i \rangle),$$

since $\alpha = o(1)$. By Lemma A.8 and A.6, we have

$$\mathrm{sgn}(g_{t,j,r}^{(t)}) = \mathrm{sgn}(-\mathrm{sgn}(j)) = -\mathrm{sgn}(j).$$

Apply Lemma C.11, for $t \in [0, \frac{n}{B}]$, we get

$$\langle \mathbf{w}_{j,r}^{(t)}, j \cdot \mathbf{v} \rangle = (1 - \lambda\eta)\langle \mathbf{w}_{j,r}^{(t-1)}, j \cdot \mathbf{v} \rangle - \Theta(\eta) \cdot \langle \mathrm{sgn}(g_{t-1,j,r}^{(t-1)}), j\mathbf{v} \rangle$$

$$\geq \langle \mathbf{w}_{j,r}^{(0)}, j \cdot \mathbf{v} \rangle + \Theta(\eta \cdot t),$$

since $\langle \mathbf{w}_{j,r}^{(0)}, j \cdot \mathbf{v} \rangle = \widetilde{\Theta}(\sigma_0)$ and $\lambda = \widetilde{O}(1)$. So we have for $t = 0$,

$$\langle \mathbf{w}_{j,r}^{\left((t+1)\cdot\frac{n}{B}\right)}, j \cdot \mathbf{v} \rangle = \langle \mathbf{w}_{j,r}^{\left(t\cdot\frac{n}{B}\right)}, j \cdot \mathbf{v} \rangle + \Theta(\eta \cdot \frac{n}{B}).$$

Now suppose the equality holds for $t = 0, \ldots, t_0$ with $t_0 \leq T_0$. We have

$$\langle \mathbf{w}_{j,r}^{\left(t_0 \cdot \frac{n}{B}\right)}, j \cdot \mathbf{v} \rangle = \langle \mathbf{w}_{j,r}^{(0)}, j \cdot \mathbf{v} \rangle + \Theta(\eta \cdot \frac{n}{B} \cdot t_0) \leq \widetilde{O}(1).$$

Since $\frac{n}{B} = \omega(s\sigma_p)$. We have

$$\langle \mathbf{w}_{j,r}^{\left(t_0 \cdot \frac{n}{B}\right)}, j \cdot \mathbf{v} \rangle = \widetilde{\Theta}(\sigma_0 + \eta \cdot \frac{n}{B} \cdot t_0),$$

$$\langle \mathbf{w}_{y_i,r}^{\left(t_0 \cdot \frac{n}{B}\right)}, \boldsymbol{\xi}_i \rangle = \widetilde{\Theta}(s^{1/2}\sigma_p\sigma_0 + \eta s \sigma_p \cdot t_0),$$

$$\langle \mathbf{w}_{j,r}^{\left(t_0 \cdot \frac{n}{B}\right)}, j \cdot \mathbf{v} \rangle \geq \langle \mathbf{w}_{y_i,r}^{\left(t_0 \cdot \frac{n}{B}\right)}, \boldsymbol{\xi}_i \rangle.$$

Therefore, for $t \in [(t_0 + 1) \cdot \frac{n}{B}, (t_0 + 2) \cdot \frac{n}{B}]$,

$$\sigma'(\langle \mathbf{w}_{j,r}^{(t)}, j\mathbf{v} \rangle) \gg \alpha\sigma'(\langle \mathbf{w}_{j,r}^{(t)}, \boldsymbol{\xi}_i \rangle).$$

Apply Lemma C.11, for $t = t_0 + 1$, we have

$$\langle \mathbf{w}_{j,r}^{\left((t+1)\cdot\frac{n}{B}\right)}, j \cdot \mathbf{v} \rangle = (1 - \lambda\eta)\langle \mathbf{w}_{j,r}^{\left((t+1)\cdot\frac{n}{B} - 1\right)}, j \cdot \mathbf{v} \rangle - \Theta(\eta) \cdot \langle \mathrm{sgn}(g_{((t+1)\cdot\frac{n}{B}-1),j,r}^{\left((t+1)\cdot\frac{n}{B} - 1\right)}), j\mathbf{v} \rangle$$

$$\geq \langle \mathbf{w}_{j,r}^{\left(t\cdot\frac{n}{B}\right)}, j \cdot \mathbf{v} \rangle + \Theta(\eta \cdot \frac{n}{B}),$$

since $\langle \mathbf{w}_{j,r}^{(t)}, j \cdot \mathbf{v} \rangle = \widetilde{O}(1)$ and $\lambda = \widetilde{O}(1)$. This completes the proof. $\qquad\square$

**Lemma C.18** (Stage II). *Given the training dataset $\mathcal{S}$, if $\frac{n}{B} \geq \Theta(n^{1/2} \vee \log \epsilon^{-1})$ and $\frac{n}{B} = \omega(s\sigma_p)$, $\eta = 1/\text{poly}(d)$, $\lambda = \widetilde{\Omega}(\frac{B^2}{n} \wedge 1)$ and $\lambda = \widetilde{O}(1)$, then for any $t > T_0$, $j \in \{\pm 1\}$, $r \in [m]$, $i \in [n]$, let $r^* = \arg\max_{r \in [m]} \langle \mathbf{w}_{j,r}^{(t)}, j\mathbf{v} \rangle$, then $\langle \mathbf{w}_{j,r^*}^{(t)}, j\mathbf{v} \rangle = \widetilde{\Theta}(1)$ and $\langle \mathbf{w}_{y_i,r}^{(t)}, \boldsymbol{\xi}_i \rangle \leq \widetilde{\Theta}(\frac{Bs\sigma_p}{n})$.*

*Proof of Lemma C.18.* By Lemma C.17, we know that $\langle \mathbf{w}_{j,r}^{(t)}, j\mathbf{v} \rangle$ increases at a faster rate than $\langle \mathbf{w}_{y_i,r}^{(t)}, \boldsymbol{\xi}_i \rangle$ since $\frac{n}{B} = \omega(s\sigma_p)$. We also have $\langle \mathbf{w}_{-y_i,r}^{(t)}, \boldsymbol{\xi}_i \rangle \in [-\widetilde{\Theta}(\eta s\sigma_p), \widetilde{\Theta}(\sigma_0)]$ following Lemma C.14.

Now suppose that $\langle \mathbf{w}_{j,r^*}^{(t)}, j\mathbf{v} \rangle \geq \left(\log \left((\lambda\eta)^{-2} - 1\right)\right)^{\frac{1}{q}}$, then for $(\mathbf{x}_i, y_i)$ with $y_i = j$,

$$
\begin{aligned}
\ell_{j,i}^{(t)} &= \frac{e^{F_{-j}(\mathbf{W}^{(t)}, \mathbf{x}_i)}}{\sum_{j \in \{-1,1\}} e^{F_j(\mathbf{W}^{(t)}, \mathbf{x}_i)}} \\
&= \frac{1}{1 + \exp\left[\sum_{r=1}^m \sigma(\langle \mathbf{w}_{j,r}^{(t)}, j\mathbf{v} \rangle) + \sigma(\langle \mathbf{w}_{j,r}^{(t)}, \boldsymbol{\xi}_i \rangle) - \sigma(\langle \mathbf{w}_{-j,r}^{(t)}, j\mathbf{v} \rangle) - \sigma(\langle \mathbf{w}_{-j,r}^{(t)}, \boldsymbol{\xi}_i \rangle)\right]} \\
&\leq \frac{1}{1 + \exp\left[\sigma(\langle \mathbf{w}_{j,r^*}^{(t)}, j\mathbf{v} \rangle)\right]} \\
&\leq \frac{1}{1 + \exp\left[(\log((\lambda\eta)^{-2} - 1))^{\frac{1}{q} \cdot q}\right]} \\
&= \Theta(\lambda^2 \eta^2),
\end{aligned}
$$

where the inequality we use $\langle \mathbf{w}_{-j,r}^{(t)}, j\mathbf{v} \rangle < 0$, $\langle \mathbf{w}_{-j,r}^{(t)}, \boldsymbol{\xi}_i \rangle) \leq \widetilde{\Theta}(\sigma_0)$. Then by Lemma C.11, C.17 and (B.5), we have

$$
\begin{aligned}
\langle \mathbf{w}_{j,r^*}^{(t+1)}, j\mathbf{v} \rangle &\leq (1 - \lambda\eta)\langle \mathbf{w}_{j,r^*}^{(t)}, j\mathbf{v} \rangle + \widetilde{\Theta}(\lambda\eta \cdot \eta) \\
&\leq \langle \mathbf{w}_{j,r^*}^{(t)}, j\mathbf{v} \rangle,
\end{aligned}
$$

since $\eta = o(1)$. Similarly, if $\langle \mathbf{w}_{y_i,r}^{(0)}, \boldsymbol{\xi}_i \rangle \geq \widetilde{\Theta}(\sigma_0)$

$$
\begin{aligned}
\langle \mathbf{w}_{y_i,r}^{(t+1)}, \boldsymbol{\xi}_i \rangle &\leq (1 - \lambda\eta)\langle \mathbf{w}_{y_i,r}^{(t)}, \boldsymbol{\xi}_i \rangle + \widetilde{\Theta}(\lambda\eta \cdot \eta s\sigma_p) \\
&\leq \langle \mathbf{w}_{y_i,r}^{(t)}, \boldsymbol{\xi}_i \rangle.
\end{aligned}
$$

Otherwise, $\langle \mathbf{w}_{y_i,r}^{(t)}, \boldsymbol{\xi}_i \rangle \in [-\widetilde{\Theta}(\sigma_0), \widetilde{\Theta}(\sigma_0)]$ and satisfies $\langle \mathbf{w}_{y_i,r}^{(t)}, \boldsymbol{\xi}_i \rangle \leq \widetilde{\Theta}(\frac{Bs\sigma_p}{n})$.

If $\langle \mathbf{w}_{j,r^*}^{(t)}, j\mathbf{v} \rangle \leq \left(\frac{1}{2m} \log \left((\lambda\eta)^{-1} - 1\right)\right)^{\frac{1}{q}}$, then for $(\mathbf{x}_i, y_i)$ with $y_i = j$,

$$
\begin{aligned}
\ell_{j,i}^{(t)} &= \frac{e^{F_{-j}(\mathbf{W}^{(t)}, \mathbf{x}_i)}}{\sum_{j \in \{-1,1\}} e^{F_j(\mathbf{W}^{(t)}, \mathbf{x}_i)}} \\
&= \frac{1}{1 + \exp\left[\sum_{r=1}^m \sigma(\langle \mathbf{w}_{j,r}^{(t)}, j\mathbf{v} \rangle) + \sigma(\langle \mathbf{w}_{j,r}^{(t)}, \boldsymbol{\xi}_i \rangle) - \sigma(\langle \mathbf{w}_{-j,r}^{(t)}, j\mathbf{v} \rangle) - \sigma(\langle \mathbf{w}_{-j,r}^{(t)}, \boldsymbol{\xi}_i \rangle)\right]} \\
&\geq \frac{1}{1 + \exp\left[2m \cdot \sigma(\langle \mathbf{w}_{j,r^*}^{(t)}, j\mathbf{v} \rangle)\right]} \\
&\geq \frac{1}{1 + \exp\left[2m \cdot \left(\frac{1}{2m} \log((\lambda\eta)^{-1} - 1)\right)^{\frac{1}{q} \cdot q}\right]} \\
&= \Theta(\lambda\eta),
\end{aligned}
$$

where the inequality we use $\langle \mathbf{w}_{-j,r}^{(t)}, j\mathbf{v} \rangle < 0$, $\langle \mathbf{w}_{-j,r}^{(t)}, \boldsymbol{\xi}_i \rangle) \leq \widetilde{\Theta}(\sigma_0)$. Then by Lemma C.11, C.17 and (B.5), we have

$$
\langle \mathbf{w}_{j,r^*}^{(t+1)}, j\mathbf{v} \rangle \geq (1 - \lambda\eta)\langle \mathbf{w}_{j,r^*}^{(t)}, j\mathbf{v} \rangle + \widetilde{\Theta}(\eta)
$$

$$\geq \langle \mathbf{w}_{j,r^*}^{(t)}, j\mathbf{v} \rangle,$$

since $\lambda = \widetilde{O}(1)$. This completes the proof. $\qquad\square$

**Lemma C.19** (Convergence). *Suppose the same conditions hold as in Lemma C.17 and C.18, if the step size satisfies $\eta = O(d^{-1/2})$, then for any t,*

$$\mathbb{E}\left[ L(\mathbf{W}^{(t+1)}) - L(\mathbf{W}^{(t)}) \right] \leq -\eta \|\nabla L(\mathbf{W}^{(t)})\|_1 + \widetilde{\Theta}(\eta^2 d).$$

*Proof of Lemma C.19.* The proof is same as Lemma C.15. We aim to prove the convergence of the objective function under the AdamW optimization algorithm in a non-convex setting. Recall the loss function for each data point $i$ is

$$L_i(\mathbf{W}) = \log \left( 1 + \frac{1}{\exp \left( F_{y_i}(\mathbf{W}, \mathbf{x}_i) - F_{-y_i}(\mathbf{W}, \mathbf{x}_i) \right)} \right),$$

where $\mathbf{W}$ represents the parameter matrix, $\mathbf{x}_i$ is the input data, $y_i$ is the true label, and $F_j(\mathbf{W}, \mathbf{x}_i)$ are the logits for class $j$. The total objective is:

$$L(\mathbf{W}) = \frac{1}{n} \sum_{i=1}^{n} L_i(\mathbf{W})$$

Since $L_i(\mathbf{W})$ is non-convex, we exploit its smoothness with respect to the logits $[F_j(\mathbf{W}, \mathbf{x}_i)]_j$. Specifically, $L_i(\mathbf{W})$ is 1-smooth in $[F_j(\mathbf{W}, \mathbf{x}_i)]_j$ due to the properties of the cross-entropy loss. Define:

$$\Delta F_{j,i} = F_j(\mathbf{W}^{(t+1)}, \mathbf{x}_i) - F_j(\mathbf{W}^{(t)}, \mathbf{x}_i).$$

Using the smoothness property, we apply a second-order Taylor-like expansion around $\mathbf{W}^{(t)}$:

$$L_i(\mathbf{W}^{(t+1)}) - L_i(\mathbf{W}^{(t)}) \leq \sum_j \frac{\partial L_i(\mathbf{W}^{(t)})}{\partial F_j(\mathbf{W}^{(t)}, \mathbf{x}_i)} \cdot \Delta F_{j,i} + \sum_j (\Delta F_{j,i})^2. \tag{C.41}$$

This upper bound arises because the second derivative of $L_i$ with respect to the logits is bounded by 1, a standard result for cross-entropy loss. The logits are defined as: $F_j(\mathbf{W}^{(t)}, \mathbf{x}_i) = \sum_{r=1}^{m} [\sigma(\langle \mathbf{w}_{j,r}^{(t)}, y_i\mathbf{v} \rangle) + \sigma(\langle \mathbf{w}_{j,r}^{(t)}, \boldsymbol{\xi}_i \rangle)]$, where $\mathbf{w}_{j,r}^{(t)}$ the $r$-th neuron in $j$-th output of $\mathbf{W}^{(t)}$, $\sigma(z) = [z]_+^q$ is a smooth activation function (e.g., with $q \geq 3$). By Lemma C.18 and C.1, we have $\langle \mathbf{w}_{j,r}^{(t)}, \mathbf{v} \rangle \leq \widetilde{\Theta}(1)$ and $\langle \mathbf{w}_{j,r}^{(t)}, \boldsymbol{\xi}_i \rangle \leq \widetilde{\Theta}(1)$, ensuring the local smoothness of $\sigma$ remains $\widetilde{O}(1)$ between $\langle \mathbf{w}_{j,r}^{(t+1)}, y_i\mathbf{v} \rangle$ and $\langle \mathbf{w}_{j,r}^{(t)}, y_i\mathbf{v} \rangle$ (similar for $\langle \mathbf{w}_{j,r}^{(t)}, \boldsymbol{\xi}_i \rangle$). Then with Taylor expansion, we have

$$\left| \sigma(\langle \mathbf{w}_{j,r}^{(t+1)}, y_i\mathbf{v} \rangle) - \sigma(\langle \mathbf{w}_{j,r}^{(t)}, y_i\mathbf{v} \rangle) - \langle \nabla_{\mathbf{w}_{j,r}} \sigma(\langle \mathbf{w}_{j,r}^{(t)}, y_i\mathbf{v} \rangle), \mathbf{w}_{j,r}^{(t+1)} - \mathbf{w}_{j,r}^{(t)} \rangle \right|$$

$$\leq \widetilde{\Theta}(1) \cdot \left\| \mathbf{w}_{j,r}^{(t+1)} - \mathbf{w}_{j,r}^{(t)} \right\|_2^2$$

$$= \widetilde{\Theta}(1) \cdot \left\| \lambda\eta \cdot \mathbf{w}_{j,r}^{(t)} + \eta \cdot \frac{\mathbf{m}_{j,r}^{(t)}}{\sqrt{\mathbf{v}_{j,r}^{(t)}} + \epsilon} \right\|_2^2$$

$$\leq \widetilde{\Theta}(\eta^2 d), \tag{C.42}$$

where the last inequality we use Lemma C.1 and $\|\mathbf{w}_{j,r}^{(t)}\|_2^2 \ll \Theta(d)$ by C.18. Similarly, we have

$$\left| \sigma(\langle \mathbf{w}_{j,r}^{(t+1)}, \boldsymbol{\xi}_i \rangle) - \sigma(\langle \mathbf{w}_{j,r}^{(t)}, \boldsymbol{\xi}_i \rangle) - \langle \nabla_{\mathbf{w}_{j,r}} \sigma(\langle \mathbf{w}_{j,r}^{(t)}, \boldsymbol{\xi}_i \rangle), \mathbf{w}_{j,r}^{(t+1)} - \mathbf{w}_{j,r}^{(t)} \rangle \right|$$

$$\leq \widetilde{\Theta}(1) \cdot \left\| \mathbf{w}_{j,r}^{(t+1)} - \mathbf{w}_{j,r}^{(t)} \right\|_2^2$$

$$\leq \widetilde{\Theta}(\eta^2 d). \tag{C.43}$$

Summing over $r$ (with $m = \widetilde{\Theta}(1)$), we get

$$|\Delta F_{j,i}| \leq \left|\langle \nabla_{\mathbf{W}} F_j(\mathbf{W}^{(t)}, \mathbf{x}_i), \mathbf{W}^{(t+1)} - \mathbf{W}^{(t)}\rangle\right| + \widetilde{\Theta}(\eta^2 d). \tag{C.44}$$

Additionally, $\|\nabla_{\mathbf{W}} F_j(\mathbf{W}^{(t)}, \mathbf{x}_i)\|_F \leq \widetilde{\Theta}(1)$ since $m = \widetilde{\Theta}(1)$, $\langle \mathbf{w}_{j,r}^{(t)}, y_i \mathbf{v}\rangle \leq \widetilde{\Theta}(1)$, $\langle \mathbf{w}_{j,r}^{(t)}, \boldsymbol{\xi}_i\rangle \leq \widetilde{\Theta}(1)$. So we have

$$|\Delta F_{j,i}| \leq \widetilde{\Theta}(\eta s \sigma_p + \eta \alpha + \eta + \eta^2 d) \leq \widetilde{\Theta}(\eta s \sigma_p + \eta^2 d). \tag{C.45}$$

Substitute (C.44) and (C.45) into (C.41):

$$L_i(\mathbf{W}^{(t+1)}) - L_i(\mathbf{W}^{(t)}) \leq \langle \nabla_{\mathbf{W}} L_i(\mathbf{W}^{(t)}), \mathbf{W}^{(t+1)} - \mathbf{W}^{(t)}\rangle + \widetilde{\Theta}(\eta^2 d). \tag{C.46}$$

For the full objective:

$$L(\mathbf{W}^{(t+1)}) - L(\mathbf{W}^{(t)}) = \frac{1}{n} \sum_{i=1}^n [L_i(\mathbf{W}^{(t+1)}) - L_i(\mathbf{W}^{(t)})]. \tag{C.47}$$

Substitute (C.46) into (C.47), we have

$$L(\mathbf{W}^{(t+1)}) - L(\mathbf{W}^{(t)}) \leq \langle \nabla L(\mathbf{W}^{(t)}), \mathbf{W}^{(t+1)} - \mathbf{W}^{(t)}\rangle + \widetilde{\Theta}(\eta^2 d). \tag{C.48}$$

Take expectation for the stochastic gradient of both side in (C.48),

$$\mathbb{E}\left[L(\mathbf{W}^{(t+1)}) - L(\mathbf{W}^{(t)})\right]$$

$$\leq \mathbb{E}\left[\langle \nabla L(\mathbf{W}^{(t)}), \mathbf{W}^{(t+1)} - \mathbf{W}^{(t)}\rangle\right] + \widetilde{\Theta}(\eta^2 d)$$

$$\leq -\eta \cdot \mathbb{E}\left[\sum_{j \in \{\pm 1\}} \sum_{r \in [m]} \left\|g_{t,j,r}^{(t)}\right\|_1\right] - \lambda\eta \cdot \langle \nabla L(\mathbf{W}^{(t)}), \mathbf{W}^{(t)}\rangle + \widetilde{\Theta}(d \cdot \eta^2) + \widetilde{\Theta}(ns \cdot \eta^2 s \sigma_p) + \widetilde{\Theta}(\eta^2 d)$$

$$\leq -\eta \cdot \sum_{j \in \{\pm 1\}} \sum_{r \in [m]} \left\|\mathbb{E}\left[g_{t,j,r}^{(t)}\right]\right\|_1 + \widetilde{O}(\lambda\eta) \cdot \|\nabla L(\mathbf{W}^{(t)})\|_1 + \widetilde{\Theta}(\eta^2 d)$$

$$\leq -\eta \|\nabla L(\mathbf{W}^{(t)})\|_1 + \widetilde{\Theta}(\eta^2 d),$$

where we use Lemma C.11 that the update aligns with the gradient's sign for large gradient and the fact that $ns^2\sigma_p = O(d)$, and Jensen's inequality, Hölder's inequality and $\lambda = \widetilde{O}(1)$, $\|\mathbf{W}^{(t)}\|_\infty \leq \widetilde{\Theta}(1)$ by Lemma C.18. This completes the proof. □

**Lemma C.20** (Generalization of Stochastic AdamW, mini-batch). *Suppose the same conditions hold as in Lemma C.19. We have the following results for $T = \frac{\mathrm{poly}(n)}{\eta}$, with training dataset $\mathcal{S}$*

- *The training error is zero:* $\mathrm{err}_{\mathcal{S}}(\mathbf{W}^{(T)}) = 0$.

- *The test error is near-zero:* $\mathrm{err}_{\mathcal{D}}(\mathbf{W}^{(T)}) = o(1)$.

*Proof of Lemma C.20.* By Lemma C.18, we have

$$\langle \mathbf{w}_{j,r^*}^{(T)}, j\mathbf{v}\rangle = \widetilde{\Theta}(1), \quad \langle \mathbf{w}_{y_i,r}^{(T)}, \boldsymbol{\xi}_i\rangle \leq \widetilde{\Theta}(\frac{Bs\sigma_p}{n}), \quad \langle \mathbf{w}_{-y_i,r}^{(T)}, \boldsymbol{\xi}_i\rangle \leq \widetilde{\Theta}(\sigma_0).$$

Recall $F_j(\mathbf{W}, \mathbf{x})$ in Definition 3.2, with $\alpha = o(1)$, we directly have

$$\mathrm{err}_{\mathcal{S}}(\mathbf{W}^{(T)}) = \mathbb{E}_{(\mathbf{x},y)\sim\mathcal{S}} \mathbb{1}\left[F_y(\mathbf{W}^{(T)}, \mathbf{x}) \leq F_{-y}(\mathbf{W}^{(T)}, \mathbf{x})\right] = 0,$$

since $F_{y_i}(\mathbf{W}^{(T)}, \mathbf{x}_i) = \widetilde{\Omega}(1)$, while $F_{-y_i}(\mathbf{W}^{(T)}, \mathbf{x}_i) \leq \widetilde{\Theta}(\sigma_0 + \alpha)$.

For test data $(\mathbf{x}, y) \sim \mathcal{D}$ with $\mathbf{x} = [y\mathbf{v}^\top, \boldsymbol{\xi}^\top]^\top$, it is clear that with high probability $\langle \mathbf{w}_{y,r^*}^{(T)}, y\mathbf{v}\rangle = \widetilde{\Theta}(1)$. Let $\mathcal{B} = \mathrm{supp}(\boldsymbol{\xi})$, $\|\mathbf{w}_{\mathcal{B}}\|_2^2 = \sum_{k\in\mathcal{B}} \mathbf{w}_{y,r}[k]^2$, $\zeta_{y,r} = \sum_{k\in\mathcal{B}} \mathbf{w}_{y,r}[k]\cdot\boldsymbol{\xi}[k] \sim \mathcal{N}(0, \|\mathbf{w}_{\mathcal{B}}\|_2^2\cdot\sigma_p^2)$, then we have

$$\langle \mathbf{w}_{y,r}^{(T)}, \boldsymbol{\xi}\rangle \leq \zeta_{y,r}.$$

Now we calculate the upper bound of $\|\mathbf{w}_{\mathcal{B}}\|_2^2$. By Lemma A.4, we know $\langle \boldsymbol{\xi}_i, \boldsymbol{\xi}_j \rangle = 0$ for $i \neq j$, $i, j \in [n]$. Then let

$$\Sigma = \sum_{i=1}^{n} \boldsymbol{\xi}_i \boldsymbol{\xi}_i^\top, \quad S = \text{span}\{\boldsymbol{\xi}_1, \dots, \boldsymbol{\xi}_n\}.$$

We have

$$\mathbf{w}_{j,r}^{(T)} = P_S \mathbf{w}_{j,r}^{(T)} + \mathbf{r} + c\mathbf{v}, \quad \mathbf{r} \perp S \cup \{\mathbf{v}\},$$

where $c = \widetilde{\Theta}(1)$. By Lemma C.18, we have

$$\lambda_{\min}^+(\Sigma) \cdot \|P_S \mathbf{w}_{j,r}^{(T)}\|_2^2 \leq (P_S \mathbf{w}_{j,r}^{(T)})^\top \Sigma (P_S \mathbf{w}_{j,r}^{(T)}) = \sum_{i=1}^{n} \langle \mathbf{w}_{j,r}^{(T)}, \boldsymbol{\xi}_i \rangle^2 = \widetilde{O}\left(\frac{B^2 s}{n}\right).$$

Since $\|\boldsymbol{\xi}_i\|_2^2 \sim \sigma_p^2 \chi_s^2$ and $s = \omega(\log n)$, we have $\|\boldsymbol{\xi}_i\|_2^2 = \widetilde{\Theta}(s\sigma_p^2)$ with high probability. Hence, $\lambda_{\min}^+(\Sigma) = \min_i \|\boldsymbol{\xi}_i\|_2^2 = \widetilde{\Theta}(s\sigma_p^2)$. With a little abuse of notation, we have

$$\|P_S \mathbf{w}_{j,r}^{(T)}\|_2^2 \leq \widetilde{\Theta}\left(\frac{B^2}{n\sigma_p^2}\right).$$

By Lemma C.11, $\lambda = \widetilde{\Omega}(\frac{B^2}{n} \wedge 1)$ and $\lambda \eta T = \omega(1)$, we have

$$\|\mathbf{r}\|_2^2 \leq \widetilde{\Theta}(\eta^2 d).$$

So the upper bound of $\|\mathbf{w}_{\mathcal{B}}\|_2^2$ is

$$\|\mathbf{w}_{\mathcal{B}}\|_2^2 \leq \|P_S \mathbf{w}_{j,r}^{(T)}\|_2^2 + \|\mathbf{r}\|_2^2 \leq \widetilde{\Theta}\left(\frac{B^2}{n\sigma_p^2}\right).$$

Finally, with high probability

$$\zeta_{y,r} \leq \widetilde{\Theta}\left(\sqrt{\frac{B^2}{n\sigma_p^2}} \cdot \sigma_p\right) = o(1),$$

since $\frac{n}{B} = \omega(n^{1/2})$. The same result holds for $\zeta_{-y,r}$. Then, we have

$$F_y(\mathbf{W}^{(T)}, \mathbf{x}) \geq \sigma(\langle \mathbf{w}_{y,r^*}^{(T)}, y\mathbf{v} \rangle) = \widetilde{\Omega}(1),$$

while

$$F_{-y}(\mathbf{W}^{(T)}, \mathbf{x}) = \sum_{r=1}^{m} \left[\sigma(\langle \mathbf{w}_{-y,r}^{(T)}, y\mathbf{v} \rangle) + \sigma(\langle \mathbf{w}_{-y,r}^{(T)}, \boldsymbol{\xi} \rangle)\right] = m \cdot [\alpha + \zeta_{-y,r}]_+^q = o(1).$$

Therefore, we have

$$\text{err}_{\mathcal{D}}(\mathbf{W}^{(T)}) = \mathbb{E}_{(\mathbf{x},y)\sim\mathcal{D}} \mathbb{1}\left[F_y(\mathbf{W}^{(T)}, \mathbf{x}) \leq F_{-y}(\mathbf{W}^{(T)}, \mathbf{x})\right] = o(1).$$

This implies that mini-batch AdamW can achieve nearly zero test error. This completes the proof. $\square$

### C.2.3    Proof of Corollary 4.6

By Conditions A.1 and A.2, along with Definition 3.2, we know that $d = \text{poly}(n)$, and hence

$$\sigma_0^{q-2} = \Theta\left(\frac{1}{d^{(q-2)/4}}\right), \quad \text{with } q \geq 3.$$

This directly implies that the effective weight decay parameter for Adam satisfies

$$\lambda_{\text{Adam}} \sim \sigma_0^{q-2} \ll \min\left\{\frac{B^2}{n}, 1\right\} \sim \lambda_{\text{AdamW}}.$$

This completes the proof.

## D  Experimental details and results

This section presents the complete details of our experiments.

### D.1  Experimental Details for Real-world Data

For the real-world experiments in Figures 1 and 2, we use the CIFAR-10 dataset, VGG16 and ResNet18 architectures, and the Adam and AdamW optimizers, all implemented in PyTorch. We do not use data augmentation in order to avoid any additional regularization effects.

In Figure 1, we report the test error as a function of batch size. The batch sizes considered are $\{16, 32, 64, 256, 1024, 4096, 8192\}$, with training conducted for 100 epochs. The weight decay is set to $5 \times 10^{-4}$ for Adam and $1 \times 10^{-2}$ for AdamW; the momentum parameters are fixed at $(\beta_1, \beta_2) = (0.9, 0.99)$ for both optimizers. Each configuration is evaluated with three learning rates: $\{5 \times 10^{-4}, 1 \times 10^{-4}, 1 \times 10^{-5}\}$, and we report the best test performance for each batch size. All experiments can be run within one hour on a single RTX 4090 GPU. The only exception is training ResNet18 with a batch size of 8192, which requires three GPUs due to memory constraints.

Figure 1(a) presents the test error versus batch size for Adam with VGG16 and ResNet18, while Figure 1(b) shows the corresponding results for AdamW. Both demonstrate that test performance degrades as batch size increases, which is consistent with our theoretical findings in Section 4, showing that small-batch Adam and AdamW outperform their large-batch counterparts.

In Figure 2, we report the test error as a function of weight decay $\lambda$ for Adam and AdamW, using VGG16 (Figure 2(a)) and ResNet18 (Figure 2(b)). We fix the batch size to 16, the learning rate to $1 \times 10^{-4}$, and set $(\beta_1, \beta_2) = (0.9, 0.99)$. The weight decay values for Adam are $\{1 \times 10^{-1}, 5 \times 10^{-2}, 1 \times 10^{-2}, 5 \times 10^{-3}, 1 \times 10^{-3}, 5 \times 10^{-4}, 1 \times 10^{-4}, 5 \times 10^{-5}, 1 \times 10^{-5}, 5 \times 10^{-6}, 1 \times 10^{-6}, 5 \times 10^{-7}\}$, and for AdamW are $\{5 \times 10^{-1}, 1 \times 10^{-1}, 5 \times 10^{-2}, 1 \times 10^{-2}, 5 \times 10^{-3}, 1 \times 10^{-3}, 5 \times 10^{-4}, 1 \times 10^{-4}\}$. All models are trained for 100 epochs.

Figure 2(a) shows results for training VGG16, and Figure 2(b) for ResNet18, both using Adam and AdamW. For a fair comparison, we scale the weight decay $\lambda$ of AdamW by a factor of $1/25$. The results show that Adam suffers from poor generalization under large weight decay values (e.g., $\lambda > 0.05$), while AdamW maintains stable performance even with larger weight decays (e.g., $\lambda = 0.5$), which aligns with our theoretical results in Section 4.

### D.2  Experimental Details for Synthetic Data

For the data model defined in Definition 3.1, we set the input dimension to $d = 1000$ and the number of training samples to $n = 200$, consisting of 100 positive and 100 negative samples. The sparsity level is set to $s = 0.1d = 100$, and the noise strength is $\sigma_p = 1/\sqrt{s} = 0.1$. The feature noise strength is set to $\alpha = 0.2$, and the model weights are initialized with standard deviation $\sigma_0 = 0.01$. The network, defined in Definition 3.2, has width $m = 20$.

All synthetic experiments are trained for $T = 10^4$ epochs with a learning rate of $\eta = 5 \times 10^{-5}$, and evaluated on a test dataset of size $10^4$. For Adam and AdamW optimizers, we adopt the default momentum hyperparameters $\beta_1 = 0.9$ and $\beta_2 = 0.999$.

We primarily focus on the following metrics:

- Training error: $\mathrm{err}_{\mathcal{S}}(\mathbf{W})$.
- Test error: $\mathrm{err}_{\mathcal{D}}(\mathbf{W})$.
- Feature learning: $\max_{r \in [m]} \langle \mathbf{w}_{j,r}, j\mathbf{v} \rangle$.
- Noise memorization: $\min_{i \in [n]:y_i=j} \max_{r \in [m]} \langle \mathbf{w}_{j,r}, \boldsymbol{\xi}_i \rangle$ or $\max_{i \in [n]:y_i=j} \max_{r \in [m]} \langle \mathbf{w}_{j,r}, \boldsymbol{\xi}_i \rangle$.

**Large-batch Adam vs. Mini-batch Adam.**  We set $\lambda = 1 \times 10^{-5}$ for both large-batch Adam (batch size $B = 100$) and mini-batch Adam (batch size $B = 2$). Table 1 presents the training and test errors of the solutions obtained by the two training methods. Although both large-batch and mini-batch Adam achieve zero training error, their generalization performance differs significantly. Specifically, large-batch Adam suffers from high test error (greater than 0.5), while mini-batch Adam achieves zero test error. This observation verifies Theorems 4.1 and 4.2.

Table 1: Training and test errors of Adam with large ($B = 100$) and mini-batch ($B = 2$) settings.

| Batch size | $B = 100$ | $B = 2$ |
|---|---|---|
| Training error | 0 | 0 |
| Test error | 0.9545 | 0 |

(a) Large-batch Adam ($B = 100$)  (b) Mini-batch Adam ($B = 2$)

Figure 3: Feature learning and noise memorization of Adam in the training.

Moreover, Figure 3(a) illustrates the dynamics of feature learning, measured by $\max_{r \in [m]} \langle \mathbf{w}_{j,r}, j\mathbf{v} \rangle$, and noise memorization, measured by $\min_{i \in [n]:y_i=j} \max_{r \in [m]} \langle \mathbf{w}_{j,r}, \boldsymbol{\xi}_i \rangle$, under large-batch Adam. The results are consistent with Figure 2 in Zou et al. (2023b). Figure 3(b) shows the corresponding dynamics for mini-batch Adam, where feature learning $\max_{r \in [m]} \langle \mathbf{w}_{j,r}, j\mathbf{v} \rangle$ increases steadily, while noise memorization $\max_{i \in [n]:y_i=j} \max_{r \in [m]} \langle \mathbf{w}_{j,r}, \boldsymbol{\xi}_i \rangle$ remains suppressed at the end of Pattern Learning Stage. In the subsequent Regularization Stage, feature learning saturates at a stable threshold and stops increasing. This behavior is consistent with Lemma C.7.

**Large-batch AdamW vs. Mini-batch AdamW.** We set $\lambda = 0.01$ for both large-batch AdamW (batch size $B = 100$) and mini-batch AdamW (batch size $B = 2$). Table 2 reports the training and test errors for both training methods. Although both large-batch and mini-batch AdamW achieve zero training error, their test performance differs significantly: large-batch AdamW suffers from high test error (exceeding 0.5), while mini-batch AdamW attains zero test error. This observation supports Theorems 4.4 and 4.5.

Figure 4(a) illustrates the dynamics of feature learning, measured by $\max_{r \in [m]} \langle \mathbf{w}_{j,r}, j\mathbf{v} \rangle$, and noise memorization, measured by $\min_{i \in [n]:y_i=j} \max_{r \in [m]} \langle \mathbf{w}_{j,r}, \boldsymbol{\xi}_i \rangle$, under large-batch AdamW. Initially, feature learning increases, but it is eventually flipped by noise memorization, which grows at a faster rate. As a result, the model begins fitting to the feature noise, which is negatively aligned with the true feature direction. Specifically, noise memorization increases rapidly during the Pattern Learning Stage and saturates at a logarithmic rate in the Regularization Stage. These behaviors are consistent with Lemmas C.12, C.13, and C.14.

Figure 4(b) shows the corresponding dynamics for mini-batch AdamW. Feature learning increases steadily and remains unaffected by noise memorization during the Pattern Learning Stage. In the Regularization Stage, feature learning saturates at a stable threshold, which causes the gradient to become small and consequently suppresses further growth of noise memorization (recall that $\boldsymbol{\xi}_i[1] = -\alpha y_i$). This behavior is consistent with Lemmas C.17 and C.18.

**Large weight decay regularization $\lambda$ hinders Adam training.** We repeat the experiments from **Large-batch Adam vs. Mini-batch Adam** using a larger weight decay parameter $\lambda = 0.05$, and

Table 2: Training and test errors of AdamW with large ($B = 100$) and mini-batch ($B = 2$) settings.

| Batch size | $B = 100$ | $B = 2$ |
|---|---|---|
| Training error | 0 | 0 |
| Test error | 0.5485 | 0 |

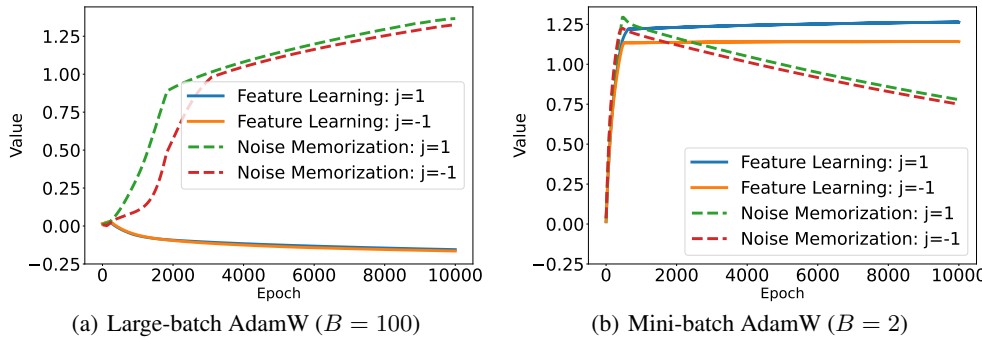

(a) Large-batch AdamW ($B = 100$)  (b) Mini-batch AdamW ($B = 2$)

Figure 4: Feature learning and noise memorization of AdamW in the training.

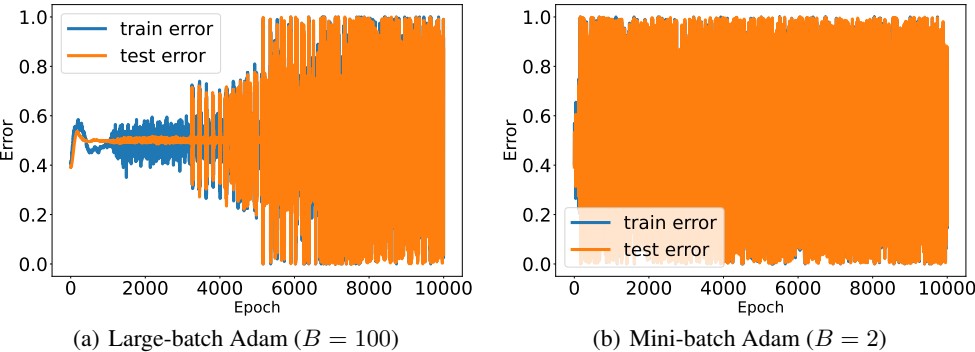

(a) Large-batch Adam ($B = 100$)  (b) Mini-batch Adam ($B = 2$)

Figure 5: Training error and test error over epochs of Adam training with $\lambda = 0.05$.

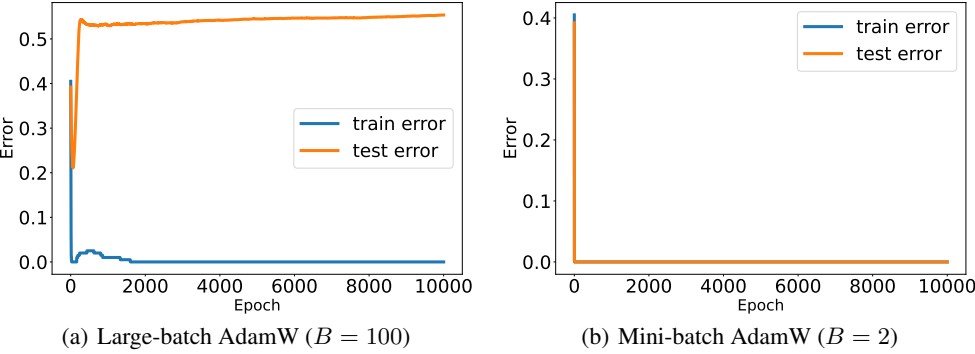

(a) Large-batch AdamW ($B = 100$)  (b) Mini-batch AdamW ($B = 2$)

Figure 6: Training error and test error over epochs of AdamW training with $\lambda = 0.5$.

those from **Large-batch AdamW vs. Mini-batch AdamW** with $\lambda = 0.5$. Figure 5 reports the training accuracy over epochs for Adam, while Figure 6 shows the same for AdamW. It can be observed that Adam fails to train under large weight decay. In contrast, AdamW remains robust and achieves results consistent with those in **Large-batch AdamW vs. Mini-batch AdamW**, even with a larger $\lambda = 0.5$. These results support Corollaries 4.3 and 4.6.

### D.3   Additional Experimental Results

**Error bars across random seeds, Figures 7 and 8.**   We provide additional results to support our theoretical findings. To assess statistical significance, we repeat the CIFAR-10 experiments from Figures 1 and 2 with five random seeds (0–4), using the same settings as in Section D.1. Figures 7 and 8 report the results, with error bars denoting the standard deviation across runs. The results confirm that both Adam and AdamW degrade in performance as the batch size increases, and that Adam is more sensitive to weight decay than AdamW.

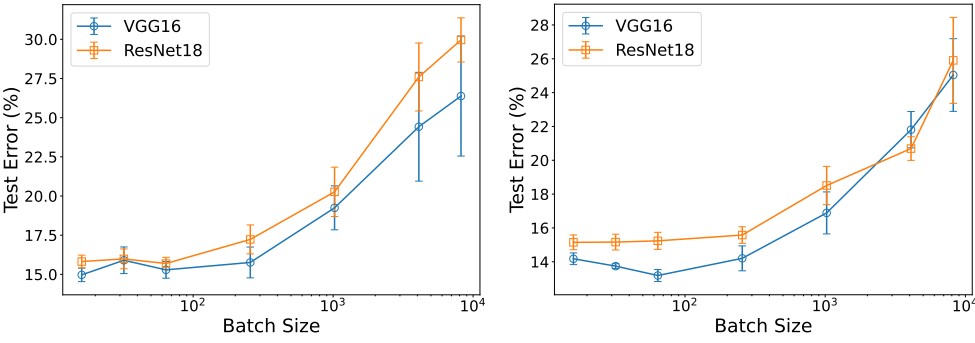

(a) Test error vs. batch size under Adam     (b) Test error vs. batch size under AdamW

Figure 7: Error bars across seeds: Test error vs. batch size for VGG16 and ResNet18 on CIFAR-10.

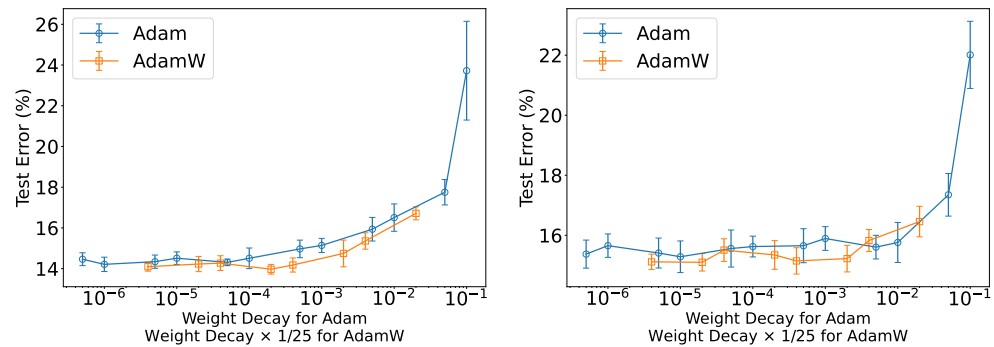

(a) VGG16                                      (b) ResNet18

Figure 8: Error bars across seeds: Test error vs. weight decay (batch size = 16), comparing Adam and AdamW.

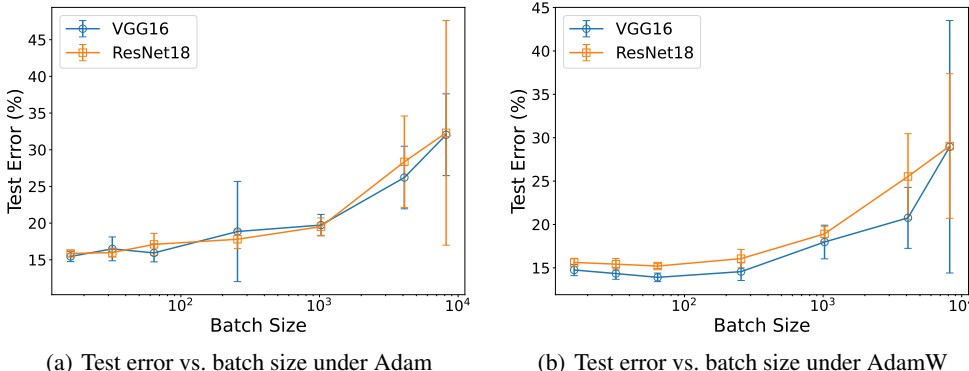

(a) Test error vs. batch size under Adam     (b) Test error vs. batch size under AdamW

Figure 9: Error bars across $(\beta_1, \beta_2)$: Test error vs. batch size for VGG16 and ResNet18 on CIFAR-10.

**Sensitivity to momentum parameters** $(\beta_1, \beta_2)$**, Figures 9 and 10.** We further study the sensitivity of Adam and AdamW to the momentum parameters $(\beta_1, \beta_2)$, which are treated as constants in our theory. We sweep over $\beta_1 \in \{0, 0.5, 0.9\}$ and $\beta_2 \in \{0.5, 0.9, 0.95\}$, yielding 8 valid combinations under $\beta_1^2 < \beta_2$, plus the standard setting $(\beta_1, \beta_2) = (0.9, 0.99)$, for a total of 9 configurations. Figures 9 and 10 report the results, with error bars showing the standard deviation across the 9 runs. The findings again confirm that both Adam and AdamW suffer performance degradation as batch size increases, and that Adam is more sensitive to weight decay than AdamW.

**Large-scale vision experiments with ResNet-50 on ImageNet-1K subset, Figures 11 and 12.** To further validate our theory, we conduct large-scale experiments on ImageNet-1K. We construct a subset by randomly sampling 100 training images per class (seed=0), ensuring a controlled large-batch regime ($\frac{n}{B} = \Theta(1)$) while keeping computation feasible. ResNet-50 is trained for 90 epochs

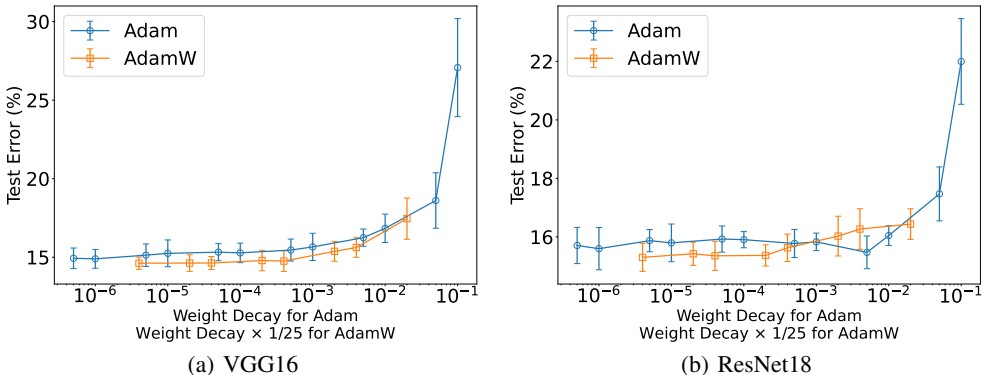

(a) VGG16  (b) ResNet18

Figure 10: Error bars across $(\beta_1, \beta_2)$: Test error vs. weight decay (batch size = 16), comparing Adam and AdamW.

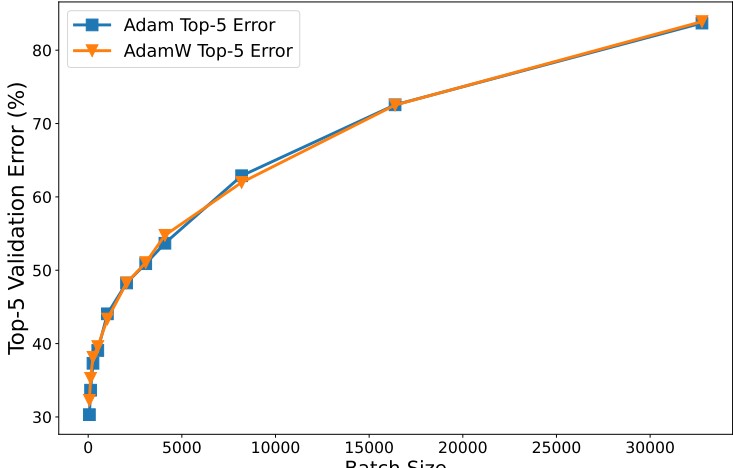

Figure 11: ImageNet-1K subset: Top-5 validation error vs. batch size for Adam and AdamW with ResNet-50.

with standard ImageNet preprocessing. We report top-5 validation error against batch size (Figure 11) and weight decay (Figure 12), comparing Adam and AdamW.

For Figure 11, we set learning rate $\eta = 1 \times 10^{-4}$, $(\beta_1, \beta_2) = (0.9, 0.99)$, and weight decay $\lambda = 1 \times 10^{-4}$ for Adam and $\lambda = 1 \times 10^{-2}$ for AdamW. Batch sizes are $\{64, 128, 256, 512, 1024, 2048, 3072, 4096, 8192, 16384, 32768\}$.

For Figure 12, we fix $B = 64$, $\eta = 1 \times 10^{-3}$, and $(\beta_1, \beta_2) = (0.9, 0.99)$. Weight decay values for Adam are $\{5 \times 10^{-7}, 1 \times 10^{-6}, 5 \times 10^{-6}, 1 \times 10^{-5}, 5 \times 10^{-5}, 1 \times 10^{-4}, 5 \times 10^{-4}, 1 \times 10^{-3}, 5 \times 10^{-3}, 1 \times 10^{-2}, 5 \times 10^{-2}, 1 \times 10^{-1}\}$, and for AdamW are $\{1 \times 10^{-4}, 5 \times 10^{-4}, 1 \times 10^{-3}, 5 \times 10^{-3}, 1 \times 10^{-2}, 5 \times 10^{-2}, 1 \times 10^{-1}, 5 \times 10^{-1}\}$.

The results again confirm that both optimizers degrade as batch size increases, and that Adam is more sensitive to weight decay than AdamW.

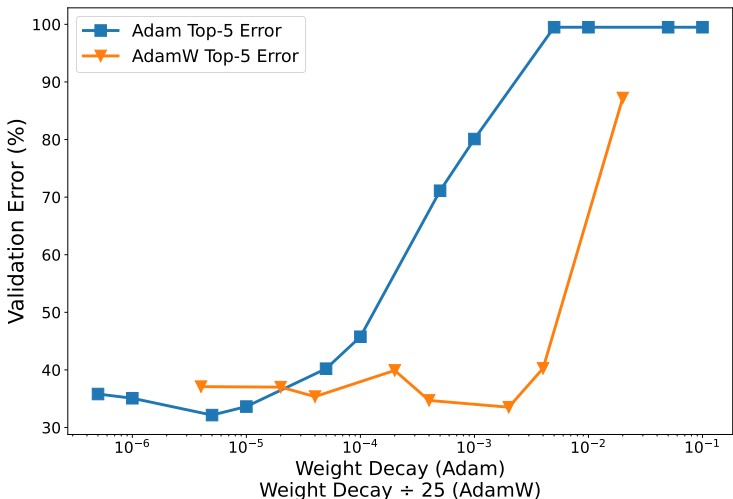

Figure 12: ImageNet-1K subset: Top-5 validation error vs. weight decay for Adam and AdamW with ResNet-50 (batch size = 64).

