# OpenReview forum: "Understanding the Generalization of Stochastic Gradient Adam in Learning Neural Networks"
_NeurIPS.cc/2025/Conference — NeurIPS 2025 poster_

### Official Review · Reviewer_qgrL · 2025-06-30

**Clarity:** 3
**Significance:** 2
**Originality:** 2
**Rating:** 5
**Confidence:** 3

**Summary:**

This paper performs a theoretical analysis of the training dynamics of Adam and AdamW on a two-layers CNN trained on a 1 dimensional synthetic dataset. In that particular example, the authors are able to show analytically that mini-batch Adam converges to lower test-error solutions than large-batch Adam. They are also able to show in that setting that AdamW admits a larger beneficial weight decay regularization range than Adam. They show experimentally that their finding hold on CIFAR10 trained with VGG16 and ResNet18.

**Questions:**

* In Definition 3.1, condition 2 in unclear. From skimming through the literature mentioned, I was able to understand that one wants the noise vector to be perpendicular to the signal vector. Is that what is meant in condition 2?

* I am struggling to understand the significance of Corollary 4.6. Is there a way to rephrase it to make its significance clearer?

**Ethical Concerns:**

["NO or VERY MINOR ethics concerns only"]

**Final Justification:**

Thank you for your detailed answers. I raised my score in consequence.

**Limitations:**

Limitations are acknowledged in the conclusion concerning the validity of the analysis to the toy model. However, some wordings sound like overstatements in this context at the same time. For instance in the conclusion we find: "In this work, we theoretically and empirically analyze the generalization performance of Adam and AdamW under varying batch sizes and weight decay parameters" although this is only done on a toy example and not in general, as the sentence may let the reader think.

**Paper Formatting Concerns:**

No formatting concerns

**Quality:**

2

**Strengths And Weaknesses:**

**Strengths:**

* The author were able to carry a fully theoretical analysis of the Adam and AdamW dynamics where the generalization advantage of mini-batch over large-batch becomes apparent
* The presentation is clear

**Weaknesses:**

* Since the theoretical claims are made in the context of a toy example, expanded experimental verification would give more credibility to the claims, and in particular experiments demonstrating the two learning stages identified (stage 1: pattern learning, and stage 2: regularization)
* Error bars on the experiment figures are missing; it seems only 1 seed has been used
* The proofs rely very much on the special structure of the toy example (including the special form of the dataset); this casts some doubt on the general validity of this type of arguments (even in the basic context of a DNN on a generic dataset). A discussion on how obstacles in this direction may be lifted would strengthen the article in my opinion. At present, it is possible to understand the theoretical results presented as being only due to non-generalizable artifacts of the toy model and datasets considered
* A Discussion/comparison with a recent line of work explaining the implicit bias of Adam toward flat and generalizable solutions in the case of fully general models and datasets is missing; in particular: https://arxiv.org/abs/2309.00079
* The language at times sounds like overstatements especially in the abstract, intro, and conclusion where the results may sound like they are proven for the general case, although they are valid only in the case of the toy example considered. Although the limitations are acknowledged, I would tone down the claims slightly

---

> ### Author Rebuttal · Authors · 2025-07-31
>
> We thank the reviewer for their concise and accurate summary of our work, and for highlighting the fully theoretical analysis revealing the generalization benefit of mini-batch Adam, as well as the broader weight‐decay range of AdamW. We’re pleased the presentation was clear. Below, we address your questions in detail.
>
> ### Weaknesses
>
> **a) Since the theoretical claims are made in the context of a toy example, expanded experimental verification would give more credibility to the claims, and in particular experiments demonstrating the two learning stages identified (stage 1: pattern learning, and stage 2: regularization)**
>
> Good point! These experiments are already included in Figures 3 and 4 (Appendix, pp. 21–22), and we will add clearer references to them in the main text.
>
> Figures 3 and 4 (Appendix, pp. 21–22) plot the dynamics of feature learning vs. noise memorization for both mini-batch and large-batch Adam/AdamW, directly visualizing the two-stage process identified by our theory (see Appendix A.2).
>
> In Figure 4(a), large-batch AdamW initially learns the true feature (measured by $\max_{r \in [m]} \langle w_{j,r}, jv \rangle$), but is quickly overtaken by noise memorization (measured by $\min_{i \in [n]: y_i = j} \max_{r \in [m]} \langle w_{j,r}, xi_i \rangle$), leading to alignment with the spurious direction $-\alpha yv$. In Stage 2 (Regularization), noise memorization grows at a logarithmic rate, matching our theoretical predictions.
>
> Figure 4(b) shows that mini-batch AdamW maintains stable feature learning throughout, while noise memorization is suppressed in Stage 2 due to vanishing gradients. These trends align closely with our lemmas and provide direct empirical support for our two-stage analysis.
>
> **b) Error bars on the experiment figures are missing; it seems only 1 seed has been used**
>
> This is an important point for ensuring experimental statistical significance. We have re-run our CIFAR-10 experiments with five random seeds (0–4), using the best hyperparameters selected based on seed 0. We will include error bars in the next version and provide full result tables (mean ± std) below. These results support and confirm our original claims.
>
> **Corresponding to Figure 1.**
>
> Adam: Test Error (%) vs. Batch Size ($B$)
>
> | Model    | 16         | 32         | 64         | 256        | 1024       | 4096       | 8192       |
> | -------- | ---------- | ---------- | ---------- | ---------- | ---------- | ---------- | ---------- |
> | VGG16    | 14.97±0.43 | 15.90±0.85 | 15.29±0.53 | 15.76±0.98 | 19.25±1.40 | 24.42±3.47 | 26.38±3.83 |
> | ResNet18 | 15.82±0.41 | 15.99±0.63 | 15.69±0.40 | 17.23±0.92 | 20.27±1.57 | 27.60±2.17 | 29.96±1.41 |
>
> AdamW: Test Error (%) vs. Batch Size ($B$)
>
> | Model    | 16         | 32         | 64         | 256        | 1024       | 4096       | 8192       |
> | -------- | ---------- | ---------- | ---------- | ---------- | ---------- | ---------- | ---------- |
> | VGG16    | 14.18±0.35 | 13.74±0.14 | 13.19±0.36 | 14.20±0.74 | 16.89±1.24 | 21.81±1.07 | 25.04±2.15 |
> | ResNet18 | 15.15±0.44 | 15.16±0.47 | 15.24±0.50 | 15.58±0.49 | 18.51±1.13 | 20.69±0.70 | 25.91±2.54 |
>
> **Corresponding to Figure 2.**
>
> ResNet18: Test Error (%) vs. Weight Decay ($\lambda$)
>
> | Optimizer | 5e-7       | 1e-6       | 5e-6       | 1e-5       | 5e-5       | 1e-4       | 5e-4       | 1e-3       | 5e-3       | 1e-2       | 5e-2       | 1e-1       | 5e-1       |
> | --------- | ---------- | ---------- | ---------- | ---------- | ---------- | ---------- | ---------- | ---------- | ---------- | ---------- | ---------- | ---------- | ---------- |
> | Adam      | 15.37±0.47 | 15.66±0.39 | 15.41±0.50 | 15.29±0.53 | 15.56±0.62 | 15.63±0.35 | 15.65±0.57 | 15.89±0.39 | 15.61±0.39 | 15.76±0.67 | 17.35±0.71 | 22.01±1.12 | —          |
> | AdamW     | —          | —          | —          | —          | —          | 15.12±0.26 | 15.10±0.29 | 15.51±0.38 | 15.35±0.48 | 15.15±0.44 | 15.22±0.44 | 15.83±0.37 | 16.46±0.51 |
>
> VGG16: Test Error (%) vs. Weight Decay ($\lambda$)
>
> | Optimizer | 5e-7       | 1e-6       | 5e-6       | 1e-5       | 5e-5       | 1e-4       | 5e-4       | 1e-3       | 5e-3       | 1e-2       | 5e-2       | 1e-1       | 5e-1       |
> | --------- | ---------- | ---------- | ---------- | ---------- | ---------- | ---------- | ---------- | ---------- | ---------- | ---------- | ---------- | ---------- | ---------- |
> | Adam      | 14.46±0.32 | 14.21±0.35 | 14.34±0.32 | 14.51±0.31 | 14.31±0.16 | 14.51±0.51 | 14.97±0.43 | 15.14±0.35 | 15.94±0.58 | 16.51±0.67 | 17.75±0.62 | 23.72±2.43 | —          |
> | AdamW     | —          | —          | —          | —          | —          | 14.10±0.22 | 14.23±0.37 | 14.27±0.36 | 13.97±0.24 | 14.18±0.35 | 14.75±0.66 | 15.36±0.40 | 16.72±0.32 |
>
> **c) The proofs rely very much on the special structure of the toy example (including the special form of the dataset); this casts some doubt on the general validity of this type of arguments (even in the basic context of a DNN on a generic dataset). A discussion on how obstacles in this direction may be lifted would strengthen the article in my opinion. At present, it is possible to understand the theoretical results presented as being only due to non-generalizable artifacts of the toy model and datasets considered**
>
> We appreciate the concern that our toy model may not fully capture real-world complexity. However, such simplified settings are standard in theoretical deep learning (Lines 135–136). Our feature–noise patch abstraction isolates the tradeoff between learning sparse, task-relevant signal and memorizing irrelevant noise—reflecting empirical patterns in CNNs (Papyan et al., 2017; Yang, 2019). Interpreting our two-layer CNN as modeling an intermediate layer in deeper networks further supports its relevance. Experiments confirm that the mechanism we analyze persists beyond the toy setting.
>
> The primary obstacles to extending this analysis to general datasets and architectures are: (1) the lack of a simple mathematical structure for real-world data distributions and (2) the highly complex, non-linear dynamics of very deep networks. We will further discuss them in the next version.
>
> **d) A Discussion/comparison with a recent line of work explaining the implicit bias of Adam toward flat and generalizable solutions in the case of fully general models and datasets is missing; in particular: [the arxiv paper]**
>
> Thank you for pointing us to the interesting work. We will add a discussion to our Related Works section.
>
> That work analysis shows that Adam implicitly penalizes the one-norm of (perturbed) gradients, biasing solutions toward flat regions of the loss landscape. Our work offers a complementary and quantitatively precise perspective: within a feature-learning framework, we analyze how Adam/AdamW suppress noise and recover true features, controlled by batch size and weight decay.
>
> While their continuous-time view explains the emergence of flat minima, our discrete-time results clarify how this flatness coincides with robustness to noise. Together, these perspectives suggest that mini-batch Adam’s bias toward flat regions is closely tied to its ability to avoid noise overfitting and promote generalization/feature learning.
>
> **e) The language at times sounds like overstatements especially in the abstract, intro, and conclusion where the results may sound like they are proven for the general case, although they are valid only in the case of the toy example considered. Although the limitations are acknowledged, I would tone down the claims slightly**
>
> We appreciate the feedback and agree that precise language is crucial. We will revise the conclusion to avoid overstatements. We also note that these statements are already mentioned in the abstract (Lines 7–8), introduction (49–51), problem setup (159–164), and limitation (359–362).
>
> ### Questions
>
> **a) In Definition 3.1, condition 2 in unclear. From skimming through the literature mentioned, I was able to understand that one wants the noise vector to be perpendicular to the signal vector. Is that what is meant in condition 2?**
>
> Thank you for asking for this clarification. The procedure in condition 2 is designed to create a sparse noise vector $\xi$ where the randomly generated Gaussian noise component has a support that is disjoint from the support of the signal vector $v$. Specifically,$v$ is 1-sparse on coordinate $\{1\}$ , while the random Gaussian noise is generated on $s$ coordinates chosen from $\{2,\ldots,d\}$. This disjoint support is a key feature. The final noise vector $\xi$ is not strictly orthogonal to the signal vector $v$ because of the added feature noise term $−\alpha yv$, which makes the learning problem more challenging and realistic.
>
> **b) I am struggling to understand the significance of Corollary 4.6. Is there a way to rephrase it to make its significance clearer?**
>
> Certainly. The significance is that it provides a clear theoretical gap between effective weight decay regimes for Adam and AdamW:
> $$
> \begin{align*}
> \lambda_{\mathrm{Adam}} \sim \sigma_0^{q-2}\ll \frac{B^2}{n}\wedge 1 \sim \lambda_{\mathrm{AdamW}}
> \end{align*}
> $$
>
> - For **Adam**, the effective weight decay λ must be extremely small (larger than 0), as its upper bound is tied to an initialization dependent parameter ($\sigma_0^{q-2}$).
> - For **AdamW**, the effective range is more independent, with any $\lambda = \widetilde\Omega(\frac{B^2}{n}\wedge 1)$ up to $\widetilde O(1)$, i.e., a broad, constant‑order window independent of intialization.
>
> This result provides the first theoretical explanation for the widely-observed empirical fact that Adam is highly sensitive to weight decay and requires careful tuning, while AdamW's decoupled design makes it significantly more robust (e.g., Loshchilov and Hutter, 2019). Our experiments in Figure 2 validate this finding. We will revise the discussion following the corollary to state this significance more directly.
>
> **limitation please see Weakness e).**

---

### Official Review · Reviewer_vcGR · 2025-07-03

**Clarity:** 3
**Significance:** 3
**Originality:** 2
**Rating:** 5
**Confidence:** 3

**Summary:**

The paper studies the trajectory of Adam and AdamW on a classification problem parametrized by a 2-layer neural network with symmetric weights. The results extend the analysis in previous works to the small batch-size settings. Theorem 4.2 and 4.4 show that, when batch size is sufficiently small compared to sample size, Adam and AdamW can converge into solutions with not only small training error, but also small test error.

**Questions:**

1) The batch size discussed in Theorem 4.2 is very small $B=O\left(\frac{n}{\log\frac{1}{\epsilon}}\right)$. What happens if $\epsilon$ is very small and $B$ is larger than the bound? Is the $O\left(\frac{1}{\log\frac{1}{\epsilon}}\right)$ upper bound on $n$ necessary for the $o(1)$ population risk?

2) Could authors summarize a few notes to practitioners about how to choose batchsize?

3) In line 248, why do we need to add feature to the noise vector $\xi$?

**Ethical Concerns:**

["NO or VERY MINOR ethics concerns only"]

**Final Justification:**

The rebuttal stated the theoretical innovations and the importance to fine-tune weight-decay and batch size together.

**Limitations:**

yes

**Quality:**

3

**Strengths And Weaknesses:**

Strengths

1) The paper tackles an important problem about the convergence of Adam and AdamW. The finding that small batch-size training improves generalization performance aligns with practitioners' findings.

2) Theoretical presentations are clear and understandable.

Weaknesses

1) It seems many technical developments are based on Zou et al. 2023, authors could further elaborate on the technical contribution, especially the novel analytical tools introduced to analyze the mini-batch Adam.

2) According to line 145, the noise $\xi$ is generated to be sparse. This is usually not the case in practice. Authors should justify such an assumption.

---

> ### Author Rebuttal · Authors · 2025-07-31
>
> We thank the reviewer for their clear summary and positive assessment of our work—particularly noting the extension to small batch-size settings and the alignment between our theoretical findings and practitioners’ observations. We’re also glad the presentation of our theorems and proofs resonated as clear and understandable. Below, we address your questions in detail.
>
> ### Weaknesses
>
> **a) It seems many technical developments are based on Zou et al. 2023, authors could further elaborate on the technical contribution, especially the novel analytical tools introduced to analyze the mini-batch Adam.**
>
> Thank you for this insightful question. While we build on Zou et al. (2023), our analysis of mini-batch Adam/AdamW introduces several novel technical tools to (i) control the stochastic, momentum-based instabilities unique to the mini-batch regime and (ii) handle the distinct, decoupled weight decay mechanism of AdamW:
>
> 1. Stochasticity induces momentum instability. Full-batch Adam simply aggregates deterministic gradients, but mini-batch sampling injects both noise and temporal "inertia" via Adam’s moving averages. These fluctuations can cause "noise spikes" that persist across iterations. We address this with a local, one-epoch decomposition that tracks the dynamics of the momentum buffers over the $\frac{n}{B}$ steps per epoch. Lemmas D.2 and D.11 then establish a rigorous sign-GD approximation bound, proving that—after sufficient mini-batch steps—the momentum direction realigns with the true gradient rather than lingering noise.
> 2. Decoupled weight decay in AdamW. In Adam, weight decay simply augments the raw gradient and is processed through the sign approximation. AdamW instead applies decay directly to the normalized momentum ($m/(\sqrt{v}+\epsilon)$), so decay and stochastic gradients jointly determine the update. We handle this via a refined iteration-drift analysis that carefully tracks how the last few momentum updates evolve under decoupled weight decay. This enables us to show that—when batch size and decay strength are co-tuned—mini-batch AdamW exhibits the same noise-suppression "phase transition" as mini-batch Adam, with even cleaner control over the regularization path.
>
> **b) According to line 145, the noise $\xi$ is generated to be sparse. This is usually not the case in practice. Authors should justify such an assumption.**
>
> Modeling the noise as sparse with disjoint support is indeed a simplification, but it is a standard abstraction in the theoretical feature-learning literature (citer in Lines 135–136) and lays the groundwork for valuable future extensions—in particular, extending to more general noise models (see below). By isolating a few small, informative “feature patches” within a vast field of irrelevant pixels or sensor noise, we obtain a tractable framework to dissect the competition between feature learning and spurious noise memorization. One can view our two-layer CNN as capturing an intermediate representation in a deeper network whose inputs follow a “sparse + background” distribution (Papyan et al. 2017; Yang 2019).
>
> In real data, noise is rarely perfectly disjoint. However, our analysis naturally extends to overlapping or structured noise by leveraging concentration bounds on the aggregate noise energy (e.g., modeling each pixel’s perturbation as sub-Gaussian and applying matrix- or vector-Bernstein inequalities). These tools demonstrate that, provided the total noise remains controlled, the same qualitative phase transition—where feature learning dominates below a threshold and noise memorization takes over above it—continues to hold. Extending our model to this general noise setting is a future work of ours.
>
> ### Questions
>
> **a) The batch size discussed in Theorem 4.2 is very small $B=O(\frac{n}{\log\frac{1}{\epsilon}})$. What happens if $\epsilon$ is very small and $B$ is larger than the bound? Is the $O(\frac{1}{\log\frac{1}{\epsilon}})$ upper bound $n$ on necessary for the $o(1)$ population risk?**
>
> Thank you for this sharp question. Let's clarify the bound and the role of $\epsilon$. In our paper, we take $\epsilon=\Theta(\lambda\eta)$. Since $\eta$ equals to $1/\mathrm{poly}(n)$, one has that $\epsilon$ can be polynomially small $1/\mathrm{poly}(n)$. Therefore, $\log(1/\epsilon)=\Theta(\log n)$, and the stated bound becomes $B\le O(n/\log n)$.
>
> - **If B is larger than this bound:** Our theoretical analysis no longer guarantees good generalization. Indeed, our large-batch theorems (Theorem 4.1 & 4.4) and empirical results (Figure 1) show that performance degrades as $B$ increases towards $n$.
> - **Is the bound necessary?** The bound is a sufficient condition that arises from our proof technique. The core mechanism it ensures is that the number of iterations per epoch ($n/B$) is large enough for Adam's momentum estimates to "forget" the gradients from any single, sparse noise patch. If $n/B$ is too small (i.e., $B$ is too large), this forgetting mechanism fails, and noise can accumulate, leading to noise overfitting. So while the bound might not be strictly necessary, our theory clearly demonstrates a phase transition where performance breaks down as the batch size becomes too large.
>
> **b) Could authors summarize a few notes to practitioners about how to choose batchsize?**
>
> Certainly! We offer the following practitioner‐focused guidelines on choosing batch size when using Adam/AdamW:
>
> 1. Smaller batches often generalize better. Our theory provides formal support for the common heuristic that smaller batch sizes can improve generalization with Adam and AdamW. Practitioners should be cautious when scaling to very large batch sizes with these optimizers.
> 2. Batch size and weight decay must be co‐tuned. These two hyperparameters interact: you cannot optimize one in isolation. Concretely, our analysis shows that mini‐batch Adam/AdamW requires $\frac{n}{B}\ge\log(\lambda\eta)^{-1}$ to control noise memorization. Thus, if you increase the batch size $B$ to boost hardware throughput, you should also increase the weight decay $\lambda$ (albeit only logarithmically) to maintain the same generalization behavior.
> 3. Find the “sweet spot” for throughput vs. accuracy.There is a trade‐off between hardware efficiency (which favors large batches) and generalization (which favors small batches). Our experiments (Figure 1) reveal a critical batch‐size threshold beyond which test error rises sharply. We recommend practitioners perform a simple grid search around that threshold to identify the largest batch size that does not compromise test performance.
>
> **c) In line 248, why do we need to add feature to the noise vector $\xi$?**
>
> This refers to the term $-\alpha y v$ added to the noise patch $\xi$. Its purpose is to make the learning problem more challenging and realistic (avoid complete orthogonality, a trivial case). In real-world data, noise is not always purely random but can contain spurious correlations (e.g., consistent background clutter in images) that are misleading.
>
> Technically, introducing this negative correlation between the noise patch and the true feature $v$ means that an optimizer that naively fits the noise patch will be actively pushed away from the correct feature direction. This setup allows our theory to crisply differentiate algorithms that successfully isolate the true feature from those that are misled by the noise, which is essential for proving the failure mode of large-batch Adam/AdamW.

---

> > ### Comment · Reviewer_vcGR · 2025-08-05
> > **Thank authors for the response**
> >
> > My concerns have been addressed. I will raise my score to support the paper.

---

> > > ### Author Response · Authors · 2025-08-06
> > >
> > > Thank you for revisiting our paper and confirming that our rebuttal has addressed your concerns. We appreciate your score increase and support. Your valuable feedback has greatly strengthened our work.

---

### Official Review · Reviewer_a6No · 2025-07-03

**Clarity:** 4
**Significance:** 3
**Originality:** 3
**Rating:** 5
**Confidence:** 4

**Summary:**

This paper studies adam and adamw and constructs a setting where the stochastic versions of these algorithms generalize and achieve training loss zero while large batch adam/adamw also get zero training loss but do not generalize. They first present their setting inspired by the classical feature-noise patch concatenation framework, the CNN architecture, the Adam/AdamW algorithms. Then, they present in Theorem 4.1 and Theorem 4.4 the large batch Adam/AdamW results showing that using these algorithms, the test error is large. On the other hand, in Theorems 4.2 and 4.5, they show that when using a mini-batch, the two algorithms achieve perfect test error.  They also show theoretically that AdamW can take larger weight decay than Adam. Finally, they validate their results on Cifar-10 using VGG and Resnet architectures. They swept over batch size in Figure 1 showing that the test error is low for small batch size and swept over weight decay in Figure 2, showing that AdamW can take larger weight decay than Adam.

**Questions:**

I listed my questions in the weaknesses section.

**Ethical Concerns:**

["NO or VERY MINOR ethics concerns only"]

**Limitations:**

I think the limitations section is a bit incomplete. I would add the following:

- In the limitations section, I would also mention that important knobs have been dismissed in the analysis: momentum, learning rate schedule, epsilon, gradient clipping.

- Also maybe in the Limitations section, I would maybe mention that current vision non-Transformer architectures actually use large batch sizes? E.g. ConvNext, PeLK, Sparse Large-kernel Network (SLaK). And this seems something that maybe your setting fails to captures. This is totally fine but maybe worth mentioning.

**Paper Formatting Concerns:**

No concern.

**Quality:**

3

**Strengths And Weaknesses:**

Strengths:

- This paper is very well-written. The related work is appropriately cited, the setting is crystal-clear and the theorems are clear and I appreciated a lot the proof sketch and intuitions that are given.
- The result is also interesting. To my knowledge,  First formal proof that the mini-batch variant of an adaptive method can strictly outperform its full-batch counterpart in a non-convex neural-network setting. Previous theory had only done this comparison for GD vs SGD or for convex problems.
- The result on the upper bound for Adam’s weight decay is also very nice.

Weaknesses:

- Here there are two components that are in a way studied: stochastic gradients and the adaptivity. The issue is that in the current paper, the contribution of each is not crystal-clear and I think the authors would gain at clarifying this. For instance, a result that would be interesting would be to show that in your framework, SGD does not get zero test error. I think that adding this result would be very nice, and would allow us to understand the importance of adaptivity in your setting.

- Related to this previous point, I think that it would be nice maybe in the proof sketch section to maybe write the optimization update of Adam so that we clearly understand the contribution of adaptivity at getting rid of the feature noise.

- I think your analysis is interesting regarding batch size. Can you maybe explain what happens in your analysis for intermediate batch size? You cover small batch size and very large batch size.

- Can we derive some optimal batch size B? This optimal batch size would be the middle-ground between speed (large batch size allows for fast updates) and small batch size (small batch size allows for generalization)?

- I think a better job could be made in the experiments section. In particular, large scale vision experiments could be conducted beyond CIFAR-10: maybe training using Resnet-50 on Imagenet. Besides, I think it would be important to do a more fine-grained sweep over beta_1, beta_2 and the learning rate for Figures 1 and 2.

- Lastly, one may wonder how incremental your derivation is from Zou et al. 2023. I think it is important to further highlight the differences of the authors' analysis and Zou et al.'s paper.

---

> ### Author Rebuttal · Authors · 2025-07-31
>
> We thank the reviewer for clear and accurate summary of our work, and for highlighting its strengths—namely the manuscript’s clarity, the novel theoretical proof that mini-batch Adam/AdamW can strictly outperform full-batch variants in a non-convex neural-network setting, and the elegant upper bound on Adam’s weight decay. Below, we address each of your questions in turn.
>
> ### Weaknesses & Questions
>
> **a) Here there are two components that are in a way studied: stochastic gradients and the adaptivity. The issue is that in the current paper, the contribution of each is not crystal-clear and I think the authors would gain at clarifying this. For instance, a result that would be interesting would be to show that in your framework, SGD does not get zero test error. I think that adding this result would be very nice, and would allow us to understand the importance of adaptivity in your setting.**
>
> This is a crucial point that we will clarify. Our work primarily studies the interplay between stochastic gradients (batch size) and weight decay within the context of adaptive methods (Adam and AdamW).
>
> Prior work, which we build upon (Zou et al., 2023), has already shown that in a similar setting, full-batch SGD generalizes well, while full-batch Adam fails. This establishes that adaptivity is the source of the large-batch generalization failure. The reason is that adaptivity induces an $\ell_1$-like normalization—approximating the behavior of SignSGD—which inherently favors sparse gradients. In our theoretical setting, this mechanism inadvertently amplifies sparse noise memorization when the noise level satisfies $s\sigma_p = \omega(1)$. As a result, in the full-batch regime (where $s \gg 1$ and batch stochasticity is absent), the rate of noise memorization outpaces feature learning, ultimately leading to poor generalization. This effect is analytically captured in the update dynamics of AdamW along the noise direction:
> $$
> \begin{align*}
> \langle w_{y_i,r}^{(t+1)},\xi_i\rangle= (1-\lambda\eta)\langle w_{y_i,r}^{(t)},\xi_i\rangle +\eta\cdot\sum_{k\in\mathcal{B}\_i}\langle\mathrm{sgn}(\ell_{y_i,i}^{(t)}\sigma'(\langle w_{y_i,r}^{(t)},\xi_i\rangle)\xi_i[k]),\xi_i[k] \rangle -\alpha y_i\eta\cdot\mathrm{sgn}(\sum_{j\in\mathcal{I}\_t}y_j\ell_{y_i,j}^{(t)}[\sigma'(\langle w_{y_i,r}^{(t)},y_j v\rangle) - \alpha\sigma'(\langle w_{y_i,r}^{(t)},\xi_j\rangle)]).
> \end{align*}
> $$
> Our key contribution is to show that stochasticity (from mini-batch), when combined with proper weight decay, rescues the adaptive method. It is the synergy between these components that yields good generalization: 1) adaptivity with exponential moving average normalizes gradients coordinate-wise; 2) stochasticity ensures that noise components are sparse and decorrelated across batches; 3) weight decay provides a constant regularization force that suppresses noise components over time. As we explain in the paper (Lines 215-223), this synergy allows Adam to maintain learning on shared features while its normalization and momentum dynamics, aided by weight decay, suppress the learning of sparse noise.
>
> **b) Related to this previous point, I think that it would be nice maybe in the proof sketch section to maybe write the optimization update of Adam so that we clearly understand the contribution of adaptivity at getting rid of the feature noise.**
>
> Good point! While the detailed update rules for Adam and AdamW are in Appendix C (Eqs. C.3–C.6), we agree it would be helpful to highlight them in the main paper. Section 5 builds on SignSGD, which we rigorously justify as an approximation of stochastic Adam (Appendix D, Lemma D.2). Adaptivity manifests through the `sgn` operator, normalizing update magnitudes. In large-batch settings, this amplifies noise; in contrast, in the mini-batch regime, stochasticity slows noise fitting, and weight decay suppresses it—together mitigating feature noise.
>
> **c) I think your analysis is interesting regarding batch size. Can you maybe explain what happens in your analysis for intermediate batch size? You cover small batch size and very large batch size.**
>
> This is an excellent question. Our theory provides proofs for two distinct regimes to establish a clear, provable separation in behavior. A precise theoretical characterization of the intermediate batch size regime (e.g., for Adam, $\frac{n}{B} = \omega(1)$ and $O(\log\epsilon^{-1})$) is significantly more complex due to the limitations of our order-wise analysis. One must carefully track how the factors \($\beta_1^{\tau_0}$) and \($\beta_2^{\tau_0/2}$) evolve over \($\tau_0\in[n/B]$) steps to decide whether the sign‐SGD–style gradient term or the decoupled weight-decay term dominates. Concretely, our update in the noise coordinates (Line 1205) reads:
> $$
> \begin{align*}
>     & \frac{m_{j,r}^{(t)}[k]}{\sqrt{v_{j,r}^{(t)}[k]}+\epsilon} = \frac{\beta_1^{\tau_0}\left(\tilde\Theta\Big(g_{t-\tau_0,j,r}^{(t)}[k]\Big) \pm \tilde\Theta\bigg(\frac{\eta s\sigma_p|\ell_{j,i}^{(t)}|}{B}\bigg)\pm \tilde\Theta(\lambda\eta)-\tilde\Theta(\lambda w_{j,r}^{(t)}[k])\right)+\tilde\Theta(\lambda w_{j,r}^{(t)}[k])\pm\tilde\Theta(\lambda\eta) }{\beta_2^{\frac{\tau_0}{2}}\left(\Theta\Big(|g_{t-\tau_0,j,r}^{(t)}[k]|\Big) \pm \tilde\Theta\bigg(\frac{\eta s\sigma_p|\ell_{j,i}^{(t)}|}{B}\bigg)\pm \tilde\Theta(\lambda\eta)-\Theta(|\lambda w_{j,r}^{(t)}[k]|)\right)+\tilde\Theta(|\lambda w_{j,r}^{(t)}[k]|)\pm\tilde\Theta(\lambda\eta)},
> \end{align*}
> $$
> For intermediate $B$, our order-wise bounds lose precision: it becomes hard to determine whether momentum gradient or weight decay will dominate, and thus impossible (with current techniques) to prove exactly how dynamics evolves. Nevertheless, our empirical sweep in Figure 1 covers these middle values of \(B\) and reveals a decline in test accuracy as batch size grows—suggesting a “critical threshold” beyond which generalization degrades sharply. Fully characterizing this intermediate regime is an important direction for future work.
>
> **d) Can we derive some optimal batch size B? This optimal batch size would be the middle-ground between speed (large batch size allows for fast updates) and small batch size (small batch size allows for generalization)?**
>
> Deriving an exact optimal batch size is theoretically difficult and remains an open direction. However, our results provide useful guidance: good generalization requires small enough batches to satisfy the conditions in Theorems 4.2 or 4.5. This highlights a trade-off—smaller $B$ supports generalization, while larger $B$ offers faster training. In practice, the optimal batch size lies at this intersection. As shown in Figure 1, there exists a range of small batch sizes that generalize well before a sharp drop in accuracy, allowing practitioners to select a computationally efficient $B$ within this range.
>
> **e) I think a better job could be made in the experiments section. In particular, large scale vision experiments could be conducted beyond CIFAR-10: maybe training using Resnet-50 on Imagenet. Besides, I think it would be important to do a more fine-grained sweep over beta_1, beta_2 and the learning rate for Figures 1 and 2.**
>
> We appreciate this valuable suggestion. We'll construct an ImageNet-1K subset (randomly sampling 100 training images per class with seed = 0) to ensure a controlled large-batch regime (so that $\frac{n}{B}$ remains $\Theta(1)$) while keeping overall compute manageable. We will train ResNet-50 on this subset and, for Figures 1 and 2, also include a systematic sweep over momentum coefficients such like $\beta_1\in [0, 0.5, 0.9]$, $\beta_2\in [0.5, 0.9, 0.95]$ (learning rates have already been swept; see Lines 274–282). These results will appear in the next version, and if our runs finish in the coming days, we'll report them here.
>
> **f) Lastly, one may wonder how incremental your derivation is from Zou et al. 2023. I think it is important to further highlight the differences of the authors' analysis and Zou et al.'s paper.**
>
> Thank you for this insightful question. While we build on Zou et al. (2023), our mini-batch Adam/AdamW analysis introduces two key novelties:
>
> 1. Momentum instability control: We use a one-epoch local decomposition (Lemmas D.2, D.11) to rigorously bound how stochastic “noise spikes” in Adam’s moving averages realign with the true gradient after sufficient steps.
> 2. Decoupled weight-decay analysis: A refined iteration-drift framework tracks how AdamW’s direct decay on normalized momentum interacts with stochastic gradients, proving that co-tuning batch size and decay strength yields the same noise-suppression phase transition as in mini-batch Adam.
>
> More details can be found in the response to question a) of the reviewer vcGR.
>
> ### Limitations
>
> **I think the limitations section is a bit incomplete. I would add the following:**
> - **In the limitations section, I would also mention that important knobs have been dismissed in the analysis: momentum, learning rate schedule, epsilon, gradient clipping.**
>
> Thank you for the insightful suggestion. We acknowledge that other critical hyperparameters—such as momentum ($\beta$ values), learning rate schedules, and gradient clipping—are not covered in our theory. Understanding their combined effect on generalization remains a complex and largely open question in the community. We will expand on it in the next version.
>
> - **Also maybe in the Limitations section, I would maybe mention that current vision non-Transformer architectures actually use large batch sizes? E.g. ConvNext, PeLK, Sparse Large-kernel Network (SLaK). And this seems something that maybe your setting fails to captures. This is totally fine but maybe worth mentioning.**
>
> Thank you for the insight. While our theory predicts challenges with large batches, some modern architectures succeed with them—possibly due to factors like normalization or architectural design that our model simplifies. We will add this as a future research direction.

---

> ### Author Response · Authors · 2025-08-07
>
> We would like to add more experimental results based on your previous questions. It takes some time and we just finished the experiments.
>
> **e) I think a better job ... large-scale vision experiments could be conducted beyond CIFAR-10 ..., more fine-grained sweep over beta_1, beta_2 and the learning rate for Figures 1 and 2.**
>
> These experiments further confirm our conclusions. The details are provided below:
>
> ## 1. Large-Scale Vision Experiments with ResNet-50 on ImageNet-1K Subset
>
> We constructed an ImageNet-1K subset (100 training images per class, seed=0) to maintain a controlled large-batch regime ($\frac{n}{B} = \Theta(1)$) while managing compute. We trained ResNet-50 on this subset for 90 training epochs, with standard ImageNet preprocessing. The key results as follows:
>
> - **Top5 Validation Error vs. Batch Size**: We tested learning rates (1e-4) with $\beta_1=0.9$, $\beta_2=0.99$, and weight decay settings (Adam: 1e-4; AdamW: 1e-2). Results are shown as follows, confirming our conclusions on batch size sensitivity for Adam and AdamW.
>
> ResNet50: Top5 Val Err (%) vs. Batch Size
>
> | Model      | 64    | 128   | 256   | 512   | 1024  | 2048  | 3072  | 4096  | 8192  | 16384 | 32768 |
> | - | - | - | - | - | - | - | - | - | - | - | - |
> | Adam | 30.31 | 33.65 | 37.32 | 39.06 | 44.06 | 48.26 | 50.89 | 53.68 | 62.88 | 72.55 | 83.69 |
> | AdamW | 32.27 | 35.27 | 38.13 | 39.59 | 43.35 | 48.27 | 51.04 | 54.76 | 61.97 | 72.52 | 83.90 |
>
> - **Top5 Validation Error vs. Weight Decay**: With batch size=64, lr=1e-3, and $\beta_1=0.9$, $\beta_2=0.99$, we analyzed weight decay ($\lambda$) effects. The following table show the results, confirming AdamW’s stability across broader decay ranges.
>
> ResNet50: Top5 Val Err (%) vs. Weight Decay
>
> | Optimizer | 5e-7  | 1e-6  | 5e-6  | 1e-5  | 5e-5  | 1e-4  | 5e-4  | 1e-3  | 5e-3  | 1e-2  | 5e-2  | 1e-1  | 5e-1  |
> | - | - | - | - | - | - | - | - | - | - | - | - | - | - |
> | Adam | 35.81 | 35.08 | 32.16 | 33.63 | 40.21 | 45.76 | 71.11 | 80.09 | 99.50 | 99.50 | 99.50 | 99.50 | —     |
> | AdamW | —     | —     | —     | —     | —     | 34.09 | 37.01 | 35.36 | 39.91 | 34.70 | 33.52 | 40.27 | 87.22 |
>
> ## 2. Fine-Grained Sweep over $\beta_1$ and $\beta_2$ for Figures 1 and 2
>
> As suggested, we performed a systematic sweep over momentum coefficients, with $\beta_1\in[0, 0.5, 0.9]$, $\beta_2\in[0.5, 0.9, 0.95]$, plus the original $\beta_1=0.9$, $\beta_2=0.99$ (total 9 combinations, satisfying $\beta_1^2 < \beta_2$). Learning rates were already swept (Lines 274–282), and key results for Figures 1 and 2 are as follows:
>
> - **Figure 1 (Test Error vs. Batch Size)**: Results (mean±std) for Adam and AdamW on VGG16/ResNet18 confirm that our conclusions on batch size behavior hold across $\beta$ combinations.
>
> Adam: Test Err (%) vs. Batch Size
>
> | Model    | 16         | 32         | 64         | 256        | 1024       | 4096       | 8192        |
> | - | - | - | - | - | - | - | - |
> | VGG16    | 15.46±0.70 | 16.49±1.63 | 15.94±1.22 | 18.86±6.83 | 19.73±1.46 | 26.22±4.27 | 32.05±5.57  |
> | ResNet18 | 15.87±0.37 | 15.98±0.62 | 17.10±1.50 | 17.80±1.27 | 19.52±1.19 | 28.36±6.24 | 32.31±15.31 |
>
> AdamW: Test Err (%) vs. Batch Size
>
> | Model    | 16         | 32         | 64         | 256        | 1024       | 4096       | 8192        |
> | - | - | - | - | - | - | - | - |
> | VGG16    | 14.75±0.65 | 14.34±0.66 | 13.91±0.49 | 14.56±1.03 | 17.98±1.94 | 20.76±3.52 | 28.96±14.55 |
> | ResNet18 | 15.63±0.47 | 15.43±0.65 | 15.21±0.28 | 16.06±1.05 | 18.92±0.88 | 25.53±4.93 | 29.04±8.33  |
>
> - **Figure 2 (Test Error vs. Weight Decay)**: Results (mean±std) for ResNet18/VGG16 show AdamW’s robustness across $\lambda$ ranges, consistent with our original findings.
>
> ResNet18: Test Err (%) vs. Weight Decay
>
> | Optimizer | 5e-7       | 1e-6       | 5e-6       | 1e-5       | 5e-5       | 1e-4       | 5e-4       | 1e-3       | 5e-3       | 1e-2       | 5e-2       | 1e-1       | 5e-1       |
> | - | - | - | - | - | - | - | - | - | - | - | - | - | - |
> | Adam      | 15.71±0.62 | 15.60±0.72 | 15.87±0.38 | 15.80±0.64 | 15.93±0.45 | 15.91±0.28 | 15.78±0.48 | 15.83±0.30 | 15.47±0.56 | 16.05±0.34 | 17.47±0.92 | 22.00±1.46 | —          |
> | AdamW     | —          | —          | —          | —          | —          | 15.30±0.48 | 15.42±0.40 | 15.35±0.50 | 15.37±0.36 | 15.63±0.47 | 16.03±0.68 | 16.27±0.69 | 16.44±0.52 |
>
> VGG16: Test Err (%) vs. Weight Decay
>
> | Optimizer | 5e-7       | 1e-6       | 5e-6       | 1e-5       | 5e-5       | 1e-4       | 5e-4       | 1e-3       | 5e-3       | 1e-2       | 5e-2       | 1e-1       | 5e-1       |
> | - | - | - | - | - | - | - | - | - | - | - | - | - | - |
> | Adam      | 14.93±0.66 | 14.90±0.60 | 15.13±0.72 | 15.25±0.85 | 15.32±0.54 | 15.28±0.62 | 15.46±0.70 | 15.66±0.87 | 16.25±0.55 | 16.84±0.90 | 18.61±1.77 | 27.07±3.12 | —          |
> | AdamW     | —          | —          | —          | —          | —          | 14.62±0.39 | 14.63±0.53 | 14.63±0.41 | 14.79±0.65 | 14.75±0.65 | 15.38±0.63 | 15.62±0.62 | 17.46±1.31 |

---

### Official Review · Reviewer_dSx3 · 2025-07-11

**Clarity:** 4
**Significance:** 3
**Originality:** 3
**Rating:** 5
**Confidence:** 3

**Summary:**

The paper studies the generalization error of a classification task with a two-layer neural network for Adam and AdamW. The findings are: 1) the two algorithms both perform better when the batch size is small, and 2) AdamW allows a larger regularization parameter $\\lambda$ while Adam's parameter selection is restricted to a small $\\lambda$ regime, consistent with practical observations.

**Questions:**

Please refer to the points under Weaknesses.

**Ethical Concerns:**

["NO or VERY MINOR ethics concerns only"]

**Final Justification:**

After the author's detailed explanation during the rebuttal, I believe my original score is reasonable for this paper and I intend to maintain it. The paper makes a solid contribution and employs sound techniques. Therefore, I recommend acceptance.

**Limitations:**

yes

**Quality:**

4

**Strengths And Weaknesses:**

Strengths:
1. The paper is well-written and easy to follow.
2. The technical contribution is solid. It provides an explanation as to why Adam/AdamW is better in a stochastic setting, i.e., when the batch size is small, and why AdamW allows for a larger regularization parameter.
3. The author has a good discussion that links the implications to the proofs in the theorems. Specifically, the feature learning vs. noise learning in the two stages.

Weaknesses:
1. __Regarding the setting.__ I am not very familiar with the data model setting in the paper, and I understand that many practical scenarios are simplified for ease of analysis, common in theorems for modern machine learning. However, how this data model relates to reality and to what extent it can be generalized, e.g., to multiple layers and with wide range of parameters, is unknown to me. Specifically, does the noise model resemble some cases in practice? How realistic is the parameter selection, like $s$, $\sigma_p$, and $\\alpha$, and why are they set to the order in the paper?
2. __Regarding parameter choices of Adam/AdamW.__ 1) Are the (order of) parameter selections, for example $\lambda$ and $B$, based on practice or to achieve the best theoretical bounds? 2) For the case of AdamW, the author used a large $\lambda$; however, they do not have a theoretical negative result on small $\lambda$. It seems that in practice a small $\\lambda$ also works in the experiments? From the theorems, how can we know that AdamW is more robust to parameter selection? For Adam, it seems that it only requires $\lambda$ to be small enough (what if we set it to be 0?), while AdamW requires the parameter to be tuned to a regime with upper and lower bounds. 3) The batch size choices in Theorem 4.2 and 4.5 are also different, with AdamW's being more restrictive. Then, how should we compare the two results? 4) From an optimization point of view, to achieve the same error, should the time $T$ and step size $\eta$ be set differently in the mini-batch and large-batch cases? For example, with a large batch, the noise is small, and the algorithm should converge faster. Will early stopping the training of large batches help?
3. __What happens to other algorithms.__ Under the same data model setting, what would be the performance for SGD, SGD with weight normalization, and Adam without weight normalization? Can we have more insights comparing also with these algorithms or to design better algorithms? This might be future directions.

---

> ### Author Rebuttal · Authors · 2025-07-31
>
> We thank the reviewer for thorough evaluation and constructive feedback, particularly acknowledging the manuscript’s clarity, solid technical contributions, and insightful feature-vs-noise learning discussion. Below, we address each of your questions in turn.
>
> ### Weaknesses & Questions
>
> **1. Regarding the setting.**
>
> **a) how this data model relates to reality and to what extent it can be generalized, e.g., to multiple layers and with wide range of parameters, is unknown to me.**
>
> Our feature–noise patch model—widely used in theory (Lines 135-136)—isolates the trade‐off between learning sparse, generalizable features and memorizing noise. Real images can be viewed as a few informative patches amid distractors, and studying a two‐layer CNN reflects the behavior of intermediate layers in deeper nets (Papyan et al., 2017; Yang, 2019). Moreover, experiments validate our theoretical predictions (Figs. 1–6). Extending our proofs to deeper, wider architectures remains a valuable direction.
>
> **b) Specifically, does the noise model resemble some cases in practice?**
>
>  Our noise model reflects real images, where sparse label-relevant patches (“signal”) are embedded in large irrelevant regions (“noise”). Empirical evidence shows CNN activations are sparse and treat distant backgrounds as independent noise (Papyan et al., 2017; Yang, 2019), supporting the realism and tractability of our model. Viewing it as an intermediate-layer input further highlights its practical relevance.
>
> **c) How realistic is the parameter selection, like $s$, $\sigma_p$, and $\alpha$, and why are they set to the order in the paper?**
>
> We select $s=\Theta(\frac{d^{1/2}}{n^2})$, $\sigma_p^2=\Theta(\frac{1}{s\cdot\mathrm{polylog}(n)})$ and $\alpha = \Theta(\sigma_p \cdot \mathrm{polylog}(n))$ (App B.1) so that only a tiny fraction of the input ("signal") convey class information, while $s$ coordinates serve as iid noise. These settings align with practice.
>
> By scaling $\sigma_p$ to match the feature magnitude and letting $\alpha\in(0,1)$ model misleading noise, we create a setting in which feature and noise contributions are initially comparable. Lemma B.3 shows that the feature learning $\langle w_{j,r}, jv\rangle$ and noise memorization $\langle w_{y_i,r}, \xi_i\rangle$ start at the same order, preventing a bias toward either true features or noise at initialization, highlighting the rigor of our conclusions.
>
> **2. Regarding parameter choices of Adam/AdamW.**
>
> **a) Are the (order of) parameter selections, for example $\lambda$ and $B$, based on practice or to achieve the best theoretical bounds?**
>
> Our theorems select parameter orders (e.g., $\frac{n}{B} = \Theta(1)$ or $\ge \Theta(\log \epsilon^{-1})$) for Adam, and $\lambda=\widetilde\Omega(\frac{B^2}{n}\wedge 1)$ up to $\widetilde O(1)$ for AdamW) to prove distinct learning regimes rigorously. Crucially, these bounds capture familiar empirical patterns: smaller batch sizes yield better generalization for both Adam and AdamW, and AdamW tolerates much larger $\lambda$ values. These insights agree with practical observations (Loshchilov & Hutter, 2019) and match our experiments (Figs. 1–2).
>
> **b) For the case of AdamW, the author used a large $\lambda$; however, they do not have a theoretical negative result on small $\lambda$. It seems that in practice a small $\lambda$ also works in the experiments?**
>
> Our bound for AdamW, $\lambda = \widetilde\Omega(\frac{B^2}{n}\wedge 1)$, is a sufficient condition to ensure weight decay is strong enough to halt further growth of noise memorization. Concretely, as shown by Eqs. C.5–C.6, noise memorization $\langle w_{y_i,r}^{(t)}, \xi_i\rangle$ grow to $\widetilde\Theta(\frac{Bs\sigma_p}{n})$ before weight decay takes effect. Lemma D.18 then requires $\lambda$ of this order to prevent additional noise amplification, and Lemma D.20 shows that only under this regime can mini‑batch AdamW drive test error to near zero.
>
> In practice, small $\lambda$ may outperform random guessing, but optimal accuracy typically requires a non‑trivial lower bound (Loshchilov & Hutter, 2019).  We will augment the manuscript to make these explicit.
>
> **c) From the theorems, how can we know that AdamW is more robust to parameter selection? For Adam, it seems that it only requires $\lambda$ to be small enough (what if we set it to be 0?), while AdamW requires the parameter to be tuned to a regime with upper and lower bounds.**
>
> AdamW's robustness stems from its wide effective $\lambda$ ranges. Corollary 4.3 requires Adam's $\lambda$ to be below a tight, initialization‑dependent upper bound (i.e., $\lambda = \omega(\sigma_0^{q-2})$ causes failure). Corollary 4.6 shows AdamW succeeds for any $\lambda = \widetilde\Omega(\frac{B^2}{n}\wedge 1)$ up to $\widetilde O(1)$, a constant‑order range independent of intialization. This explains why AdamW is more robust.
>
> For Adam w/ $\lambda=0$: Our conclusions need $\lambda > 0$; otherwise, the batch size condition for Adam becomes more restrictive. We will clarify this in the next version.
>
> Our analysis shows that weight decay is necessary to prevent noise overfitting by suppressing noise memorization. Without weight decay ($\lambda=0$), Adam fits both feature and noise, and generalization depends on the relative rates of feature learning ($\frac{n}{B}$) and noise memorization ($s\sigma_p$) per epoch. With $\lambda=0$, the feature learning and noise memorization updates (Eqs. C.3 & C.4) become:
>
>
> $$
> \begin{align*}
> \langle w_{j,r}^{(t+1)},jv \rangle = \langle w_{j,r}^{(t)},jv \rangle + j\eta\cdot\mathrm{sgn}( \sum_{i\in\mathcal{I}\_t}y_i\ell_{j,i}^{(t)}[\sigma'(\langle w_{j,r}^{(t)},y_iv\rangle)-\alpha\sigma'(\langle w_{j,r}^{(t)},\xi_i\rangle)]),
> \end{align*}
> $$
> $$
> \begin{align*}
> \langle w_{y_i,r}^{(t+1)},\xi_i\rangle = \langle w_{y_i,r}^{(t)},\xi_i\rangle +\eta\cdot\sum_{k\in\mathcal{B}\_i}\langle\mathrm{sgn}(\ell_{y_i,i}^{(t)}\sigma'(\langle w_{y_i,r}^{(t)},\xi_i\rangle)\xi_i[k]),\xi_i[k] \rangle -\alpha y_i\eta\cdot\mathrm{sgn}(\sum_{j\in\mathcal{I}\_t}y_j\ell_{y_i,j}^{(t)}[\sigma'(\langle w_{y_i,r}^{(t)},y_jv\rangle) - \alpha\sigma'(\langle w_{y_i,r}^{(t)},\xi_j\rangle)]),
> \end{align*}
> $$
> where $\ell_{j,i}^{(t)}:=\mathbb{1}\_{y_i=j}-\mathrm{logit}\_j(F,x)$ and $\mathrm{logit}\_j(F,x) = \frac{e^{F_{j}(w,x_i)}}{\sum_{k\in\{-1,1\}} e^{F_k(w,x_i)}}$. Since $\ell_{y_i,i}>0$ and $\alpha=o(1)$, both $\langle w_{j,r}^{(t)},jv \rangle$ and $\langle w_{y_i,r}^{(t)},\xi_i\rangle$ initially grow at rates $\Theta(\frac{n}{B})$ and $\widetilde{\Theta}(s \sigma_p)$, respectively. Later dynamics depend on which grows faster. For mini-batch Adam w/o weight decay to generalize well, the feature learning rate must dominate: $\frac{n}{B} = \omega(s \sigma_p)$, similar to AdamW. Otherwise, noise memorization overtakes feature learning, as in the large-batch case. This condition imposes a stronger batch size constraint on Adam w/o weight decay, so we have $\lambda > 0$.
>
> **d) The batch size choices in Theorem 4.2 and 4.5 are also different, with AdamW's being more restrictive. Then, how should we compare the two results?**
>
> As similar in response 2 c), AdamW’s batch size condition is stricter due to proof technicalities. AdamW’s decoupled weight decay acts only in later training (Eq. C.5–C.6). Before that, feature learning and noise memorization both grow (Lemma D.17). Weight decay kicks in once weights are large enough (Lemma D.18), requiring feature learning to outpace noise ($\frac{n}{B} = \omega(s\sigma_p)$) and noise memorization not to harm test error ($\frac{n}{B} \ge \Theta(n^{1/2})$, Lemma D.20, Lines 1615-1616). The main insight is that for both Adam and AdamW, small batch size is key to avoid failure.
>
> **e) From an optimization point of view, to achieve the same error, should the time $T$ and step size $\eta$ be set differently in the mini-batch and large-batch cases? For example, with a large batch, the noise is small, and the algorithm should converge faster. Will early stopping the training of large batches help?**
>
> Great point! However, our analysis focuses on the properties of the converged solution from a feature learning perspective, rather than optimization speed.
>
> Regarding early stopping, our analysis shows that large-batch AdamW initially learns features but eventually reverses and overfits to noise (Figure 3a & 4a), and early stopping could potentially halt the training before this reversal (recall the gradient: $\sigma'(\langle w_{j,r}^{(t)},y_i v\rangle)-\alpha\sigma'(\langle w_{j,r}^{(t)},\xi_i\rangle)$)—thus preserving some generalization.
>
> **3. What happens to other algorithms.**
>
> **a) Under the same data model setting, what would be the performance for SGD, SGD with weight normalization, and Adam without weight normalization?**
>
> Prior work (Zou et al., 2023) shows that SGD generalizes well in settings where full-batch adaptive methods fail. Regarding SGD with WN, this introduces additional dynamics beyond our current scope, but we agree it is a promising direction for future work. For Adam without weight normalization—if referring to weight decay—its performance depends on the relative rates of feature learning and noise memorization; please see our response to 2 c) for details.
>
> **b)  Can we have more insights comparing also with these algorithms or to design better algorithms? This might be future directions.**
>
> Indeed, our conclusions suggest promising directions for future work, such as designing noise-aware update rules that explicitly suppress sample-specific noise, or adopting stage-wise optimization strategies—e.g., using Adam in early training and switching to SGD later for better generalization.

---

### Comment · Area_Chair_A5Ym · 2025-08-04

Dear reviewers,

Thank you for your valuable time and your expertise in reviewing. Engaging with authors is really important, and allows both authors and reviewers to gain deeper understanding of cutting edge topics. This is a unique opportunity of interaction in our community.

The author rebuttal phase is about to close, and we kindly request your prompt attention to ensure a thorough discussion.

The discussion period **ends in less than 3 days** (on Aug. 6, 11:59pm AOE ). To maintain the review timeline, we ask that you:

- Review the rebuttals,

- Engage in any ongoing discussion with fellow reviewers/authors (if applicable),

- Finalize your assessment.

If you have already completed this step, we sincerely appreciate your efforts.

Thank you for your collaboration!

Best regards,

AC

---

### Note · Authors · 2025-08-14

We extend our sincere gratitude to all reviewers for their thorough evaluation, insightful feedback, and encouraging summaries of our work. We appreciate their recognition of the paper’s clarity, solid technical contributions, and its significance in bridging theory and practice.

As noted by Reviewers dSx3, a6No, vcGR, and qgrL, our work provides a rigorous proof that mini-batch Adam/AdamW can strictly outperform their large-batch counterparts in non-convex neural networks. This explains the widely observed empirical success of small-batch training with adaptive methods. Reviewers also valued our theoretical explanation for why AdamW tolerates a much more robust range of weight decay values than Adam (dSx3, a6No, qgrL), calling it a “very nice” and significant result that grounds a common practice in rigorous analysis.

In response to concerns on generality and realism (dSx3, vcGR, qgrL), we clarified the motivation for our feature-noise patch model and its parameter scalings, and added discussion on extensions to more complex model. On empirical validation (a6No, qgrL), we added large-scale experiments on an ImageNet-1K subset with ResNet-50 (a6No), re-ran all CIFAR-10 experiments over five random seeds for statistical robustness (qgrL), and conducted fine-grained sweeps over $\beta_1,\beta_2$ (a6No). These confirm the generality of our findings on batch-size sensitivity and weight-decay robustness.

To address questions on technical novelty (a6No, vcGR), we emphasized our two key innovations: a one-epoch local decomposition that rigorously controls momentum-induced noise spikes, and a refined analysis of decoupled weight decay. This distinction from prior work (e.g., Zou et al. 2023) is made explicit.

Finally, we carefully revised the language to avoid overstating generality in conclusion section (qgrL), linked theoretical predictions such as two-stage learning dynamics directly to empirical visualizations (Fig. 3-4) in the appendices (qgrL), and highlighted practitioner guidelines showing how batch size and weight decay should be co-tuned (vcGR).

We believe these revisions substantially improve the clarity, rigor, and practical relevance of our work, and we thank the reviewers again for their expert guidance.

---

### Decision · Program_Chairs · 2025-09-17

**Decision:**

Accept (poster)

**Comment:**

The paper at hand presents a modern, thorough analysis of Adam(W) generalization capabilities on deep neural networks. All reviewers agree that the paper deserves publication, and I think that this work is of great importance towards understanding the fundamental properties of Adam near the solution. Despite this, I find an evident lack of theoretical and practical discussion of weight decay effects on LM training, which was, for instance, explored in [A]. While this setting is different and I consider the paper great, I think the authors should also discuss modern setups in their revised version.

I recommend acceptance, also in light of the novel theoretical tools and techniques presented, which are valuable for the community.

[A] D'Angelo, Francesco, et al. "Why do we need weight decay in modern deep learning?" Advances in Neural Information Processing Systems 37 (2024): 23191-23223.